# Proteomic risk score for early prediction of kidney disease progression in individuals with *APOL1* high-risk genotypes

Individuals of African ancestry carrying *APOL1* (apolipoprotein L1) high-risk genotypes face a markedly increased risk of kidney failure, yet tools to identify those individuals likely to progress to chronic kidney disease are lacking. Here we profiled plasma proteomes of 851 Penn Medicine BioBank participants of African ancestry (285 males and 566 females) with *APOL1* high-risk genotypes and preserved estimated glomerular filtration rate (eGFR) (≥60 ml min$^{-1}$ 1.73 m$^{-2}$). Using elastic net Cox regression adjusted for age, sex, eGFR and albuminuria, we derived a nine-protein APOL1 Proteomic Risk Score (APRS) that predicts a composite outcome of ≥40% eGFR decline, kidney failure or death. APRS achieved a time-dependent area under the receiver operating characteristic curve (tAUC) of 86.5%, outperforming the Kidney Failure Risk Equation (66.1%) and polygenic risk scores, with 10-year event rates of 62.5% versus 3.3% across risk quintiles. External validation in Atherosclerosis Risk in Communities and UK Biobank cohorts confirmed robust accuracy (tAUC 82–85%) and consistent performance across demographic and clinical subgroups. Plasma levels of APRS component proteins correlated with kidney tissue fibrosis and tubular injury pathways, indicating strong biological plausibility. By enabling early and accurate prediction of disease progression in *APOL1* high-risk individuals, APRS bridges the gap between genetic susceptibility and clinical translation. This scalable and biologically informed approach provides a precision medicine framework for early intervention and may accelerate development of APOL1-targeted therapies to reduce kidney disease disparities.

Kidney failure (also known as end-stage kidney disease (ESKD)) is a life-threatening condition that requires dialysis or kidney transplantation for survival and imposes enormous global and societal costs. Worldwide[1,2], chronic kidney disease (CKD) is estimated to affect more than 800 million people, and, in the United States alone, over 800,000 people are living with kidney failure. Medicare expenditures exceeded $52 billion in 2021, with per-person costs more than twice those without kidney failure[3–5].

The burden of kidney failure is disproportionately high among African ancestry individuals, who develop kidney failure at nearly four times the rate of European ancestry individuals[6]. This disparity reflects social determinants of health, unequal access to care and genetic susceptibility. Variants in *APOL1* (refs. 7,8), discovered in 2010, are among the strongest genetic risk factors for kidney failure. An estimated 4–5 million African Americans[9,10] and tens of millions worldwide carry the high-risk genotype (two *APOL1* risk alleles, G1 and/or G2)[11,12]. Although most high-risk carriers remain disease free, an estimated one in five progresses to kidney failure—substantially higher than in individuals with zero or one risk allele—making APOL1 a critical driver of racial disparities.

Therapies targeting APOL1 biology, such as the investigational inhibitor inaxaplin[13,14], are now emerging and hold promise for

✉e-mail: ksusztak@pennmedicine.upenn.edu

preventing kidney failure in high-risk individuals. However, their use is constrained by the inability to identify which carriers are most likely to progress before CKD develops. Despite the major personal and economic burden of kidney failure, current prognostic tools remain inadequate. Clinical equations such as the Kidney Failure Risk Equation (KFRE)[15] perform well only after CKD is established[16], when much of the damage is irreversible. Genetic approaches, including polygenic risk scores (PRSs) and known *APOL1* modifiers[17], provide only modest discrimination and are not clinically actionable.

Plasma proteomics offers a potential solution. Protein levels are tightly regulated, reflect dynamic biology and can reveal subclinical injury not captured by standard measures[18–23]. We, therefore, performed broad-scale plasma proteomic profiling in *APOL1* high-risk individuals with preserved eGFR (≥60 ml min$^{-1}$ 1.73 m$^{-2}$) to develop and validate a prognostic biomarker score in two external cohorts. This approach directly addresses the gap in early risk prediction and provides a framework for precision medicine aimed at reducing disparities in APOL1-associated kidney disease.

## Results

### Baseline characteristics and outcomes

Of the 57,170 participants in the Penn Medicine BioBank (PMBB) who underwent exome sequencing, 1,310 carried the *APOL1* high-risk (G1/G1, G2/G2 or G1/G2) genotype. After excluding 109 with prior kidney transplantation and 88 with kidney failure, 1,113 participants remained eligible for analysis (Fig. 1), including 262 with eGFR <60 ml min$^{-1}$ 1.73 m$^{-2}$. As shown in Table 1, participants with eGFR ≥60 ml min$^{-1}$ 1.73 m$^{-2}$ were younger (mean age, 49.2 ± 15.1 years versus 57.7 ± 15.8 years) and more often female (66.5% versus 51.5%). The mean eGFR in this group was 90.6 ± 17.0 ml min$^{-1}$ 1.73 m$^{-2}$, and the median urine albumin–creatinine ratio (UACR) was 17.2 mg g$^{-1}$ (interquartile range (IQR), 11–31). Hypertension (62.4% versus 85.1%), diabetes (27.9% versus 37.4%) and cardiovascular disease (9.2% versus 14.5%) were more common in the eGFR <60 group, whereas the protective p.N264K allele was

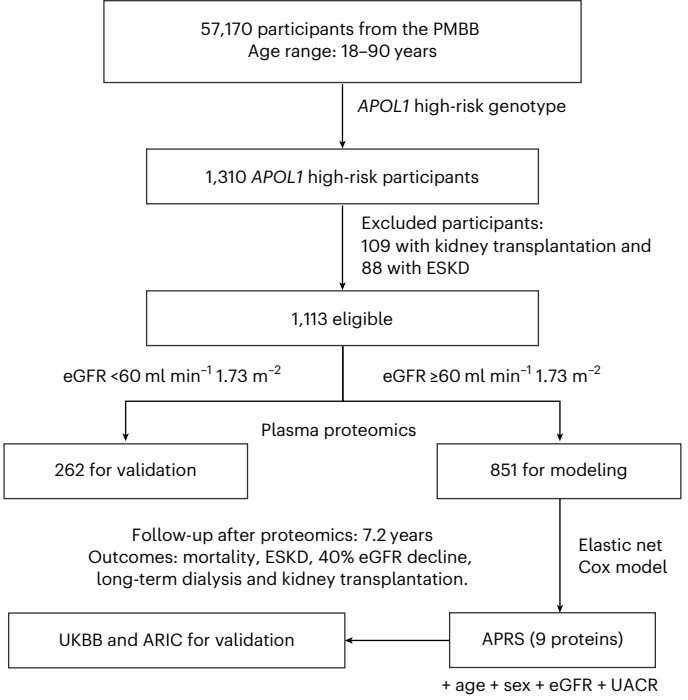

**Fig. 1 | Study design and analysis.** Data from the PMBB were used to integrate clinical, genetic and proteomic information over a 10-year follow-up period. Primary outcomes included mortality, diagnosis of ESKD, a ≥40% decline in eGFR, long-term dialysis and kidney transplantation. Proteomic profiling for ARIC validation was performed on visit 2 biospecimens.

**Table 1 | Baseline characteristics of the study participants**

| | | *APOL1* high-risk | |
|---|---|---|---|
| | Total | eGFR≥60 ml min$^{-1}$1.73 m$^{-2}$ | eGFR <60 ml min$^{-1}$1.73 m$^{-2}$ |
| Number | 1,113 | 851 | 262 |
| Age - years | 51.22±15.67 | 49.22±15.07 | 57.73±15.84 |
| Female - number (%) | 701 (62.98) | 566 (66.51) | 135 (51.53) |
| Systolic blood pressure - mmHg | 129.09±18.3 | 128.54±17.79 | 130.87±19.81 |
| Diastolic blood pressure - mmHg | 77.04±12.02 | 77.59±11.9 | 75.23±12.26 |
| Body mass index - kg m$^{-2}$ | 31.95±7.8 | 32.27±7.84 | 30.93±7.58 |
| Hemoglobin A1c (%) | 6.59±1.81 | 6.57±1.81 | 6.66±1.81 |
| Creatinine (mg dl$^{-1}$) | 1.14±0.68 | 0.88±0.21 | 1.99±0.95 |
| Blood urea nitrogen (mg dl$^{-1}$) | 16.73±10.59 | 13.04±4.98 | 28.69±14.44 |
| eGFR - ml min$^{-1}$1.73 m$^{-2}$ | 78.26±27.96 | 90.59±17.04 | 38.19±16.67 |
| UACR (IQR) - mg g$^{-1}$ | 17.23 (11–44) | 17.23 (11–31) | 31.67 (15–124) |
| 3–299 - number (%) | 163 (14.64) | 112 (13.16) | 51 (20.23) |
| ≥300 - number (%) | 60 (5.39) | 25 (2.93) | 35 (13.89) |
| Diagnostic group - number (%) | | | |
| Hypertension | 754 (67.74) | 531 (62.4) | 223 (85.11) |
| Diabetes mellitus | 335 (30.1) | 237 (27.85) | 98 (37.4) |
| Cardiovascular disease | 116 (10.42) | 78 (9.17) | 38 (14.5) |
| p.N264K | 45 (4.04) | 40 (4.7) | 5 (1.91) |
| Event - number (%) | | | |
| Composite event | 298 (26.77) | 153 (17.98) | 145 (55.34) |
| Deceased | 119 (10.69) | 62 (7.29) | 57 (21.76) |
| Kidney event | 245 (22.01) | 120 (14.1) | 125 (47.71) |
| ≥40% eGFR decline | 202 (18.15) | 110 (12.93) | 92 (35.11) |
| Kidney transplantation | 36 (3.23) | 10 (1.18) | 26 (9.92) |
| ESKD | 76 (6.83) | 26 (3.06) | 50 (19.08) |
| Dialysis | 59 (5.3) | 15 (1.76) | 44 (16.79) |
| Follow-up time - years | 7.06±3.05 | 7.18±2.89 | 6.67±3.5 |
| Time to event - years | 5.93±3.36 | 6.51±3.07 | 4.02±3.57 |

more common in the eGFR ≥60 group than in the eGFR <60 group (4.7% versus 1.9%)[17,24].

Over 10 years of follow-up, the composite outcome (≥40% eGFR decline, kidney failure or death) occurred in 18.0% of those with baseline eGFR ≥60 ml min$^{-1}$ 1.73 m$^{-2}$ and in 55.3% of those with eGFR <60 ml min$^{-1}$ 1.73 m$^{-2}$. These findings demonstrate a substantial event burden even before CKD is clinically apparent, underscoring the need for improved risk prediction.

### Proteomic profiling and biomarker selection

Baseline plasma samples were profiled with SomaScan, quantifying 7,549 proteoforms. In participants with eGFR ≥60 ml min$^{-1}$ 1.73 m$^{-2}$, 2,161 proteoforms were significantly associated with outcomes (Benjamini–Hochberg-adjusted $P < 0.01$; Extended Data Fig. 1). Because many proteins were highly correlated with each other and with baseline eGFR or UACR, we sought a panel of markers that predict progression independent of routinely measured clinical parameters. To develop and validate such a panel, the eGFR ≥60 group was randomly partitioned, with 80% of participants being used for marker selection and the remaining 20% reserved for independent testing (Supplementary Table 1). Elastic

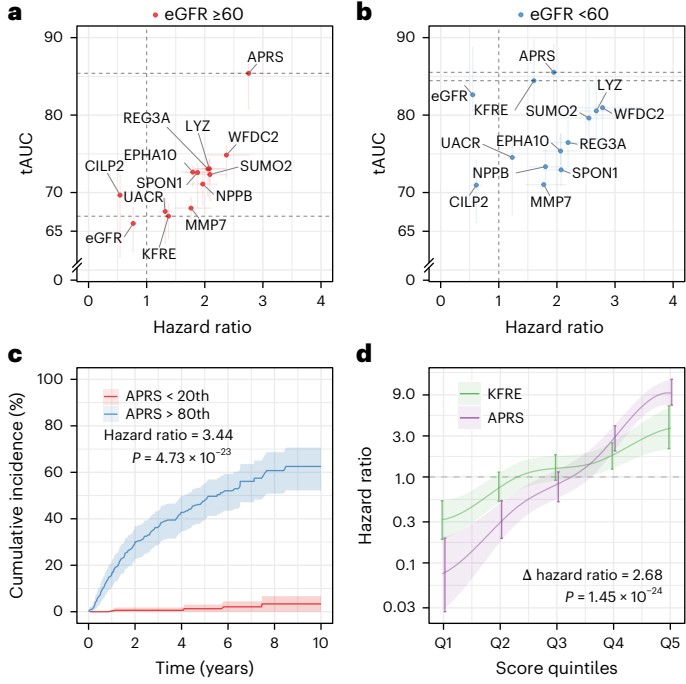

**Fig. 2 | Proteomic risk prediction in *APOL1* high-risk individuals. a**, tAUC and hazard ratios for risk prediction models and individual biomarkers among participants with eGFR ≥60 ml min$^{-1}$ 1.73 m$^{-2}$ (*n* = 680). The APRS, conventional markers including eGFR (per 5 ml min$^{-1}$ 1.73 m$^{-2}$), UACR (per doubling), the KFRE and individual biomarkers are shown for comparison. Shaded lines represent 95% confidence intervals. **b**, Similar analysis among participants with baseline eGFR <60 ml min$^{-1}$ 1.73 m$^{-2}$ (*n* = 262). **c**, Cumulative event curves over follow-up (*n* = 851). Solid lines represent cumulative incidence estimates and shaded areas represent 95% confidence intervals, stratified by APRS above the 80th percentile versus below the 20th percentile in participants with eGFR ≥60 ml min$^{-1}$ 1.73 m$^{-2}$. Corresponding hazard ratios were estimated using Cox proportional hazards models with two-sided Wald tests. **d**, Hazard ratios (solid lines) with 95% confidence intervals (shaded areas) by quintile for APRS and KFRE for composite event in participants with eGFR ≥60 ml min$^{-1}$ 1.73 m$^{-2}$ (*n* = 851). Δ hazard ratios were estimated by comparing regression coefficients within a Cox proportional hazards model using a two-sided Wald test.

net Cox regression with cross-validation reduced redundancy and identified a nine-protein signature that independently predicted risk. To identify potential substitutes, we examined correlations between the nine proteins and the broader proteomic dataset. Although some proteins were highly correlated, suitable substitutes were difficult to identify for several markers (Extended Data Fig. 2). Within the nine-protein panel itself, correlations were weak in the eGFR ≥60 group (Pearson's correlation coefficient, *R*, 0.1–0.4) and strong (0.3–0.7) when eGFR <60 ml min$^{-1}$ 1.73 m$^{-2}$ (Extended Data Fig. 3).

To understand and improve the potential biological plausibility of our biomarkers as risk predictors, we examined the associations between circulating proteins and kidney tissue pathology in an independent cohort of human kidney samples (*n* = 474 for RNA sequencing (RNA-seq) and *n* = 325 for proteomics)[23,25–27]. Because fibrosis is an independent and strong predictor of kidney failure, we focused on this pathological feature and found that four of the nine proteins were associated with interstitial fibrosis in human kidney tissue (*P* < 0.01 at both the mRNA and protein levels; Extended Data Fig. 4 and Supplementary Fig. 1). This suggests that the panel may captured biologically relevant injury pathways not reflected by conventional clinical measures.

## Development of the APRS
Having established the prognostic potential of nine individual markers that demonstrated prognostic discrimination in the eGFR ≥60 group

(tAUC >67%; Extended Data Table 1), we next sought to integrate the nine proteins. These proteins were combined with age, sex, eGFR and UACR, using the same clinical variables as in the KFRE but reestimating their coefficients in our cohort, to generate the APOL1 proteomic risk score (APRS). The APRS substantially outperformed the KFRE in participants with eGFR ≥60 ml min$^{-1}$ 1.73 m$^{-2}$ (Fig. 2a and Table 2). The KFRE was included as a comparator because it is the most widely validated clinical prediction tool for kidney outcomes, although its discrimination is known to diminish in individuals with normal eGFR. By contrast, in participants with eGFR <60 ml min$^{-1}$ 1.73 m$^{-2}$, where the KFRE performs strongly, the APRS showed similar discrimination (Fig. 2b), with much of its variance explained by eGFR ($R^2$ = 0.54; Extended Data Fig. 5, which shows *R* correlations between eGFR and APRS for any eGFR ≥60 and <60 groups). For eGFR <60 group, the $R^2$ is 0.54.

Risk stratification by quintiles revealed clear and consistent separation of outcomes: 10-year cumulative incidence ranged from 3.3% in the lowest APRS quintile to 62.5% in the highest (*P* = 4.73 × 10$^{-23}$; Fig. 2c). Across every quintile, APRS provided better discrimination than KFRE, with a significantly steeper gradient of risk (Δ hazard ratio = 2.68; *P* = 1.45 × 10$^{-24}$; Fig. 2d). Subgroup analyses showed that APRS effect size was robust across strata defined by age, sex, hypertension and diabetes (Supplementary Fig. 2). Interaction testing confirmed significant effect modification by *APOL1* genotype (interaction hazard ratio = 1.35; *P* = 5.11 × 10$^{-3}$), indicating that the prognostic strength of APRS was particularly pronounced in *APOL1* high-risk compared to low-risk individuals (Extended Data Table 2 and Supplementary Fig. 3).

## Performance of the APRS
To further assess its prognostic value, we examined the performance of the APRS in the eGFR ≥60 test cohort.

In this group, APRS achieved a tAUC of 86.5% for the composite outcome, 85.7% for mortality and 88.1% for kidney events (Fig. 3a,b, Extended Data Fig. 5 and Extended Data Table 3). At 5 years, APRS showed a sensitivity of 76.3% and a specificity of 86.7%, with concordance indexes (C-indexes) of 82.0% for the composite outcome, 84.4% for kidney outcomes and 78.5% for mortality (Fig. 3c). Decision curve analysis demonstrated greater net clinical benefit than either a treat-all or a treat-none strategy across a range of thresholds (Fig. 3d). Assuming an effective treatment (that is, inaxaplin) that reduces the risk of the composite endpoint by 27%[13,14], APRS (≥95th percentile) nearly halved the number needed to treat (NNT; 4.9 with APRS versus 8.4 with KFRE and 23.9 with CKD PRS), underscoring its potential clinical utility (Extended Data Fig. 6a). Because mortality may act as a competing risk for kidney events, we further evaluated model performance using competing-risk analyses. Discrimination remained similar when death was treated as a competing event, with tAUC values of 86.5% for the composite outcome, 87.3% for kidney events and 82.6% for mortality (Extended Data Fig. 6b). Incorporating competing risks did not materially improve predictive performance, indicating that the final model retains robust discrimination in the presence of competing events.

To benchmark APRS against existing tools (scores), we compared its performance to the KFRE, to the Chronic Renal Insufficiency Cohort (CRIC) proteomic score[28] and to CKD PRS[29]. The CRIC proteomic score is notable as it contains 65 proteins derived from patients with established CKD (eGFR <60 ml min$^{-1}$ 1.73 m$^{-2}$) yet shares only three proteins with our nine-protein APRS panel (Supplementary Table 2). In our primary target population with eGFR ≥60 ml min$^{-1}$ 1.73 m$^{-2}$, APRS achieved higher discrimination than KFRE (tAUC 86.5% versus 66.2%) and outperformed the CRIC proteomic score (79.0%). APRS maintained this level of performance in the eGFR <60 stratum (262 participants and 145 events) where CRIC and KFRE were originally developed, performing similarly to KFRE (84.2% versus 82.4%) and CRIC (83.2%). PRS performed poorly across both eGFR strata (tAUC 58.5% and 53.8%), consistent with previous observations of limited predictive value for static genetic measures in diverse populations.

**Table 2 | tAUC over 10 years for predicting composite outcomes in different cohorts using various predictive models**

| Group | Ancestry | APOL1 genotype | eGFR ml min⁻¹1.73 m⁻² | Cohort | Number | Events % | KFRE | CRIC | APRS |
|---|---|---|---|---|---|---|---|---|---|
| Training | African | High-risk | ≥60 | PMBB | 680 | 17.8 | 66.6 (1.5) | 79.7 (1.9) | 86.7 (1.4) |
| Test | African | High-risk | ≥60 | PMBB | 171 | 18.7 | 66.2 (4.3) | 79.0 (4.0) | 86.5 (3.9) |
| | | | <60 | PMBB | 262 | 55.3 | 82.4 (2.1) | 83.2 (1.7) | 84.2 (1.9) |
| | | | ≥60 | ARIC | 296 | 10.5 | - | 52.0 (1.6) | 77.5 (1.1)* |
| | | | | UKBB | 195 | 5.4 | 67.3 (6.8) | 80.0 (5.5) | 81.6 (5.9) |
| | | | Any | ARIC | 314 | 13.7 | - | 55.8 (1.9) | 82.2 (1.2) |
| | | | | UKBB | 204 | 6.3 | 75.5 (7.4) | 80.4 (5.1) | 84.7 (5.3) |
| | | Low-risk | ≥60 | ARIC | 1,932 | 13.6 | - | 52.3 (0.7) | 74.2 (0.8) |
| | | | | PMBB | 874 | 15.4 | 52.8 (3.7) | 72.6 (3.8) | 80.2 (4.1) |
| | | | | UKBB | 942 | 6.2 | 55.8 (4.4) | 75.6 (5.1) | 74.2 (5.0) |
| | | | Any | ARIC | 2,021 | 15.9 | - | 53.7 (0.5) | 78.5 (1.2) |
| | | | | PMBB | 912 | 15.1 | 60.1 (1.5) | 69.3 (1.3) | 73.1 (2.1) |
| | | | | UKBB | 967 | 6.8 | 63.8 (4.5) | 78.1 (4.6) | 74.8 (3.9) |
| | European | Low-risk | ≥60 | ARIC | 8,437 | 9.9 | - | 61.4 (0.3) | 70.2 (0.5) |
| | | | | PMBB | 658 | 13.2 | 54.6 (5.0) | 64.0 (3.9) | 63.3 (7.0) |
| | | | | UKBB | 45,623 | 8.5 | 58.2 (4.5) | 68.0 (3.3) | 71.2 (3.3) |
| | | | Any | ARIC | 8,602 | 10.5 | - | 63.6 (0.6) | 71.9 (0.9) |
| | | | | PMBB | 698 | 13.5 | 58.9 (2.3) | 66.7 (2.1) | 66.9 (2.0) |
| | | | | UKBB | 46,863 | 9.4 | 62.5 (4.1) | 70.8 (3.2) | 73.5 (3.2) |

This table presents the tAUC values (standard deviation) for different predictive models across various populations and cohorts over a 10-year period. The table compares the performance of KFRE and APRS in predicting composite outcomes. *UACR is not available for ARCI at visit 2.

## External validation of the APRS

APRS showed robust and reproducible performance across training, test and external cohorts (Table 2). In the PMBB training cohort of participants with eGFR ≥60 ml min⁻¹ 1.73 m⁻², discrimination was strong (tAUC, 86.7%) and remained similar in the independent PMBB test set, consistently outperforming KFRE and CRIC score. Among APOL1 low-risk African ancestry participants in the PMBB, APRS provided substantially better discrimination (tAUC, 73.1%) than KFRE (60.1%) and CRIC score (69.3%). Similarly, among APOL1 low-risk European ancestry participants in the PMBB, APRS outperformed KFRE (66.9% versus 58.9%) but was similar to CRIC score (66.7%).

External validation in independent population-based cohorts (Extended Data Table 4) confirmed these findings. The Atherosclerosis Risk in Communities (ARIC) study included 314 APOL1 high-risk African American participants with preserved kidney function at baseline, representing a community-dwelling population with mean age of 55.8 years and mean eGFR of 96.2 ml min⁻¹ 1.73 m⁻². Only 18 participants in this subgroup had eGFR below 60 ml min⁻¹ 1.73 m⁻². In this validation set, APRS maintained strong discrimination with a tAUC of 82.2% over 10 years, with 43 composite events observed during follow-up. When restricted to participants with preserved eGFR (≥60 ml min⁻¹ 1.73 m⁻²), discrimination was modestly attenuated (tAUC, 77.5%), likely reflecting the absence of UACR measurements in ARIC; nevertheless, APRS substantially outperformed CRIC score in this setting (52.0%). The UK Biobank (UKBB) enrolled 204 APOL1 high-risk participants of African ancestry with similar baseline characteristics (mean age 52.2 years and mean eGFR 87.1 ml min⁻¹ 1.73 m⁻², with nine participants having eGFR below 60 ml min⁻¹ 1.73 m⁻²). In this cohort, APRS showed strong discrimination with a tAUC of 84.7%. Despite differences in recruitment era, geographic location and healthcare systems across cohorts, APRS showed consistent performance beyond the academic medical center setting of PMBB. We additionally evaluated APRS, CRIC score and KFRE in participants without APOL1 high-risk genotypes across ancestries (Table 2). Although all three models retained some discriminatory ability in APOL1 low-risk populations, performance was uniformly attenuated compared to APOL1 high-risk groups, with APRS remaining similar to CRIC score and consistently outperforming KFRE.

## Discussion

We developed and validated a plasma proteomic risk score that substantially improves prediction of kidney outcomes in individuals carrying high-risk APOL1 genotypes. By integrating nine protein biomarkers (SPON1, SUMO2, EPHA10, REG3A, WFDC2, LYZ, MMP7, NPPB and CILP2) with limited clinical covariates, APRS markedly outperformed established clinical equations and genetic risk scores, especially among individuals with eGFR ≥60 ml min⁻¹ 1.73 m⁻². This addresses a critical unmet need: once CKD is clinically apparent, patients already face elevated risks of cardiovascular disease, mineral and bone disorders and premature death, and progression to kidney failure can be rapid. Although APOL1 high-risk genotypes are among the strongest genetic predictors of kidney failure, genotype information alone has not been actionable for clinical decision-making. APRS overcomes this limitation by translating static genetic risk into a dynamic, clinically usable prediction tool.

Existing risk prediction tools have important limitations. The KFRE, although widely validated and accurate in advanced CKD, performs poorly when eGFR is normal. PRSs capture inherited susceptibility but remain static and ancestry dependent. Proteomic profiling, by contrast, provides a dynamic readout of ongoing biology[18,19], integrating the cumulative influence of APOL1 risk variants and environmental exposures. In our study, more than 80% of participants had preserved kidney function without albuminuria—a group rarely included in prior risk prediction studies—underscoring both the novelty and the clinical relevance of this approach[30–33]. The APRS was developed and validated to address the critical unmet need in APOL1 high-risk individuals, achieving a tAUC of 86.5% in this group with preserved eGFR. As demonstrated in our full cross-population testing, this performance is substantially superior to the predictive value retained in APOL1 low-risk carriers of African ancestry (tAUC up to 80.2%) and European ancestry (tAUC up to 66.9%). APRS also performed similarly to KFRE in established CKD, suggesting that the biomarker panel captures both

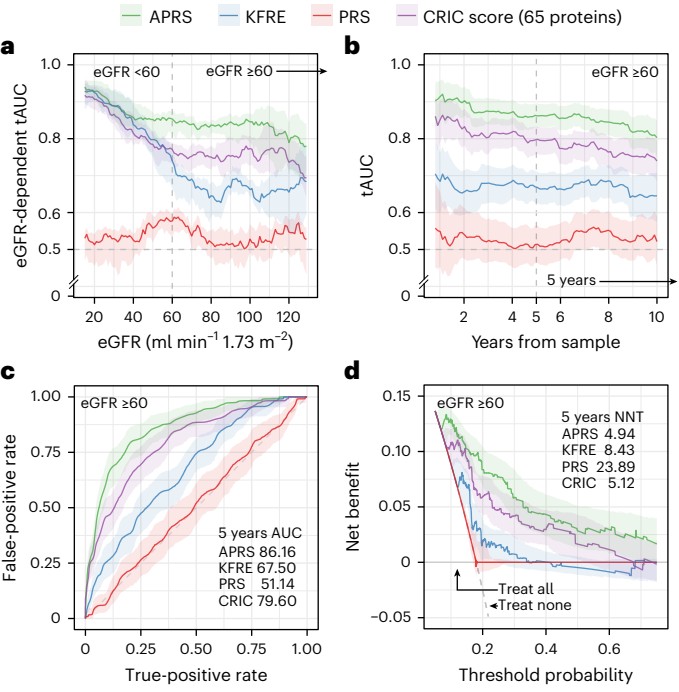

**Fig. 3 | Comparative discrimination and clinical utility of four risk scores in *APOL1* high-risk individuals with eGFR above 60 ml min⁻¹ 1.73 m⁻².** **a**, tAUC plotted as a function of baseline eGFR (ml min⁻¹ 1.73 m⁻²), illustrating how discriminative accuracy varies across levels of kidney function ($n = 433$). **b**, tAUC plotted over follow-up time (years), showing how discrimination for each model evolves longitudinally ($n = 171$). **c**, AUC for 5-year event prediction for each risk score ($n = 171$). **d**, Decision curve analysis over 5 years across clinical decision thresholds, with reference lines for 'treat all' and 'treat none'; NNTs were calculated among individuals above the 95th percentile of each score distribution ($n = 171$). In **a** and **b**, solid lines represent median tAUC estimates, whereas, in **c** and **d**, lines represent bootstrap mean estimates; shaded ribbons indicate 95% confidence intervals in all panels.

shared pathways of CKD progression and *APOL1*-specific mechanisms. These findings indicate that aptamer-based proteomics is an effective framework for risk prediction; although clinical implementation is most feasible and cost-effective in the enriched *APOL1* high-risk group, the approach could also be extended to develop analogous models in other populations.

Proteins incorporated in APRS are associated with pathways plausibly involved in kidney injury that may not be captured by conventional measures in individuals with preserved function. In the preserved GFR group, APRS showed weak correlations with eGFR, and kidney tissue expression of these proteins was associated with fibrosis severity, suggesting that the signature may relate to early tissue damage before functional decline becomes apparent. Even individuals with preserved eGFR and low albuminuria remain at risk of progression, indicating that reliance on UACR alone could miss early disease signals. Specific proteins point to biologically plausible hypotheses for future testing: MMP7 has been reported as a marker of tubular injury and fibrosis[23]; WFDC2 correlates with interstitial fibrosis and rapid decline[34,35]; and LYZ is associated with fibroblast proliferation and tubular cell senescence[36]. Together, these findings generate hypotheses that APRS may reflect tubular stress, immune activation and extracellular matrix remodeling, which are implicated in APOL1-mediated nephropathy[37–40]. However, these associations are correlative; whether these proteins actively drive disease progression or reflect altered renal clearance due to reduced nephron number requires experimental validation. Thus, although APRS predicts risk, its biological interpretation remains hypothesis generating rather than mechanistically proven.

The clinical implications of APRS are substantial. First, APRS may enable earlier identification of high-risk individuals for intensified surveillance long before CKD is detected by standard measures. Second, it could guide use of emerging APOL1-targeted therapies such as inaxaplin, by identifying those most likely to benefit and by serving as a pharmacodynamic readout of treatment effect[13,14]. APRS nearly halved NNT, making it a more efficient tool for targeting interventions. Third, serial measurements may permit longitudinal monitoring of risk trajectories in clinical practice. Finally, APRS could transform trial design by enriching enrollment with high-risk individuals, thereby increasing event rates, reducing sample size and accelerating therapeutic development. Conversely, low APRS values could provide reassurance for carriers unlikely to progress, potentially avoiding unnecessary interventions. Notably, given the disproportionate burden of CKD and kidney failure among individuals of African ancestry, early risk stratification with APRS offers a precision medicine strategy to help narrow longstanding health disparities. In this context, the reliance of APRS on aptamer-based proteomic profiling may further support its translational potential, as this technology offers advantages in simplicity, scalability and cost-effectiveness compared to traditional antibody-based protein assays such as ELISA. Nevertheless, low APRS values should not be interpreted as a substitute for standard clinical monitoring, and APRS should be considered an adjunct to established risk assessment approaches. Furthermore, prospective decision impact studies will be required to determine whether APRS-guided strategies meaningfully improve clinical outcomes before routine clinical implementation.

Several limitations should be acknowledged. Because this was an observational study, residual confounding cannot be excluded, and limitations inherent to electronic health record (EHR) codes, including potential misclassification, incomplete capture of clinical events and variable coding practices across sites, may have affected outcome ascertainment. External validation in ARIC and UKBB confirmed robust performance, but event counts in these cohorts were modest, leading to wide confidence intervals. Cross-platform differences (SomaScan versus Olink) required harmonization, underscoring the need for assay standardization. Finally, although aptamer technology is scalable and cost-effective relative to antibody-based assays, technical complexity may limit near-term clinical deployment. Ongoing improvements in proteomic platforms and decreasing costs are likely to reduce these barriers. Prospective trials will be required to establish how best to integrate the APRS into everyday clinical care.

The APRS offers a valuable early risk stratification framework for individuals with high-risk *APOL1* genotypes. With targeted therapies such as inaxaplin already advancing through clinical trials, the roadmap to translate this finding into clinical utility involves several complementary steps. Independent validation in prospective cohorts represents an important next phase to confirm efficacy in real-world settings. Concurrently, the APRS could serve as a subject enrichment tool for these therapeutic trials, potentially improving efficiency and reducing costs. Finally, adapting the current proteomic methodology into a simplified, high-throughput clinical assay would facilitate broader accessibility for routine risk assessment. In conclusion, a plasma proteomic risk score enables accurate and early prediction of adverse events in *APOL1* high-risk individuals, before CKD is clinically evident. By transforming *APOL1* genetic risk into a clinically actionable prediction tool, the APRS provides a precision medicine framework to support early intervention and reduce longstanding racial disparities in kidney failure.

## Online content

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

## Methods

### PMBB cohort

The PMBB is a large academic biobank that recruits participants from the University of Pennsylvania Health System, with recruitment beginning in 2008. To date, the PMBB has enrolled over 250,000 participants—approximately 30% of whom are from non-European ancestries—and around 57,170 individuals have undergone whole-exome sequencing[41]. Demographic information, medical history (via International Classification of Disease (ICD) codes), medication use and clinical assessments were extracted from the EHR. Laboratory tests, including serum creatinine, blood urea nitrogen and electrolyte levels at baseline, were also obtained. The eGFR was calculated using the creatinine-based 2021 Chronic Kidney Disease Epidemiology Collaboration (CKD-EPI) equation[42].

All participants provided written informed consent for the use of their biospecimens, genetic data and EHR data for research. Genomic DNA samples were transferred to the Regeneron Genetics Center and stored at −80 °C until sample preparation. Whole-exome sequencing was performed with reads mapped to Genome Reference Consortium Build 38 (GRCh38); samples failing quality metrics (for example, low sequencing coverage) were excluded, as described previously. Ancestry was estimated by exome data using principal component analysis. A set of high-quality, common single-nucleotide polymorphisms overlapping with HapMap3 was extracted. Principal components were first calculated for HapMap3 samples and then used as the reference space, onto which all study samples were projected. To classify ancestry, a kernel density estimation approach was applied to the joint distributions of the first four principal components for each HapMap3 population. For each sample, normalized likelihoods of belonging to each reference population were obtained. Reference populations were then assigned if likelihoods exceeded prespecified thresholds. Based on these assignments, samples were grouped into one of the following ancestral classes: African, European, East Asian, South Asian or Admixed American. Subsequently, the *APOL1* was analyzed in all African ancestry participants with available exome sequence data[41].

Individuals were classified into high-risk *APOL1* genotype groups based on the presence of the G1 (G1a and G1b) and G2 risk alleles. Participants with two risk alleles (G1/G1, G2/G2 or G1/G2) were classified as high-risk[11,12], whereas those with zero or one risk allele (G0/G0, G0/G1 or G0/G2) were classified as low-risk. We included all participants aged ≥18 years with available high-risk *APOL1* genotype and at least one subsequent follow-up record or documented clinical event (for example, mortality and dialysis). Exclusion criteria included known ESKD or with kidney transplant at baseline, missing or ambiguous *APOL1* data and excessive missing data in key clinical covariates. To minimize reverse causation, clinical diagnoses (for example, hypertension, diabetes and cardiovascular disease) were required to have been recorded in the EHR prior to the blood draw. Laboratory values used as baseline covariates were extracted from tests performed within a prespecified window of 2 months before to 1 month after the plasma draw; for each participant, we used the single measurement closest to the draw date to reduce temporal misalignment. For UACR, we used the most recent measurement obtained within the 2 years preceding the plasma draw. When only a urine protein–creatinine ratio or a dipstick protein result was available, we converted these to estimated UACR using the conversion equations from ref. 43. This work was conducted under University of Pennsylvania Institutional Review Board (IRB)-approved protocols (815796, 813913, 855821 and 857403; study protocol).

### Outcomes and follow-up

The primary outcomes were defined as a composite of kidney events (long-term dialysis, ESKD diagnosis, kidney transplantation or a ≥40% decline in eGFR from baseline) and all-cause mortality. Mortality was included given its clinical importance and as a critical competing risk for kidney failure[44]. For *APOL1* high-risk individuals with eGFR ≥60 ml min$^{-1}$ 1.73 m$^{-2}$, the mean follow-up duration was 7.18 ± 2.89 years, with a mean time to event of 6.51 ± 3.07 years. Outcome data were obtained through medical record review and ICD-10 codes. Participants were censored at the time of death, loss to follow-up or the end of a 10-year follow-up period, whichever occurred first.

### Proteomic profiling

PMBB plasma samples were collected at baseline and subjected to proteomic profiling using the SomaScan platform (SomaLogic), which employs modified aptamers (SOMAmers) to probe the human proteome[22]. SomaScan v.4.1 pools 7,524 SOMAmers to probe 6,386 distinct proteins (https://menu.somalogic.com/). The SomaScan assay is a highly multiplexed, sensitive and reproducible proteomic technology that has been extensively validated and used in numerous clinical studies[23,30–33,45–48]. The SomaScan assay is based on the use of modified DNA aptamers, called SOMAmers, which are designed to bind specific protein targets with high affinity and specificity[22]. Each SOMAmer is uniquely tagged with a DNA barcode that allows for quantification using a custom DNA microarray. In brief, the diluted plasma samples were incubated with a pool of SOMAmers, and, subsequently, the SOMAmer–protein complexes were captured on streptavidin-coated beads, washing away unbound proteins and SOMAmers. The bound SOMAmers were then eluted and hybridized to a custom DNA microarray containing complementary sequences to the SOMAmer barcodes. The microarrays were scanned using a SureScan Dx Microarray Scanner (Agilent Technologies), and the fluorescence intensity of each SOMAmer was quantified as a measure of the relative abundance of its corresponding protein target.

Raw data were processed using SomaScan Data Analysis Software (SomaLogic) to generate relative fluorescence units (RFUs) for each SOMAmer. The RFUs were then normalized using a set of internal calibrator samples to adjust for any assay-specific biases and to ensure comparability across different plates and runs. Rigorous quality control measures were implemented throughout the proteomic profiling process to ensure data integrity and reliability. These included the use of internal calibrator samples, quality control metrics for sample and assay performance and the exclusion of any samples or SOMAmers that failed to meet predefined quality criteria[49]. The log$_2$-transformed RFUs were then used for subsequent statistical analyses and model development.

### UKBB cohort

The UKBB is a large-scale, community-based cohort comprising over 500,000 participants aged 40–69 years at recruitment (2006–2010) from 22 assessment centers across the United Kingdom. For validation purposes, we included all participants who underwent plasma proteomic profiling[50]. This subset is representative of the overall UKBB population. Among these, 1,171 individuals of African descent were identified—204 with *APOL1* high-risk. *APOL1* genotypes were imputed (using Data-Field 21007) based on the TOPMed R2 reference panel after phasing with Eagle v.2.4 and converting from GRCh37 to GRCh38 via LiftOver[51]. ESKD was identified using data_coding_19 codes N185 and N180 in fields 41270 and 41280 and using READV3_CODE mapped to ICD-10 codes N185 and N180 in Table 1060; hypertension was captured using data_coding_19 with the regular expression 'I1[012345]' in fields 41270 and 41280 and using READV3_CODE mapped to ICD-10 with the same expression in Table 1060; diabetes was defined using data_coding_19 with the regular expression 'E1[01234]' in fields 41270 and 41280 and using READV3_CODE mapped to ICD-10 with the same expression in Table 1060; kidney transplant history was ascertained using data_coding_19 code Z940 in fields 41270 and 41280 and using READV3_CODE mapped to ICD-10 code Z940 in Table 1060; dialysis treatment was identified through procedure_concept entries matching '[Dd]ialysis' in Table 936 and TERMV3_DESC entries matching '[Dd]ialysis' in Table 1060; mortality outcomes were obtained from death register

data in Table 1058; serum creatinine measurements were drawn from measurement_concept_id 37392176 and 3020564 in Table 931, from repeat-measure identifiers p30700_i.* and p23478_i.* in field 30700 and from TERMV3_DESC entries matching '[Ss]erum [Cc]reatinine$' in Table 1060; and cystatin C was obtained via measurement_concept_id 3030366 in Table 931. Olink measurements were $\log_2$ transformed and normalized to harmonize scales. Missing proteins in Olink were imputed by mapping overlapping proteins to the SomaScan scale and predicting absent targets with a multi-output penalized regression trained on SomaScan data (statistical analysis protocol). This study was conducted under application number 273810. The validation from the UKBB for this project was approved by the University of Pennsylvania IRB (protocol 855821).

## ARIC study
The ARIC study enrolled 15,792 participants (aged 45–65 years) from four US communities (Washington County, Maryland; Forsyth County, North Carolina; Jackson, Mississippi; and Minneapolis, Minnesota) between 1987 and 1989, with follow-up visits every 3 years initially and subsequently at varying intervals[52]. We included 314 African American participants with *APOL1* high-risk genotyping (performed using TaqMan assays for G1 and G2)[8,53] who attended visit 2 (approximately 3 years after baseline) with eGFR≥60 ml min$^{-1}$ 1.73 m$^{-2}$; proteomic profiling on these visit 2 biospecimens was performed using the SomaScan 5K assay. To ensure comparability with the PMBB, we truncated follow-up at 10 years from baseline. This study was conducted under application number MP4524. The validation from the ARIC for this project was approved by the University of Pennsylvania IRB (protocol 855821). All validation analyses were independently performed by statisticians from different institutions.

## Human kidney samples
Kidney tissue samples (n = 474 for RNA-seq and n = 325 for proteomics) for this study were procured from surgical nephrectomies, ensuring that only the normal parts of the tissue, specifically those at least 2 cm from any cancerous lesions, were used for analysis. An honest broker deidentified the samples and collected corresponding clinical information, such as age, race, sex and diabetes and hypertension status, in addition to creatinine values. The eGFR was subsequently determined using the latest CKD-EPI equations[42]. Kidney samples were formalin fixed, paraffin embedded and stained with periodic acid–Schiff. Whole-tissue imaging was conducted using the Aperio system, which is a platform that digitizes and assists in analyzing pathology slides. Samples were scored in an unbiased manner by a specialized renal pathologist[23,25,26]. The use of these samples and data was approved by the University of Pennsylvania IRB under the category of 'exempt', negating the need for informed consent due to the deidentified nature of the study samples.

## Kidney tissue RNA-seq and data processing
RNA isolation, sequencing and analysis were performed as previously published[27]. Total RNA was isolated from kidney tissue using the RNeasy Mini Kit (Qiagen) according to the manufacturer's instructions, including the DNase digestion step. RNA quality was assessed by Agilent Bioanalyzer 2100. The cDNA library was prepared using NEBNext Ultra II RNA Library Prep Kit for Illumina. Then, cDNA libraries were sequenced on an Illumina NovaSeq 6000 platform using the NovaSeq PE150 protocol. Adaptor and lower-quality bases were trimmed with TrimGalore (v.0.4.5). Reads were aligned to the human genome (hg19) using STAR (v.2.7.3a). Gene and isoform expression levels of transcripts per million were estimated using RSEM (v.1.3.0).

## Kidney tissue proteomics and data processing
Kidney tissue samples were snap frozen, cryopulverized (CryoMill; Retsch) and lysed using T-PER extraction reagent supplemented with protease inhibitors (Roche Diagnostics). Protein concentrations were determined via bicinchoninic acid assay (Thermo Fisher Scientific). Similar to plasma proteomics, the tissue proteomic profiling was performed using the SomaScan v.4.1 platform (SomaLogic), which uses slow off-rate modified DNA aptamers (SOMAmers) to quantify protein targets. To ensure data consistency, quality control measures included hybridization controls, pooled calibrators and buffer-only replicates for monitoring background signals and batch effects. Data underwent normalization to correct for within-run hybridization variability, followed by intrastudy median normalization.

## PRS
A previously published genome-wide PRS for CKD[29] was applied. For each individual, the PRS was calculated as the weighted sum of risk alleles, with weights corresponding to the effect sizes reported in the original study. In brief, the PRS was computed as PRS = $\sum_{i=1}^{M} \beta_i \times \text{dosage}_i$, where $M$ is the number of variants with non-zero weights and $\beta_i$ is the effect size for variant $i$. To enable genome-wide PRS computation, we used imputed genotype data from the PMBB (Release 2.0), which includes participants genotyped using the Illumina Global Screening Array (Freeze 2.0). Genotypes were phased with Eagle2 and imputed using Minimac4 on the Michigan TOPMed r3 reference panel (GRCh38). The imputed dataset was filtered to retain high-confidence variants (average $R^2 > 0.3$ or directly genotyped in either batch) and minor allele frequency > 0.01, ensuring compatibility with published PRS weights. PRS values were computed in PLINK 2.0 using the overlapping variant set between the published PRS and the PMBB-imputed genotypes. The PRS analysis was included primarily as a benchmark to contextualize the incremental predictive value of the APRS.

## CRIC proteomics model
For benchmarking, we applied the previously published CRIC proteomic risk score, which was developed using SomaScan profiling of 65 plasma proteins in patients with established CKD[28]. The original coefficients were applied to our SomaScan data after $\log_2$ transformation and harmonization of units. For each participant, a CRIC score was calculated as the weighted sum of these proteins. The performance of the CRIC score was evaluated using tAUC and C-index for the composite outcome in the PMBB.

## Marker selection and model construction
To identify the optimal combination of proteins for outcome prediction, we implemented an elastic net penalized Cox proportional hazards modeling framework based on the random generation of candidate protein panels. This strategy is conceptually related to random subspace approaches[54] and ensemble feature selection methods[55,56] but differs in that resampling is performed on the feature space rather than on individuals, and each candidate protein panel is evaluated independently. This was implemented through the following sequential steps: (1) initial protein screening and candidate panel generation by identifying proteins showing significant univariate associations (Benjamini–Hochberg-adjusted $P < 0.05$) with the outcome (top 20%) and then randomly sampling 40–90% of these proteins without replacement for each candidate panel; (2) elastic net modeling with grid search, fitting an elastic net Cox model for each sampled protein panel and conducting a comprehensive grid search across the mixing parameter $\alpha$ (ranging from 0.1 to 0.9 in 0.1 increments) and the regularization strength $\lambda$ (spanning 100 logarithmically spaced values); (3) performance evaluation via eight-fold cross-validation using the mean C-index and large-scale combinatorial exploration of approximately one million unique candidate protein panels to thoroughly explore the vast combinatorial space of protein interactions and identify the optimal combination; (4) intermediate feature selection by stability, retaining proteins with a selection frequency of at least 30% across candidate panels to avoid multiple testing problems and reduce noise, ensuring

that only consistently predictive proteins were retained; and (5) final model construction and refinement using the intermediate protein set with another round of elastic net penalized Cox proportional hazards modeling to eliminate highly correlated proteins and optimize model parameters based on the most promising candidates identified in the initial screening and evaluation process. The final proteomic signature was defined as the panel of proteins and corresponding penalty parameters that achieved the highest average cross-validated C-index across all iterations.

The model risk score is: score $= \exp\left(\sum_j \beta_j \times x_j(t)\right)$, where $(\beta_j)$ are the coefficients and $(x_j(t))$ are the protein markers. The APRS formula is:

$$\hat{y}(t|\mathbf{x}) = \exp(1.72 \times \text{SPON1} + 1.18 \times \text{SUMO2} + 1.08 \times \text{EPHA10}$$

$$+0.89 \times \text{REG3A} + 1.37 \times \text{WFDC2} + 1.56 \times \text{LYZ} + 0.68 \times \text{MMP7}$$

$$+0.55 \times \text{NPPB} - 0.95 \times \text{CILP2} + 0.22 \times \text{UACR}_{\text{per doubling}}$$

$$+0.19 \times \text{Age}_{\text{per 10 years}} - 0.27 \times \text{eGFR}_{\text{per 5 mL min}^{-1}\,1.73\,\text{m}^{-2}} + 0.49 \times \text{Male})$$

## Model performance evaluation

We evaluated the performance of the predictive model using a comprehensive set of metrics designed for survival analysis. The evaluation function takes as input the true survival outcomes from the training and test sets, along with the predicted risk scores and the timepoints at which the predictions are made. The function first prepares the data by aligning the predicted risk scores with the true survival outcomes and applying inverse probability of censoring weights (IPCW) to account for censoring in the test set. The IPCW for each sample $i$ at time $t$ is calculated as: $\text{IPCW}_{i,t} = \frac{1}{\hat{S}(t|\mathbf{X}_i)}$ where $\hat{S}(t | \mathbf{X}_i)$ is the estimated probability of being uncensored at time $t$ given the covariates $\mathbf{X}_i$. The samples are then sorted by descending risk score at each timepoint. Let $n$ be the number of samples and $m$ be the number of timepoints. For each timepoint $t$, we define: $\text{TP}_t$: true positives, $\text{FP}_t$: false positives, $\text{TN}_t$: true negatives, $\text{FN}_t$: false negatives. Next, the function calculates various performance metrics at each timepoint, including: AUC: $\text{AUC}_t = \int_0^1 \text{TPR}_t(\text{FPR}_t)\,\text{dFPR}_t$ where $\text{TPR}_t$ is the true-positive rate (sensitivity) and $\text{FPR}_t$ is the false-positive rate (1 – specificity) at time $t$. Model sensitivity, specificity F1 score, accuracy, Matthews correlation coefficient (MCC), positive predictive value (PPV) and negative predictive value (NPV) were calculated as: $\text{Sensitivity}_t = \frac{\text{TP}_t}{\text{TP}_t+\text{FN}_t}$, $\text{Specificity}_t = \frac{\text{TN}_t}{\text{TN}_t+\text{FP}_t}$, $\text{F1}_t = \frac{2\times\text{Precision}_t\times\text{Recall}_t}{\text{Precision}_t+\text{Recall}_t}$ where $\text{Precision}_t = \frac{\text{TP}_t}{\text{TP}_t+\text{FP}_t}$, and $\text{Recall}_t = \frac{\text{TP}_t}{\text{TP}_t+\text{FN}_t}$, $\text{Accuracy}_t = \frac{\text{TP}_t+\text{TN}_t}{\text{TP}_t+\text{FP}_t+\text{TN}_t+\text{FN}_t}$, $\text{MCC}_t = \frac{\text{TP}_t\times\text{TN}_t-\text{FP}_t\times\text{FN}_t}{\sqrt{(\text{TP}_t+\text{FP}_t)(\text{TP}_t+\text{FN}_t)(\text{TN}_t+\text{FP}_t)(\text{TN}_t+\text{FN}_t)}}$, $\text{PPV}_t = \frac{\text{TP}_t}{\text{TP}_t+\text{FP}_t}$ and $\text{NPV}_t = \frac{\text{TN}_t}{\text{TN}_t+\text{FN}_t}$. These metrics are computed by considering the predicted risk scores as a binary classifier at each timepoint, with the threshold determined by the point that maximizes the sum of sensitivity and specificity. Finally, if multiple timepoints are evaluated, the function computes the mean of each performance metric weighted by the probability of survival at each timepoint:

$$\bar{M} = \frac{\sum_{t=1}^m M_t \times \hat{S}(t)}{\sum_{t=1}^m \hat{S}(t)}$$

where $M_t$ is the performance metric at time $t$ and $\hat{S}(t)$ is the estimated probability of survival at time $t$. The evaluation function returns a dictionary containing the performance metrics at each timepoint and, if applicable, the weighted mean of each metric across all timepoints. All performance metrics are reported on a 0–100% scale for simplicity.

## Statistical analysis

Continuous variables are presented as median ± s.d. or IQR, and categorical variables are presented as frequencies and percentages. The independent $t$-test was used for normally distributed continuous variables, the Mann−Whitney $U$-test for non-normally distributed continuous variables and Pearson's $\chi^2$ test for categorical variables. Missing data were handled across the entire PMBB dataset. Variables with more than 15% missingness were excluded. Remaining missing values were imputed using multiple imputation by chained equations; imputation was performed separately within the training and test sets to avoid information leakage. Survival analyses were conducted using the Kaplan−Meier method, and differences in survival probabilities among the study groups were assessed using the log-rank test. The proportional hazards assumption was assessed using covariate-specific and global tests based on Schoenfeld residuals (Grambsch−Therneau), together with graphical evaluation of log−minus−log survival plots and plots of scaled Schoenfeld residuals versus time. Where evidence of non-proportionality was observed, time-dependent effects were modeled by introducing interactions between the covariate and functions of time (for example, log(time)) or by fitting stratified Cox models. The predictive performance of the developed models was evaluated using a comprehensive set of metrics, including AUC, C-index, specificity, sensitivity, F1 score, precision, recall, accuracy, FPR, FNR, MCC, PPV and NPV. To systematically identify proteins correlated with the nine APRS proteins, we computed multiple similarity metrics between each APRS protein vector and all other proteins measured by SomaScan. In the radar plot, proteins from APRS were ordered using a greedy traversal of the core−core correlation matrix, iteratively selecting the most strongly correlated unvisited protein. Non-APRS protein angles were determined by a weighted circular mean of APRS protein angles, with weights derived from their absolute correlations to APRS proteins.

The *APOL1* high-risk with eGFR ≥60 ml min$^{-1}$ 1.73 m$^{-2}$ group was divided into an 80% training set and a 20% test set using stratified sampling. In the training set, an elastic net Cox model was applied to select protein markers. Model performance was evaluated using eight-fold cross-validation to minimize overfitting, and predictive accuracy was assessed with AUC and tAUC at various timepoints[57]. The final model included nine proteins plus age, sex, baseline eGFR and log$_2$(UACR) and was fitted using an elastic net Cox model. Sample size was estimated using the Schoenfeld method for the Cox proportional hazards model and an events per variable (EPV) approach (with EPV set at 15). The EPV method yielded the required sample size of 640 participants. By contrast, based on a two-sided significance level of 0.05, an expected 30% marker-positive rate, an anticipated event rate of 25% and a target hazard ratio of 1.648−assuming median dichotomization of the model score−a total of 685 participants was determined to provide an effective statistical power of approximately 85%. Net benefits from decision curve analysis were calculated by weighting false positives and false negatives under the assumption of equal misclassification costs[58]. Feature stability was checked by bootstrap enumeration. NNT was calculated as the reciprocal of the absolute risk reduction, where absolute risk reduction = control event rate × relative risk reduction (RRR = 0.27, reported for inaxaplin)[13]. Two-tailed $P$ values less than 0.05 were considered statistically significant. All analyses were performed using Python 3.9.13 and R 4.3.2.

## Reporting summary

Further information on research design is available in the Nature Portfolio Reporting Summary linked to this article.

## Data availability

The datasets analyzed in this study are not publicly available owing to participant privacy and data use agreements but may be accessed through application to the PMBB (https://pmbb.med.upenn.edu/). Access requires approval by the PMBB data access committee and execution of the PMBB data use agreement. Requests for proteomic data or verification analyses may be directed to the corresponding author (ksusztak@pennmedicine.upenn.edu). An initial response is generally provided within approximately 2 weeks. ARIC data may be

requested from the ARIC Data Coordinating Center by obtaining study approval, executing a data and materials distribution agreement and submitting a data request form to aricdata@unc.edu. Alternatively, ARIC data are available through the NHLBI BioLINCC (https://biolincc.nhlbi.nih.gov/) repository and the dbGaP (https://dbgap.ncbi.nlm.nih.gov/beta/study/phs000280.v9.p3) subject to their application procedures. Review timelines typically require approximately 4–8 weeks. For datasets derived from the UKBB, data access is governed by the UKBB's established policies. Researchers must apply through the UKBB Access Management System, available at https://www.ukbiobank.ac.uk/, outlining the purpose of the intended use. Applications are reviewed by the UKBB, and decisions are generally provided within approximately 4–6 weeks. Source data are provided with this paper.

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

## Acknowledgements

This work was supported by National Institutes of Health (NIH), National Institute of Diabetes and Digestive and Kidney Diseases (NIDDK) grants R01 DK076077, R01 DK087635 and R01 DK105821 (to K.S.). The ARIC study has been funded in whole or in part with federal funds from the National Heart, Lung, and Blood Institute (NHLBI), NIH, Department of Health and Human Services, under contract numbers 75N92022D00001, 75N92022D00002, 75N92022D00003, 75N92022D00004 and 75N92022D00005. SomaLogic conducted SomaScan assays in exchange for use of ARIC data, which was supported in part by NIH/NHLBI grants R01 HL134320 and R01 DK124399. T.V. was a summer research student from Riverdale Country School at the time of this work; her listed affiliation reflects where the research was conducted. We acknowledge the PMBB for providing data and thank the patients of Penn Medicine who consented to participate in this research program. We would also like to thank the PMBB team and Regeneron Genetics Center for providing genetic variant data for analysis. The PMBB is approved under IRB protocol 813913 and is supported by the Perelman School of Medicine at the University of Pennsylvania, by a gift from the Smilow family and by the National Center for Advancing Translational Sciences of the NIH under Clinical and Translational Science Awards number UL1TR001878. PMBB members include D. J. Rader, M. D. Ritchie, J. Weaver, N. Naseer, G. Sirugo, A. Poindexter, J. Dever, A. Harvey, S. Linn, N. Srivastava, M. Livingstone, F. Vadivieso, S. DerOhannessian, T. Tran, J. Stephanowski, S. Santos, N. Haubein, J. Dunn, A. Verma, C. Morse Kripke, M. Risman, R. Judy, C. Wollac, S. S. Verma, S. Damrauer, Y. Bradford, S. Dudek, T. Drivas and Z. Rodriguez. The authors thank the staff and participants of the ARIC study for their important contributions. This work was conducted in collaboration with Novartis and the Susztak laboratory.

## Author contributions

Concept and design: K.S., L.L.J., W.F.D., N.F., J.J.L., S.M.R., A.A., G.Q. and C.L. Paper drafting: C.L. and K.S. Statistical analysis: C.L., S.M.R., S.M., V.B., H.L. and K.S. Data collection and interpretation: S.M.R., N.F., R.P., X.Q., T.S., M.Z., T.V., L.L.J., A.V., M.R., D.J.R., C.L. and K.S. External validation: A.S., J.C., M.E.G. and A.K. Paper revision: C.L., S.M.R., V.B., R.P., X.Q., T.S., N.F., J.J.L. and K.S. All authors critically reviewed and agreed to the submission of the final paper.

## Competing interests

The laboratory of K.S., including C.L., G.Q., A.A., M.Z., T.V. and S.M., receives research support from Gilead, Novo Nordisk, Novartis, GlaxoSmithKline, Boehringer Ingelheim, Regeneron, Genentech and Calico Life Sciences. J.C. is a scientific advisor receiving fees from SomaLogic and Healthy.io. S.M.R., R.P., V.B., X.Q., T.S., J.J.L., N.F., W.F.D. and L.L.J. are employees and stockholders of Novartis. A.K. is an employee of Novo Nordisk. The remaining authors declare no competing interests.

## Additional information

**Extended data** is available for this paper at https://doi.org/10.1038/s41591-026-04337-2.

**Correspondence and requests for materials** should be addressed to Katalin Susztak.

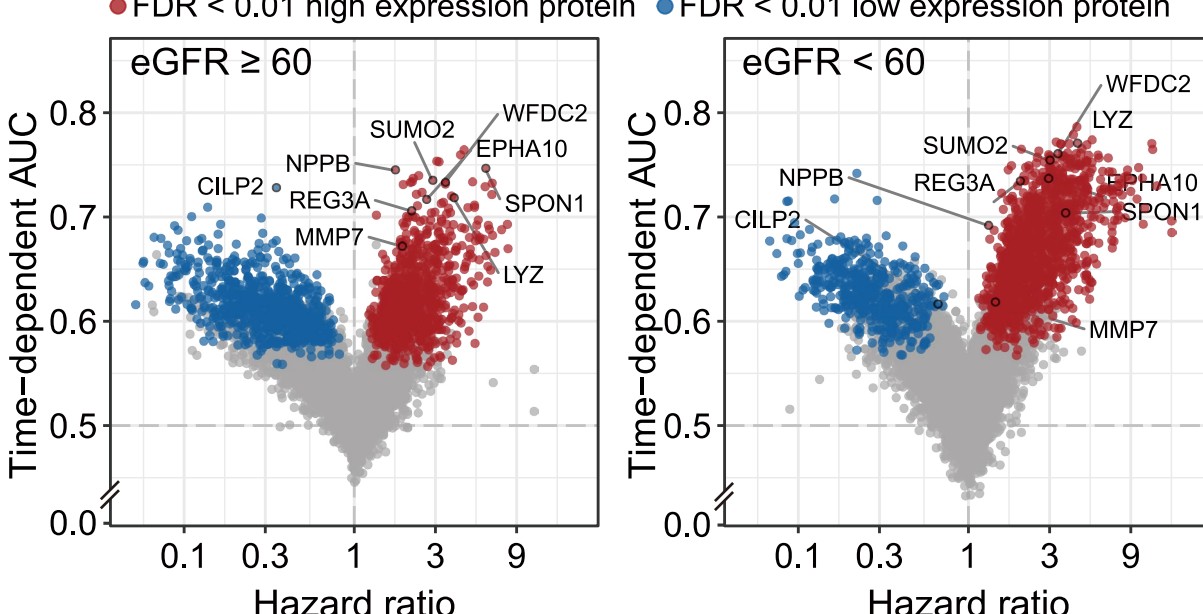

**Extended Data Fig. 1 | Hazard Ratios and Time-dependent AUC Values for proteomics for composite event.** Volcano plots showing hazard ratios (x-axis) and AUC values (y-axis) of protein biomarkers predicting kidney events and mortality *APOL1* high-risk individuals with eGFR <60 (n = 680) and ≥60 mL/min/1.73 m² (n = 262). Hazard ratios were estimated using univariable

Cox proportional hazards models with two-sided Wald tests. False discovery rate (FDR) was calculated using the Benjamini–Hochberg method. eGFR, estimated glomerular filtration rate; AUC, area under the receiver-operating-characteristic curve.

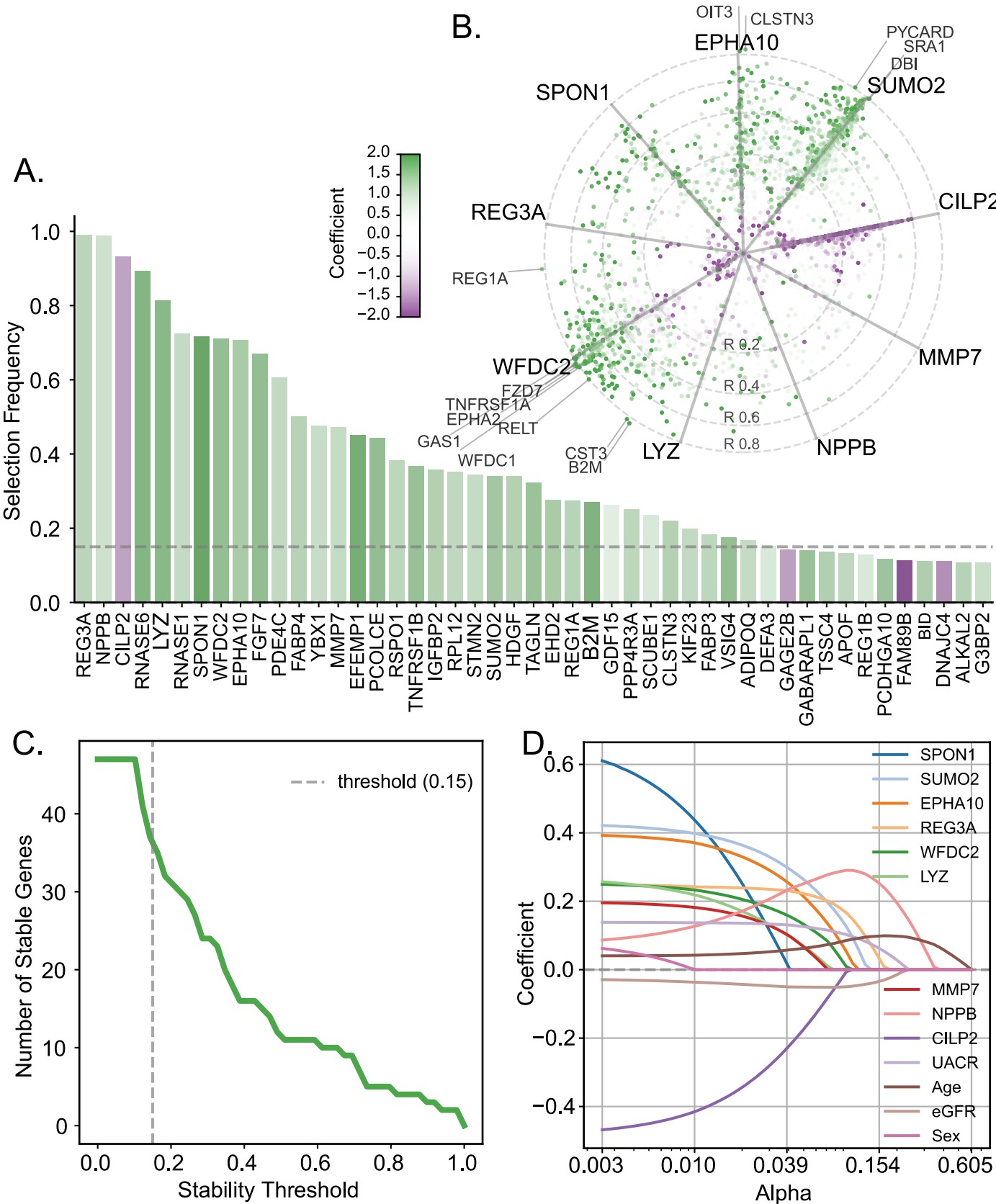

**Extended Data Fig. 2 | See next page for caption.**

**Extended Data Fig. 2 | Marker Selection Stability Analysis. (A)** Each bar represents a protein, with the height corresponding to the frequency at which the protein was selected. **(B)** Radial protein correlation network. APRS proteins are positioned on the outer ring (R = 0.8) and ordered by hierarchical clustering. Peripheral proteins (dots) are placed by correlation strength (radial distance) and direction (angular position) relative to the core. Color represents regression coefficients from the Cox model, with a diverging scale from negative to positive. **(C)** The cumulative distribution function plot shows the proportion of genes selected by threshold. The x-axis represents the selection frequency, and the y-axis represents the cumulative probability. **(D)** Coefficient paths for selected features across different alpha values. Coefficient trajectories are shown for key features in a penalized regression model as the regularization parameter alpha varies.

**Extended Data Fig. 3 | Correlation matrices of biomarkers in *APOL1* high-risk individuals with eGFR <60 and ≥60 mL/min/1.73 m².** Pearson correlation coefficients between 9 proteins and age, estimated glomerular filtration rate (eGFR), sex and urine albumin-to-creatinine ratio (UACR) are shown for *APOL1* high-risk individuals with eGFR <60 (upper right, n = 262) and ≥60 mL/min/1.73 m² (lower left, n = 680). Each cell represents the pairwise correlation between two proteins, and the numeric value of the correlation appears inside each cell.

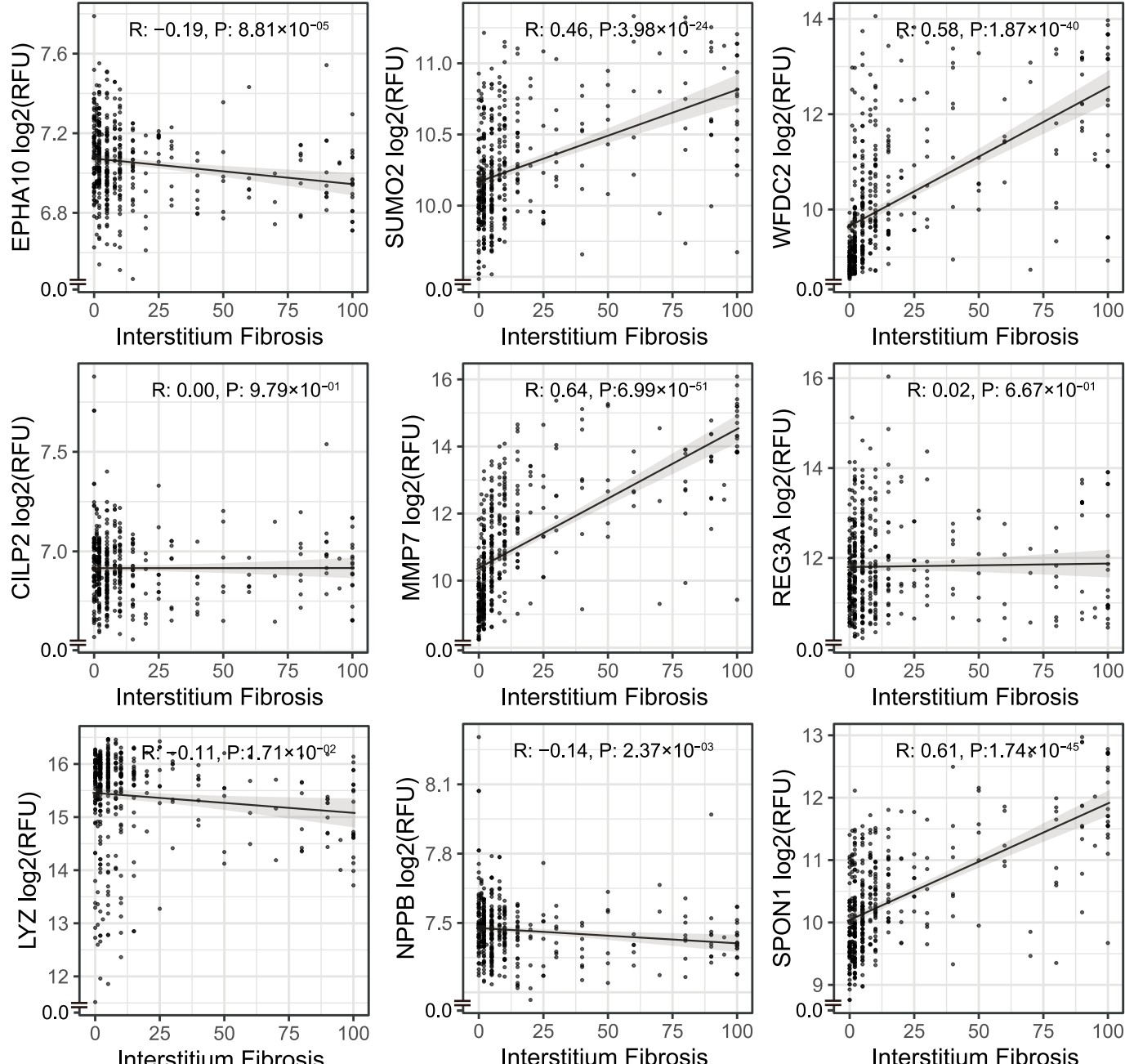

**Extended Data Fig. 4 | Correlation between kidney fibrosis and biomarker protein levels.** Scatter plots showing the association between kidney interstitium fibrosis score (x-axis) and $\log_2$-transformed protein levels (relative fluorescence units, RFUs, n = 325) of nine biomarkers (y-axis). Pearson correlation coefficients (R) and corresponding two-sided P values are shown in each panel. Solid lines represent fitted mean values from linear regression models, and shaded areas indicate 95% confidence intervals.

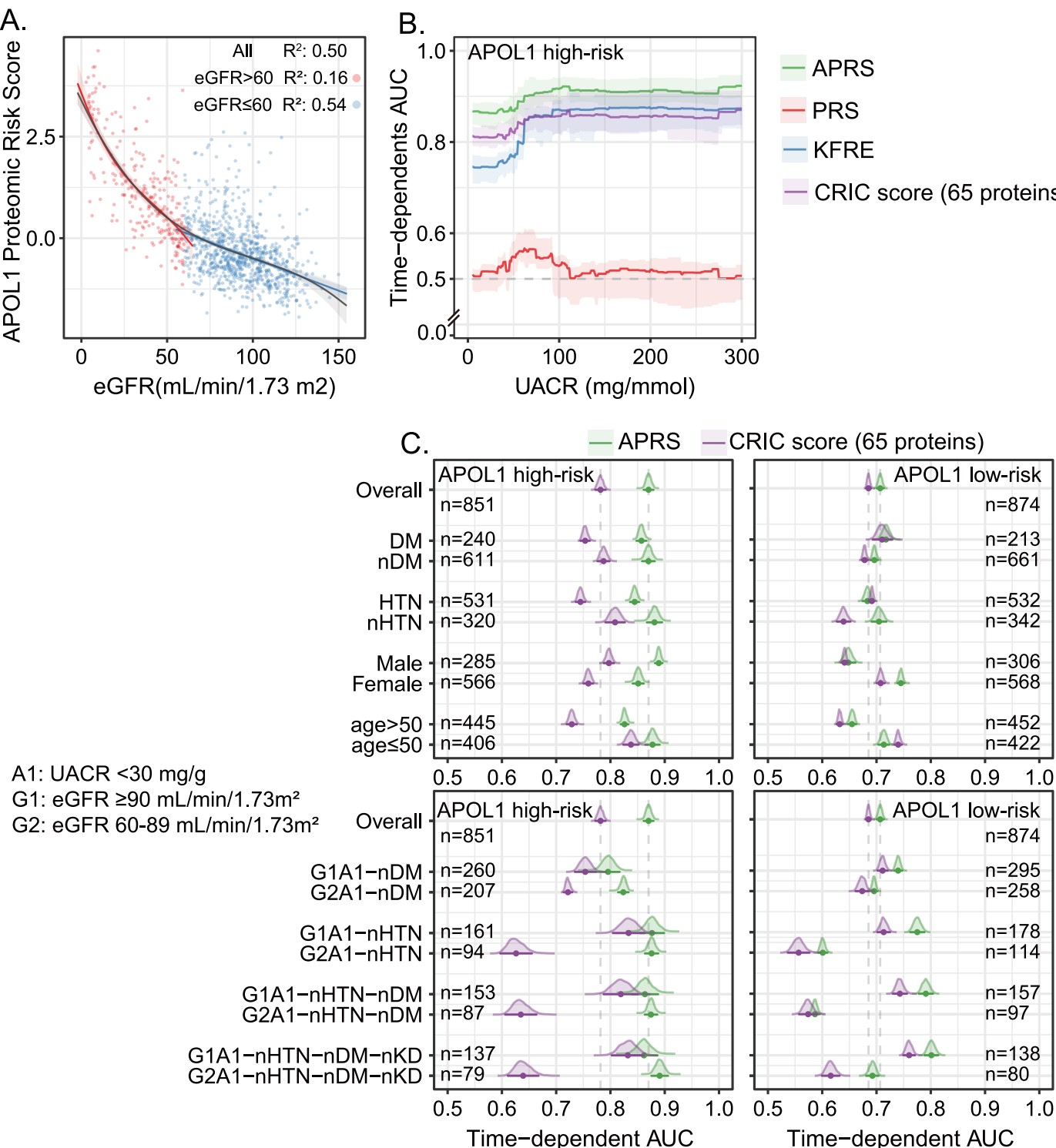

A1: UACR <30 mg/g
G1: eGFR ≥90 mL/min/1.73m²
G2: eGFR 60-89 mL/min/1.73m²

**Extended Data Fig. 5 | Performance of proteomic risk scores in predicting kidney outcomes stratified by *APOL1* genotype and clinical subgroups.** (**A**) Scatter plot showing the correlation between APRS and estimated glomerular filtration rate (eGFR) across all participants (black), participants with normal eGFR (≥60 mL/min/1.73 m², n = 851), and participants with reduced eGFR (<60 mL/min/1.73 m², n = 262). Solid lines represent locally weighted mean estimates from locally estimated scatterplot smoothing, and shaded areas indicate 95% confidence intervals. Corresponding R² values are shown for each subgroup. (**B**) Time-dependent area under the curve and 95% confidence intervals plotted as a function of baseline urine albumin-to-creatinine ratio (UACR, mg/mmol), illustrating how discriminative accuracy varies across levels of kidney function (n = 171). (**C**) Subgroup-specific time-dependent AUCs for APRS. Points represent mean tAUC estimates. Horizontal lines and shaded densities represent 95% confidence intervals. The dashed vertical line indicates the overall median tAUC. DM, diabetes mellitus; HTN, hypertension. G1 and G2 denote eGFR categories (G1: eGFR ≥90; G2: 60–89 mL/min/1.73 m²). A1 denotes UACR categories <30 mg/g. Combined labels represent joint eGFR and UACR strata, and suffixes non-DM (nDM), non-HTN (nHTN); nKD excludes any history of kidney disease, including hematuria or ureteral disorders.

A.

| APRS | ACR | eGFR | Age | 5 years event | | | | Age | 10 years event | | | | Age | 5 years event | | | | Age | 10 years event | | | |
|---|---|---|---|---|---|---|---|---|---|---|---|---|---|---|---|---|---|---|---|---|---|---|
| | | | | Non | DM | HTN | Both | | Non | DM | HTN | Both | | Non | DM | HTN | Both | | Non | DM | HTN | Both |
| 18.1 (Q5) | <30 | >90 | | 0.4 | 0.8 | 0.9 | 2.0 | | 0.7 | 1.3 | 1.6 | 3.5 | | 0.5 | 1.0 | 1.3 | 2.8 | | 0.9 | 1.8 | 2.2 | 4.7 |
| | | 60-89 | | 0.4 | 0.9 | 1.1 | 2.3 | | 0.8 | 1.5 | 1.8 | 3.9 | | 0.6 | 1.2 | 1.4 | 3.1 | | 1.0 | 2.0 | 2.5 | 5.2 |
| | >30 | >90 | | 0.5 | 0.9 | 1.1 | 2.4 | | 0.8 | 1.6 | 1.9 | 4.2 | | 0.6 | 1.2 | 1.5 | 3.3 | | 1.1 | 2.1 | 2.6 | 5.6 |
| | | 60-89 | | 0.5 | 1.0 | 1.3 | 2.7 | | 0.9 | 1.8 | 2.2 | 4.7 | | 0.7 | 1.4 | 1.7 | 3.7 | | 1.2 | 2.4 | 3.0 | 6.3 |
| 18.8 (Q20) | <30 | >90 | | 0.7 | 1.4 | 1.7 | 3.6 | | 1.2 | 2.3 | 2.9 | 6.1 | | 1.0 | 1.8 | 2.3 | 4.9 | | 1.6 | 3.2 | 3.9 | 8.3 |
| | | 60-89 | | 0.8 | 1.5 | 1.9 | 4.0 | | 1.4 | 2.6 | 3.2 | 6.8 | | 1.1 | 2.1 | 2.5 | 5.4 | | 1.8 | 3.5 | 4.3 | 9.2 |
| | >30 | >90 | | 0.8 | 1.6 | 2.0 | 4.3 | | 1.5 | 2.8 | 3.4 | 7.3 | | 1.1 | 2.2 | 2.7 | 5.8 | | 2.0 | 3.8 | 4.7 | 9.9 |
| | | 60-89 | | 0.9 | 1.8 | 2.3 | 4.8 | | 1.6 | 3.1 | 3.8 | 8.1 | | 1.3 | 2.5 | 3.1 | 6.5 | | 2.2 | 4.2 | 5.2 | 11.0 |
| 19.3 (Q35) | <30 | >90 | | 1.0 | 2.0 | 2.5 | 5.3 | | 1.8 | 3.4 | 4.2 | 8.9 | | 1.4 | 2.7 | 3.4 | 7.1 | | 2.4 | 4.6 | 5.7 | 12.0 |
| | | 60-89 | | 1.2 | 2.2 | 2.7 | 5.9 | | 2.0 | 3.8 | 4.6 | 9.8 | | 1.6 | 3.0 | 3.7 | 7.9 | | 2.7 | 5.2 | 6.4 | 13.3 |
| | >30 | >90 | | 1.3 | 2.4 | 3.0 | 6.3 | | 2.1 | 4.1 | 5.0 | 10.6 | | 1.7 | 3.3 | 4.0 | 8.5 | | 2.9 | 5.6 | 6.8 | 14.2 |
| | | 60-89 | | 1.4 | 2.7 | 3.3 | 7.0 | | 2.4 | 4.5 | 5.6 | 11.8 | | 1.9 | 3.7 | 4.5 | 9.5 | | 3.2 | 6.2 | 7.6 | 15.8 |
| 19.7 (Q50) | <30 | >90 | >50 | 1.5 | 2.8 | 3.4 | 7.3 | >50 | 2.5 | 4.7 | 5.7 | 12.2 | ≤50 | 2.0 | 3.8 | 4.7 | 9.9 | ≤50 | 3.4 | 6.5 | 7.9 | 16.4 |
| | | 60-89 | | 1.6 | 3.1 | 3.8 | 8.1 | | 2.8 | 5.2 | 6.3 | 13.4 | | 2.2 | 4.2 | 5.2 | 11.0 | | 3.8 | 7.2 | 8.8 | 18.1 |
| | >30 | >90 | | 1.8 | 3.4 | 4.1 | 8.8 | | 3.0 | 5.7 | 6.9 | 14.5 | | 2.4 | 4.6 | 5.6 | 11.8 | | 4.1 | 7.7 | 9.5 | 19.4 |
| | | 60-89 | | 2.0 | 3.7 | 4.6 | 9.7 | | 3.4 | 6.3 | 7.6 | 16.0 | | 2.6 | 5.1 | 6.3 | 13.1 | | 4.5 | 8.6 | 10.5 | 21.4 |
| 20.2 (Q65) | <30 | >90 | | 2.3 | 4.3 | 5.2 | 11.1 | | 3.9 | 7.1 | 8.7 | 18.2 | | 3.1 | 5.9 | 7.2 | 15.0 | | 5.2 | 9.8 | 12.0 | 24.3 |
| | | 60-89 | | 2.5 | 4.7 | 5.8 | 12.3 | | 4.3 | 7.8 | 9.5 | 19.9 | | 3.4 | 6.5 | 8.0 | 16.6 | | 5.8 | 10.9 | 13.3 | 26.7 |
| | >30 | >90 | | 2.7 | 5.2 | 6.3 | 13.3 | | 4.7 | 8.6 | 10.4 | 21.6 | | 3.7 | 7.0 | 8.6 | 17.7 | | 6.3 | 11.8 | 14.3 | 28.5 |
| | | 60-89 | | 3.0 | 5.7 | 7.0 | 14.7 | | 5.2 | 9.5 | 11.5 | 23.6 | | 4.1 | 7.8 | 9.6 | 19.6 | | 7.0 | 13.0 | 15.8 | 31.3 |
| 21.0 (Q80) | <30 | >90 | | 4.5 | 8.2 | 9.9 | 20.6 | | 7.6 | 13.2 | 15.8 | 32.0 | | 6.0 | 11.2 | 13.7 | 27.4 | | 10.1 | 18.4 | 22.2 | 42.2 |
| | | 60-89 | | 5.0 | 8.9 | 10.8 | 22.5 | | 8.4 | 14.3 | 16.9 | 34.4 | | 6.7 | 12.4 | 15.1 | 30.0 | | 11.2 | 20.2 | 24.2 | 45.6 |
| | >30 | >90 | | 5.3 | 9.8 | 11.9 | 24.3 | | 9.0 | 15.9 | 19.0 | 37.4 | | 7.2 | 13.4 | 16.3 | 32.0 | | 12.0 | 21.8 | 26.2 | 48.4 |
| | | 60-89 | | 6.0 | 10.8 | 13.0 | 26.6 | | 10.1 | 17.2 | 20.5 | 40.3 | | 8.0 | 14.8 | 18.0 | 35.0 | | 13.4 | 23.9 | 28.6 | 52.1 |
| 22.9 (Q95) | <30 | >90 | | 19.6 | 29.5 | 33.8 | 61.4 | | 31.4 | 40.1 | 44.0 | 74.0 | | 25.5 | 41.9 | 48.4 | 76.6 | | 39.9 | 58.7 | 65.2 | 89.7 |
| | | 60-89 | | 21.6 | 30.8 | 34.8 | 63.2 | | 34.4 | 40.2 | 43.4 | 73.8 | | 28.1 | 44.9 | 51.4 | 79.6 | | 43.5 | 61.3 | 67.3 | 90.7 |
| | >30 | >90 | | 23.1 | 35.3 | 40.3 | 69.0 | | 36.4 | 47.7 | 52.3 | 81.0 | | 29.9 | 48.4 | 55.4 | 82.8 | | 45.9 | 66.1 | 72.5 | 93.5 |
| | | 60-89 | | 25.4 | 37.0 | 41.7 | 71.0 | | 39.8 | 48.3 | 52.1 | 80.9 | | 32.8 | 51.7 | 58.6 | 85.4 | | 49.7 | 68.8 | 74.7 | 94.1 |

B.

Both: Diabetes and Hypertension
Non: Neither Diabetes nor Hypertension
DM: Diabetes only
HTN: Hypertension only

Cumulative Incidence (%)
0.5  1.3  2.2  21.0  57.5  90.0

**Extended Data Fig. 6 | Risk of Kidney Failure and Mortality by Age, APRS, and Comorbidity Status.** (**A**) Cumulative incidence (%) estimates for individuals aged ≤50 and >50, across APRS quantiles (Q5, Q20, Q35, Q50, Q65, Q80, Q95) and for four comorbidity groups: neither diabetes nor hypertension (Non), diabetes only (DM), hypertension only (HTN), and both diabetes and hypertension (Both). Higher APRS values and the presence of comorbidities are associated with increased risks. For example, individuals with an APRS of 20, eGFR >90, UACR < 30, age <50, and without DM or HTN have a 5-year risk of 1.5-2.3%. (**B**) The graph illustrates the relationship between mortality (%) and kidney failure (%) across APRS using Fine-Gray Cause-Specific Cox model, highlighting the upward trend of both outcomes with increasing APRS scores.

**Extended Data Table 1 | Biomarkers metrics on composite event in *APOL1* high-risk population with eGFR ≥60 mL/min/1.73 m²**

| SOMAmer | NCBI Gene | UniProt | Protein alias | β | P | C | tAUC | Spec. | Sens. | F1 | Prec. | Rec. | Acc. | FPR | FNR | MCC | PPV | NPV |
|---|---|---|---|---|---|---|---|---|---|---|---|---|---|---|---|---|---|---|
| seq.11388.75 | *WFDC2* | Q14508 | HE4 | 1.37 | $5.19 \times 10^{-20}$ | 73.53 | 74.57 | 73.20 | 67.14 | 58.76 | 53.29 | 66.56 | 70.50 | 26.8 | 32.86 | 36.38 | 52.41 | 80.76 |
| seq.15304.1 | *REG3A* | Q06141 | PAP1 | 0.89 | $4.91 \times 10^{-17}$ | 71.31 | 72.48 | 70.05 | 64.16 | 56.79 | 48.38 | 70.35 | 68.56 | 29.95 | 35.84 | 31.76 | 50.06 | 79.62 |
| seq.19555.1 | *SUMO2* | P61956 | SUMO2 | 1.18 | $7.84 \times 10^{-20}$ | 71.81 | 72.07 | 64.48 | 71.13 | 56.14 | 47.6 | 68.99 | 66.70 | 35.52 | 28.87 | 31.56 | 47.42 | 80.91 |
| seq.2789.26 | *MMP7* | P09237 | MMP-7 | 0.68 | $1.45 \times 10^{-8}$ | 66.68 | 67.72 | 76.69 | 54.92 | 53.82 | 44.39 | 69.34 | 69.44 | 23.31 | 45.08 | 29.62 | 51.08 | 76.90 |
| seq.4297.62 | *SPON1* | Q9HCB6 | Spondin1 | 1.72 | $6.58 \times 10^{-22}$ | 70.88 | 71.91 | 79.08 | 59.06 | 57.34 | 51.41 | 67.26 | 72.38 | 20.92 | 40.94 | 35.92 | 55.33 | 78.83 |
| seq.4920.10 | *LYZ* | P61626 | Lysozyme | 1.56 | $7.41 \times 10^{-18}$ | 71.80 | 72.97 | 75.42 | 63.85 | 57.90 | 51.78 | 66.55 | 72.49 | 24.58 | 36.15 | 36.48 | 53.63 | 80.52 |
| seq.6036.78 | *EPHA10* | Q5JZY3 | EPHAA | 1.08 | $3.31 \times 10^{-14}$ | 70.92 | 72.50 | 75.46 | 60.76 | 56.89 | 47.00 | 73.14 | 71.23 | 24.54 | 39.24 | 33.84 | 52.89 | 79.10 |
| seq.7655.11 | *NPPB* | P16860 | BNP | 0.55 | $4.21 \times 10^{-26}$ | 71.40 | 71.38 | 71.80 | 64.19 | 56.25 | 45.99 | 73.08 | 69.11 | 28.2 | 35.81 | 32.35 | 50.13 | 79.23 |
| seq.8841.65 | *CILP2* | Q8IUL8 | CILP2 | -0.95 | $7.08 \times 10^{-18}$ | 69.77 | 68.70 | 54.94 | 75.72 | 55.11 | 43.61 | 76.94 | 62.34 | 45.06 | 24.28 | 27.35 | 43.67 | 81.20 |
| Age (pre 10 years) | | | | 0.19 | $3.73 \times 10^{-14}$ | 66.13 | 66.92 | 61.99 | 65.11 | 53.50 | 40.95 | 79.57 | 60.43 | 38.01 | 34.89 | 23.88 | 44.11 | 77.26 |
| Gender (male = 1, female = 0) | | | | 0.49 | $4.83 \times 10^{-3}$ | 56.63 | 56.39 | 68.61 | 44.17 | 47.82 | 34.09 | 86.72 | 60.59 | 31.39 | 55.83 | 11.59 | 39.49 | 71.16 |
| UACR (per doubling) | | | | 0.22 | $2.20 \times 10^{-5}$ | 63.57 | 63.62 | 72.82 | 50.56 | 51.16 | 39.53 | 73.41 | 64.30 | 27.18 | 49.44 | 21.64 | 46.04 | 74.29 |
| eGFR (per 5 mL/min/1.73 m²) | | | | -0.27 | $1.61 \times 10^{-8}$ | 64.56 | 65.08 | 56.91 | 68.62 | 51.99 | 39.76 | 76.98 | 61.64 | 43.09 | 31.38 | 22.72 | 42.40 | 78.11 |

β, regression coefficient from a univariable Cox proportional hazards model; P, two-sided Wald test P value from the Cox model, adjusted for multiple comparisons using the Benjamini–Hochberg method; C, concordance index; AUC, time-dependent area under the receiver-operating-characteristic curve; Spec., specificity; Sens., sensitivity; F1, F1 score; Prec., precision; Rec., recall; Acc., accuracy; FPR, false positive rate; FNR, false negative rate; MCC, Matthews correlation coefficient; PPV, positive predictive value; NPV, negative predictive value.

**Extended Data Table 2 | Hazard Ratio for APOL1 Proteomic Risk Score (APRS) Stratified by *APOL1* Genotype**

| African ancestry eGFR≥60 | N | Variable | β | Hazard Ratio | β 95% CI | Hazard Ratio 95% CI | z | p-value |
|---|---|---|---|---|---|---|---|---|
| *APOL1* low-risk | 874 | APRS | 0.75 | 2.11 | 0.57-0.92 | 1.77-2.51 | 8.39 | $4.86 \times 10^{-17}$ |
| *APOL1* high-risk | 851 | APRS | 1.01 | 2.75 | 0.89-1.13 | 2.45-3.1 | 16.78 | $3.42 \times 10^{-63}$ |
| Interaction | 1,725 | *APOL1* × APRS | 0.3 | 1.35 | 0.09-0.51 | 1.09-1.67 | 2.8 | $5.11 \times 10^{-3}$ |

eGFR, estimated glomerular filtration rate (mL/min/1.73 m$^2$); β, regression coefficient from Cox proportional hazards model for composite event; hazard ratios with 95% confidence intervals (CIs) are reported; z, Wald test statistic; P, two-sided Wald test P value.

**Extended Data Table 3 | Discrimination performance of risk scores for different events across *APOL1* risk groups and eGFR strata**

| Ancestry *APOL1* genotype | eGFR | Group | N | Score | All event tAUC | All event events% | Death tAUC | Death events% | Kidney event tAUC | Kidney event events% |
|---|---|---|---|---|---|---|---|---|---|---|
| African ancestry High risk | ≥60 | Training | 680 | APRS | 86.7(1.4) | 18.5 | 88.3(1.6) | 7.2 | 86.0(2.7) | 14.1 |
| | | | | CRIC | 79.7(1.9) | | 76.6(4.2) | | 80.8(2.6) | |
| | | | | KFRE | 66.6(1.5) | | 65.4(1.5) | | 70.1(2.2) | |
| | | | | PRS | 48.0(2.9) | | 41.8(6.8) | | 52.8(1.3) | |
| | ≥60 | Test | 171 | APRS | 86.5(3.9) | 15.8 | 85.7(1.8) | 7.6 | 88.1(1.6) | 14.0 |
| | | | | CRIC | 79.0(4.0) | | 79.2(3.3) | | 83.4(1.6) | |
| | | | | KFRE | 66.2(4.3) | | 46.6(3.3) | | 67.8(2.3) | |
| | | | | PRS | 58.5(5.2) | | 48.1(3.8) | | 48.4(4.6) | |
| | <60 | Test | 262 | APRS | 84.2(1.9) | 55.3 | 85.3(2.1) | 21.8 | 86.4(1.8) | 47.7 |
| | | | | CRIC | 83.2(1.7) | | 77.1(3.2) | | 86.1(1.2) | |
| | | | | KFRE | 82.4(2.1) | | 79.9(4.2) | | 86.4(1.6) | |
| | | | | PRS | 53.8(1.4) | | 56.8(1.4) | | 53.3(1.5) | |
| African ancestry Low risk | Any | Test | 912 | APRS | 73.1(2.1) | 15.1 | 79.9(2.7) | 7.7 | 69.1(1.6) | 9.6 |
| | | | | CRIC | 69.3(1.3) | | 71.6(1.0) | | 69.0(2.3) | |
| | | | | KFRE | 60.1(1.5) | | 53.9(2.2) | | 59.3(1.0) | |
| | | | | PRS | 50.5(1.1) | | 50.3(1.4) | | 46.8(1.3) | |

eGFR, estimated glomerular filtration rate (mL/min/1.73 m²); tAUC, time-dependent area under the receiver operating characteristic curve over 10 years (standard deviation); APRS, APOL1 Proteomic Risk Score; CRIC, Chronic Renal Insufficiency Cohort proteomic risk score; KFRE, Kidney Failure Risk Equation; PRS, genome-wide polygenic risk score for chronic kidney disease.

**Extended Data Table 4 | Demographics and clinical outcomes of validation cohorts**

| | *APOL1* high-risk | | African ancestry *APOL1* low-risk | | | European ancestry *APOL1* low-risk | | |
| --- | --- | --- | --- | --- | --- | --- | --- | --- |
| | ARIC | UKBB | PMBB | ARIC | UKBB | PMBB | ARIC | UKBB |
| Number | 314 | 204 | 912 | 2021 | 967 | 698 | 8,602 | 46,863 |
| Age - yr | 55.8 ±5.6 | 52.2±7.8 | 50.1±14.1 | 56.4±5.8 | 51.95±8.0 | 54.2±15.2 | 57.2 ±5.7 | 57.6±8.1 |
| Female - no. (%) | 206 (65.6) | 81(39.7) | 593(65.0) | 1277 (63.2) | 461(47.7) | 314(45.0) | 4,631(53.8) | 21,628(46.1) |
| Body mass index - kg/m² | 30.0±6.2 | 30.1±5.3 | 32.3±7.9 | 30.0±6.2 | 29.5±5.5 | 28.7±6.5 | 27.3 ±4.9 | 27.4±4.7 |
| eGFR – mL/min/1.73m² | 96.2±20.1 | 87.1±15.6 | 89.9±19.0 | 97.6±18.9 | 90.7±15.6 | 91.5±18.8 | 98.2 ±15.8 | 93.1±13.8 |
| Diabetes Mellitus - no. (%) | 81 (26.0) | 7(3.4) | 223(24.5) | 534 (26.6) | 37(3.8) | 135(19.3) | 989 (11.5) | 1,097(2.3) |
| Hypertension- no. (%) | 192 (61.5) | 20(9.8) | 560(61.4) | 1101 (54.9) | 86(8.9) | 340(48.7) | 2,521 (29.4) | 4,378(9.3) |
| Kidney event - no. (%) | | 7(3.4) | 88(9.7) | | 46(4.8) | 63(9.0) | | 2,230(4.8) |
| Years to kidney event | | 5.7±3.9 | 3.8±2.8 | | 5.1±2.6 | 7.3±3.0 | | 4.8±3.2 |
| Death - no. (%) | | 6(2.9) | 70(7.7) | | 21(2.2) | 44(6.3) | | 2,465(5.3) |
| Years to Death | | 5.9±2.4 | 4.03±3.1 | | 5.1±2.6 | 5.1±2.4 | | 5.4±2.5 |
| Composite Event - no. (%) | 43 (13.7) | 13(6.4) | 138(15.1) | 321 (15.9) | 66(6.8) | 94(13.5) | 907 (10.5) | 4,385(9.4) |
| Follow-up time - yr | 9.4 ±1.7 | 8.39±4.9 | 7.2±2.9 | 9.3 ±2.0 | 8.8±4.6 | 7.3±3.0 | 9.5 ±1.6 | 11.1±4.0 |

ARIC, Atherosclerosis Risk in Communities study; UKBB, UK Biobank; PMBB, Penn Medicine BioBank; eGFR, estimated glomerular filtration rate

# Reporting Summary

## Statistics

For all statistical analyses, confirm that the following items are present in the figure legend, table legend, main text, or Methods section.

| n/a | Confirmed | |
|---|---|---|
| ☐ | ☒ | The exact sample size (*n*) for each experimental group/condition, given as a discrete number and unit of measurement |
| ☐ | ☒ | A statement on whether measurements were taken from distinct samples or whether the same sample was measured repeatedly |
| ☐ | ☒ | The statistical test(s) used AND whether they are one- or two-sided<br>*Only common tests should be described solely by name; describe more complex techniques in the Methods section.* |
| ☐ | ☒ | A description of all covariates tested |
| ☐ | ☒ | A description of any assumptions or corrections, such as tests of normality and adjustment for multiple comparisons |
| ☐ | ☒ | A full description of the statistical parameters including central tendency (e.g. means) or other basic estimates (e.g. regression coefficient) AND variation (e.g. standard deviation) or associated estimates of uncertainty (e.g. confidence intervals) |
| ☐ | ☒ | For null hypothesis testing, the test statistic (e.g. *F*, *t*, *r*) with confidence intervals, effect sizes, degrees of freedom and *P* value noted<br>*Give P values as exact values whenever suitable.* |
| ☒ | ☐ | For Bayesian analysis, information on the choice of priors and Markov chain Monte Carlo settings |
| ☐ | ☒ | For hierarchical and complex designs, identification of the appropriate level for tests and full reporting of outcomes |
| ☐ | ☒ | Estimates of effect sizes (e.g. Cohen's *d*, Pearson's *r*), indicating how they were calculated |

*Our web collection on statistics for biologists contains articles on many of the points above.*

## Software and code

Policy information about availability of computer code

| | |
|---|---|
| Data collection | The PMBB WES Release 3.0 used bcl2fastq (Illumina), BWA-MEM, DeepVariant v0.10 (Parabricks-accelerated), GLnexus v1.4.3, PLINK v1.9, bcftools norm, VEP v113.0 with plugins (AlphaMissense, CADD, ClinVar, dbNSFP 4.9, gnomAD, LOFTEE, SpliceAI), KING, R (KernSmooth), Python, smartpca (EIGENSOFT), and the SAIGE-GENE pipeline (Nextflow). STAR (v2.7.3a) was used to align the bulk RNA-seq data to the human genome.Proteomic profiling was performed using SOMAscan v4.1 with data processed in the SOMAscan Data Analysis Software (SomaLogic) and microarrays scanned on the SureScan Dx Microarray Scanner (Agilent Technologies). |
| Data analysis | For RNA-seq data processing, TrimGalore (v0.4.5) was used for trimming adaptors and low-quality bases, and STAR (v2.7.3a) was employed for aligning reads to the human genome. RSEM (v1.3.0) was used to estimate gene and isoform expression levels in transcripts per million. The Python environment for this analysis included lifelines 0.28.0 for survival analysis, matplotlib 3.7.5 for plotting, numpy 1.24.4 and scipy 1.9.3 for numerical and statistical computations, pandas 1.5.3 for data manipulation, scikit-learn 1.3.2 for model building, sksurv 0.22.2 for survival models with censored data, and all running on Python 3.10.12 (GCC 11.4.0). |

For manuscripts utilizing custom algorithms or software that are central to the research but not yet described in published literature, software must be made available to editors and reviewers. We strongly encourage code deposition in a community repository (e.g. GitHub). See the Nature Portfolio guidelines for submitting code & software for further information.

# Data

Policy information about availability of data

All manuscripts must include a data availability statement. This statement should provide the following information, where applicable:

- Accession codes, unique identifiers, or web links for publicly available datasets
- A description of any restrictions on data availability
- For clinical datasets or third party data, please ensure that the statement adheres to our policy

The datasets analyzed in this study are not publicly available due to participant privacy and data use agreements but may be accessed upon reasonable application to the PMBB (https://pmbb.med.upenn.edu/), subject to approval. Requests for proteomic data or verification analyses may be directed to the corresponding author (ksusztak@pennmedicine.upenn.edu). An initial response is generally provided within approximately two weeks. ARIC data may be requested from the ARIC Data Coordinating Center by obtaining study approval, executing a Data and Materials Distribution Agreement, and submitting a Data Request Form to aricdata@unc.edu. Alternatively, ARIC data are available through the NHLBI BioLINCC (https://biolincc.nhlbi.nih.gov/) repository and dbGaP subject to their application procedures (https://dbgap.ncbi.nlm.nih.gov/beta/study/phs000280.v9.p3). Review timelines typically require approximately 4–8 weeks. For datasets derived from the UKBB, data access is governed by the UK Biobank's established policies. Researchers must apply through the UKBB Access Management System, available at https://www.ukbiobank.ac.uk/, outlining the purpose of the intended use. Applications are reviewed by UK Biobank, and decisions are generally provided within approximately 4–6 weeks.

# Research involving human participants, their data, or biological material

Policy information about studies with human participants or human data. See also policy information about sex, gender (identity/presentation), and sexual orientation and race, ethnicity and racism.

| | |
|---|---|
| Reporting on sex and gender | For PMBB cohort, sex was determined based on genetic data obtained from whole genome sequencing for the study participants. The data included information on participants being male or female; there were no sex/gender-based inclusion or exclusion criteria. |
| Reporting on race, ethnicity, or other socially relevant groupings | Race and ethnicity information was determined based on genetic data obtained from whole genome sequencing for the study participants. |
| Population characteristics | Baseline characteristics are described in Table 1, Supplemental Table 1 and Extended Data Table 4. |
| Recruitment | The PMBB, which has enrolled over 250,000 participants since 2008, has performed whole exome sequencing on approximately 57,170 individuals. Of these, 1,310 carried the high-risk APOL1 genotype, and after excluding 197 participants with prior kidney transplantation or kidney failure, 1,113 African American participants (G1/G1, G2/G2, or G1/G2) were included in this study. Additional 912 African ancestry APOL1 low-risk participants and 698 European ancestry APOL1 low-risk were included as external reference. |
| Ethics oversight | The data and biospecimen used in this study in PMBB were approved by the University of Pennsylvania Institutional Review Board (protocol 815796, 813913 and 857403) and all participants provided informed consent for genetic and EHR research. The validation approved by the University of Pennsylvania Institutional Review Board (protocol 855821). ARIC and UK Biobank investigations were conducted under their respective ethics approvals (UKB ethics: https://www.ukbiobank.ac.uk/about-us/how-we-work/ethics/ ; ARIC application MP4524; UKB application 273810). The ARIC Study adhered to ethics regulations from and was approved by a single Institutional Review Board at Johns Hopkins School of Medicine (FWA00005752; IRB00311861) and Institutional Review Boards at all participating institutions: University of North Carolina at Chapel Hill, Johns Hopkins University School of Public Health, University of Minnesota, Wake Forest University Health Sciences, University of Mississippi Medical Center, Baylor College of Medicine, University of Texas Houston Health Science Center, and Brigham and Women's Hospital. Study participants provided written informed consent at all study visits. All procedures will comply with the Declaration of Helsinki and applicable regulations. Participant privacy will be protected by using de-identified data, secure servers with encryption, and HIPAA-compliant protocols. Data use agreements restrict access to authorized personnel, and data transfer (for validation analyses) will employ secure, encrypted channels. |

Note that full information on the approval of the study protocol must also be provided in the manuscript.

# Field-specific reporting

Please select the one below that is the best fit for your research. If you are not sure, read the appropriate sections before making your selection.

☒ Life sciences ☐ Behavioural & social sciences ☐ Ecological, evolutionary & environmental sciences

For a reference copy of the document with all sections, see nature.com/documents/nr-reporting-summary-flat.pdf

# Life sciences study design

All studies must disclose on these points even when the disclosure is negative.

| | |
|---|---|
| Sample size | Sample size was estimated using the Schoenfeld method for the Cox proportional hazards model and an events per variable (EPV) approach (with EPV set at 15). The EPV method yielded the required sample size of 640 participants. In contrast, based on a two-sided significance level of 0.05, an expected 30% marker-positive rate, an anticipated event rate of 25%, and a target hazard ratio of 1.648—assuming median dichotomization of the model score, a total of 685 participants was determined to provide an effective statistical power of ~85%. |
| Data exclusions | We excluded 197 participants with prior kidney transplantation or kidney failure. |
| Replication | The APRS was independently validated in the Atherosclerosis Risk in Communities study and the UK Biobank, and the analyses in each cohort were performed independently by separate analysts.<br>To ensure reproducibility and consistent results, all analyses were conducted using fixed random seeds (42 or 0), implemented in Python using random.seed() and numpy.random.seed(). |
| Randomization | This is not application as this was an observational cohort study. |
| Blinding | Blinding was not applicable to this study. |

# Reporting for specific materials, systems and methods

We require information from authors about some types of materials, experimental systems and methods used in many studies. Here, indicate whether each material, system or method listed is relevant to your study. If you are not sure if a list item applies to your research, read the appropriate section before selecting a response.

## Materials & experimental systems

| n/a | Involved in the study |
|---|---|
| ☒ | Antibodies |
| ☒ | Eukaryotic cell lines |
| ☒ | Palaeontology and archaeology |
| ☒ | Animals and other organisms |
| ☒ | Clinical data |
| ☒ | Dual use research of concern |
| ☒ | Plants |

## Methods

| n/a | Involved in the study |
|---|---|
| ☒ | ChIP-seq |
| ☒ | Flow cytometry |
| ☒ | MRI-based neuroimaging |

## Plants

| | |
|---|---|
| Seed stocks | *Report on the source of all seed stocks or other plant material used. If applicable, state the seed stock centre and catalogue number. If plant specimens were collected from the field, describe the collection location, date and sampling procedures.* |
| Novel plant genotypes | *Describe the methods by which all novel plant genotypes were produced. This includes those generated by transgenic approaches, gene editing, chemical/radiation-based mutagenesis and hybridization. For transgenic lines, describe the transformation method, the number of independent lines analyzed and the generation upon which experiments were performed. For gene-edited lines, describe the editor used, the endogenous sequence targeted for editing, the targeting guide RNA sequence (if applicable) and how the editor was applied.* |
| Authentication | *Describe any authentication procedures for each seed stock used or novel genotype generated. Describe any experiments used to assess the effect of a mutation and, where applicable, how potential secondary effects (e.g. second site T-DNA insertions, mosiacism, off-target gene editing) were examined.* |

