## [Peer Review File · Nature Medicine]

Proteomic risk score for early prediction of kidney disease progression in individuals with APOL1 high-risk genotypes

Corresponding Author: Professor Katalin Susztak

Version 0:

Reviewer comments:

Reviewer #1

(Remarks to the Author)

Individuals with APOL1 high risk genotypes for kidney disease have an approximately 20% likelihood of developing advanced chronic kidney disease (CKD). To date, clinical actionable prognostication tools to identify the individuals with greatest risk have not been developed. Such algorithms are critical to effective management and counseling of individuals with the APOL1 risk genotype. In addition, such a tool would accelerate clinical trials as the pipeline for targeted APOL1 kidney disease therapies grows. It also fills a significant need to address, in part, the excess risk for advanced CKD in Black people worldwide.

These authors derived an APOL1 Proteomic Risk Score (APRS) using a training cohort of available individuals (80%) with high-risk APOL1 genotypes (APOL1-HR) in the PMBB, tested it in the remaining 20% and validated the tool in 2 additional cohorts of Black individuals in ARIC and UKBB with available APOL1 genotypes. The APRS had excellent discrimination and excellent risk stratification between the lowest and highest outcome quintiles and was the top performer in a clinical decision analysis with other prognostic algorithms including the widely used Kidney Risk Failure Equation [KFRE] (Figs 2 & 3D). Performance for CKD prediction was comparable in Black individuals in the ARIC and UKBB cohorts (Table 3). The methods for protein marker selection and model construction and evaluation are thoughtful and complete. I have the following specific comments:

1. Strengths of this study are its methodological rigor, focus on a critical public health disparity and analysis of a population with preserved kidney function that is high-risk for progression, a group who would benefit from the availability of a prognostic tool.

2. The metrics show the algorithm performs well to identify individuals who will progress at the composite outcome, no matter baseline eGFR (Fig2A & SuppFig5A). However, I am less convinced that the APRS is APOL1-specific. Rather it appears to be an excellent tool for CKD prognostication. Its performance is quite similar to CRIC score (PMID: 37816758). The CRIC score was derived from a cohort with more advanced kidney failure and undoubtedly included participants with APOLHR genotype based on earlier publications from this group. This paper shows the CRIC score also works well in individuals with preserved eGFRs. The APRS has an advantage in that it only requires quantification of 9 vs 65 proteins and may be easier to move into the clinic. Do the CRIC score proteins overlap with the APRS protein markers?

The KFRE is widely used since it can be easily used with routine clinical and demographic data. Both the APRS and CRIC score significantly improve on the KFRE perhaps justifying additional expense from better discrimination as more CKD therapies move through the developmental pipeline.

The hazard ratios in Supplementary Table 3 are similar for APOL-HR and APOL-LR participants in the PMBB, although the APOL1 X APRS interaction HR suggests APOL1 has a significant contribution. This performance support the contention that the APRS has utility as a CKD predictor but will perform well in APOL1-HR individuals.

3. Supplementary Table 2 shows the APOL-HR training and testing cohort has a prevalence of ~70% for diabetes (DM). Table 2., study participant baseline characteristics, reports DM prevalence of ~30%. Please explain. In addition, the high prevalence of DM could further confound the interpretation that the prognostic tool is for APOL1 kidney disease. Did the investigators remove the DM participants for a sensitivity analysis

4. Supplemental Table 4: The biopsy source and histologic score method are not in the manuscript.

5. The recent publication from H3 Africa shows tremendous heterogeneity APOL1 prevalence suggesting that genetic background may modify APOL1 effect. DID the investigators perform a PCA ancestry analysis of the 3 cohorts used in this study?

Reviewer #2

(Remarks to the Author)

A large number of individuals of African ancestry carry APOL1-high-risk genotypes that increase their risk of kidney failure, but current tools cannot accurately predict progression before chronic kidney disease (CKD) develops. The authors developed and validated a nine-protein plasma-based APOL1 Proteomic Risk Score (APRS) that predicts composite kidney outcomes in high-risk APOL1 individuals with preserved kidney function. APRS significantly outperforms existing clinical and genetic risk prediction models, correlates with kidney tissue fibrosis, and may provide mechanistic insight into subclinical injury. This APRS can potentially enable earlier identification of individuals at risk, and may improve clinical trial design for HR APOL1 genotype carriers. However, the present study doesn't provide sufficient supporting evidence for the clinical validity, scalability, cost-effectiveness, or clinical utility to eventually reduce health disparities in APOL1-associated kidney failure. I have the following concerns about the study design, methods, results, and interpretation.

- The study population is not clearly defined and justified. Seems eGFR level at baseline was used as a convenient cut-off. However, other markers or clinical conditions (e.g., UACR) were not considered. The research question focused on "develop kidney failure before CKD". However, current definition (eGFR>60 at baseline) doesn't support such a population. I would expect many of the study participant will develop CKD very soon, especially those with borderline eGFR and fast eGFR decline. With 10-year follow-up, the study population may not be relevant to the main research question.
- I have concerns regarding the study's main findings and conclusions, which rely entirely on commercial proteomics platforms. While these platforms are suitable for exploratory research, they are known to suffer from limitations in annotation accuracy, quantitation reliability, and interpretability. For clinical validity and utility, it is essential to validate protein biomarkers using absolute concentration measurements. Additionally, the high cost and specialized infrastructure required for these platforms restrict their accessibility and scalability, particularly in low-resource settings. As a result, the proposed protein score may inadvertently exacerbate existing health disparities among at-risk populations, and the study should be reframed as a discovery effort with a clear discussion of its limitations in future clinical and translational applications.
- Proteomic methods of the PMBB need to be included in the main manuscript.
- The primary outcomes should be clearly justified. "A composite of kidney events (long-term dialysis, ESKD diagnosis, kidney transplantation, or a $\geq 40\%$ decline in eGFR from baseline) and all-cause mortality". Why would the all-cause mortality in a 10-year follow-up study as part of the outcome of kidney failure? Furthermore, mortality was also described as a competing risk (line 106-107). Mortality cannot be used the outcome and competing risk in the same model.
- How did the authors calculate the PRS using whole exome sequencing data? WES platform only covers a very small proportion of the genome and cannot support the PRS calculation, especially the genome-wide versions.
- Figure 1: "Total follow-up: 16.5 years" is misleading since the present study censored follow-up up to 10 years.
- Table 2: in the test group, ARIC and UKBB should be similarly compared (eGFR cut-off) as the PMBB cohort. There is a typo in line 493, "ACRI" should be "ARIC".
- Clarification in technical details would be helpful for reproducibility. For example, how did the eight-fold CV was implemented given the training-testing sampling? The authors "repeatedly sampled" subsets of proteins. It is unclear why and how this approach was implemented. How many protein panels have been evaluated using this approach?
- Age difference in AUC is very strong (higher in age >50, Fig 5C). It is critical to consider such difference in the application of the Proteomic risk score.
- Supp Table 3: Does the proportion of HR- vs LR-APOL1 genotypic groups reflect the distribution in PMBB? The APOL1 HR genotype has much lower frequency in African ancestry populations (e.g., ARIC and UKBB in supp Table 4) than the numbers from PMBB. What would be the impact of the sampling bias on the interaction results and the interpretation?
- It would be helpful to report the other measured proteins strongly correlated with identified proteins. Other potential protein panels can be developed based on correlated markers with similar performance.
- Additional covariates need to be considered such as medication use (antihypertensive, antidiabetic et al.).

Reviewer #3

(Remarks to the Author)

The aim of the current study was to use proteomic analysis in large biobanks to establish biomarkers predictive of kidney failure in people with APOL1 risk genotypes since, as the authors argue, early and reliable identification of those at risk of kidney failure would have substantial clinical utility especially if targeted therapies for APOL1-associated kidney disease become available.

The study is well executed and key findings validated across multiple datasets. The main areas for improvement relate to the clarity of phenotypic definition and to specific (and in places quite bold) claims about the interpretability and mechanistic relevance value of the composite APRS biomarker they have developed.

Splitting the cohort into groups termed "Preserved GFR" and "CKD" based on eGFR cut-off of 60 ml/min is probably not the clearest nomenclature: within the APOL1 high risk genotype-eGFR >60 ml/min group the mean eGFR was 90.6 ml/min, suggesting that a substantial proportion of those in this group would be categorised as CKD stage 2 (eg those with albuminuria and eGFR <90 ml/min) so it is not quite correct to say that CKD is not clinically apparent (as they do on page 6) in this group. This distinction is important because it seems possible that they have developed a biomarker set that picks up

early CKD (which is a risk factor for progression to later stages of CKD) as opposed to predicting future CKD. While their model did include baseline eGFR and UACR it would strengthen the paper to see the performance of the APRS in those with no hypertension, hematuria, albuminuria or structural urinary tract disease and eGFR >90 ml/min, and also in those with eGFR between 60 and 90 ml/min with none of these other features of kidney disease since it is possible that the association observed with APRS could have been driven among those with early, as opposed to no clinically apparent, CKD. Doing this would better delineate the predictive value of APRS just among the group that really do not have evidence of CKD and would help support confidence in their assertion that “The clinical implications of APRS are substantial. First, APRS may enable earlier identification of high-risk individuals for intensified surveillance long before CKD is detected by standard measures.” Given that 62.4% of the “preserved GFR” group had hypertension, and over a quarter had albuminuria >30 mg/g, it remains to be established that APRS allows identification of patients at risk ‘long’ before standard measures.

A second area where I think the claims made could have been better supported by data was the statement in the discussion that the APRS “captures both shared pathways of CKD progression and APOL1-specific mechanisms” since what is reported is association without experimental or other evidence to support mechanism. For instance, it would seem possible that reduced nephron number could result in altered levels of blood analytes simply as a consequence of how they are removed by the kidney, rather than as a consequence of the analyte’s biologic function. In places, circumstantial evidence (eg: “Specific proteins point to biologically plausible pathways: MMP-7 as a marker of tubular injury and fibrosis; WFDC2 as a correlate of interstitial fibrosis and rapid decline; and LYZ as a mediator of fibroblast proliferation and tubular cell senescence”) is provided to support claims that these proteins play a role in APOL1-mediated nephropathy. While it may have been reasonable to present these data as biologically plausible hypotheses, in the manuscript they are presented as evidence of mechanistic insights which would only come from actually testing these hypotheses. So, while there may be important and revealing relationships with biology underlying some of the associations observed, the paper would be more rigorous if this possibility was discussed in more balanced terms and claims of ‘mechanistic insights’ substantially toned down.

The KFRE is known to perform poorly among individuals with primary renal diseases, as opposed to the predominantly diabetic and renovascular CKD cohorts in which it was developed. Comparison with APRS therefore requires cautious interpretation, particularly as APRS is being evaluated in the same clinical population from which it was derived. Given that KFRE performs suboptimally in those with high-risk APOL1 genotypes, the comparison risks being perceived as a somewhat artificial benchmark. Consequently, the finding that APRS outperforms KFRE adds relatively little to the paper’s overall strength. Furthermore, even if APRS performance is cross-validated internally and across related biobank cohorts, it remains a biomarker developed within a similar population structure and proteomic platform. So outperforming KFRE here does not prove superior real-world predictive ability – only that APRS better fits this dataset. This consideration may also prompt the authors to reconsider the wording of their claims of clinical applicability.

The apparent stronger performance of APRS among APOL1 high-risk individuals might reflect higher event rates rather than greater biological specificity. If so, APRS could be acting as a marker of early CKD rather than as a genotype-specific prognostic tool. Exploring this possibility would clarify whether APRS captures general CKD biology or something specific to APOL1-mediated disease, which is relevant to the issue of mechanism. Perhaps it would be of value to test APRS predictive performance among individuals in the highest decile of genome-wide CKD polygenic risk but without clinical CKD, to assess broader generalisability?

The authors propose that a low risk APRS could provide reassurance to patients. I think this idea could be better explored with a heat map or risk-surface plot showing risk of outcome over seven or ten years in the setting of different combinations of APRS and conventional (ie clinical) risk factors (hypertension, eGFR, albuminuria). This would allow readers to appreciate how APRS might be interpreted clinically. However, I would advise extreme caution before making assertions of clinical utility of a potential biomarker such as this since, for individual patient-level decision making the statistical relationships seen in large real world cohorts have to be very strong indeed: knowing one is qualitatively at lower risk of an outcome is of little use if the magnitude of risk reduction is tiny and could result in false reassurance (and reduced eg blood pressure surveillance) that actually causes inadvertent harm.

In summary, this is an ambitious and technically sophisticated study that uses large-scale biobanks to address an important clinical question. The manuscript would benefit from more cautious language regarding mechanistic inference and clinical applicability, and from greater clarity around the phenotypic definitions used to derive the model. Addressing these points would ensure that the work’s substantial strengths and potential to yield clinically impactful future developments relevant to the growing field of APOL1-targeted therapy are conveyed more precisely.

Version 1:

Reviewer comments:

Reviewer #1

(Remarks to the Author)

No additional comments. The authors have adequately addressed the issues raised in the review.

Reviewer #3

(Remarks to the Author)

The authors have addressed all the issues I raised previously.

Reviewer #1 (Remarks to the Author):

Individuals with APOL1 high risk genotypes for kidney disease have an approximately 20% likelihood of developing advanced chronic kidney disease (CKD). To date, clinical actionable prognostication tools to identify the individuals with greatest risk have not been developed. Such algorithms are critical to effective management and counseling of individuals with the APOL1 risk genotype. In addition, such a tool would accelerate clinical trials as the pipeline for targeted APOL1 kidney disease therapies grows. It also fills a significant need to address, in part, the excess risk for advanced CKD in Black people worldwide.

These authors derived an APOL1 Proteomic Risk Score (APRS) using a training cohort of available individuals (80%) with high-risk APOL1 genotypes (APOL1-HR) in the PMBB, tested it in the remaining 20% and validated the tool in 2 additional cohorts of Black individuals in ARIC and UKBB with available APOL1 genotypes. The APRS had excellent discrimination and excellent risk stratification between the lowest and highest outcome quintiles and was the top performer in a clinical decision analysis with other prognostic algorithms including the widely used Kidney Risk Failure Equation [KFRE] (Figs 2 & 3D). Performance for CKD prediction was comparable in Black individuals in the ARIC and UKBB cohorts (Table 3). The methods for protein marker selection and model construction and evaluation are thoughtful and complete. I have the following specific comments:

1. Strengths of this study are its methodological rigor, focus on a critical public health disparity and analysis of a population with preserved kidney function that is high-risk for progression, a group who would benefit from the availability of a prognostic tool.

Response: We appreciate the reviewer's positive assessment of the study's rigorous methodology and clear focus. This encouragement is valuable; our work aimed to build on prior proteomic studies by focusing specifically on APOL1 high-risk individuals with preserved kidney function.

2. The metrics show the algorithm performs well to identify individuals who will progress at the composite outcome, no matter baseline eGFR (Fig2A & SuppFig5A). However, I am less convinced that the APRS is APOL1-specific. Rather it appears to be an excellent tool for CKD prognostication. Its performance is quite similar to CRIC score (PMID: 37816758). The CRIC score was derived from a cohort with more advanced kidney failure and undoubtedly included participants with APOLHR genotype based on earlier publications from this group. This paper shows the CRIC score also works well in individuals with preserved eGFRs. The APRS has an advantage in that it only requires quantification of 9 vs 65 proteins and may be easier to move into the clinic. Do the CRIC score proteins overlap with the APRS protein markers?

Response: Thank you.

1. The CRIC panel is very large (65 proteins) limiting its clinical implementation.

2. It was developed in patients with advanced CKD (<60, mean eGFR of 30) and it only performs well in patients with advanced CKD. In this subgroup the KFRE variables also perform well, further limiting the clinical need for such markers. A comprehensive comparison of all markers from both CRIC and PMBB reveals hazard ratios for our selected proteins are systematically higher in participants with preserved kidney function (eGFR ≥ 60 mL/min/1.73m²) compared to those with established CKD (eGFR <60). This suggests that these markers may be particularly informative in the preserved eGFR population and highlights the biological differences between these two groups. We found that only 3 proteins overlap with the markers derived from CRIC. This table has been included in the supplementary materials.

Comparison of Protein Hazard Ratios Between CRIC and PMBB Cohorts and Overlap

id	Target	Hazard Ratios per log ₂ (95%CI) adjusted for eGFR				Include in APRS
		CRIC (N. 65) eGFR<60	PMBB APOL1 high-risk			
			all	eGFR ≥ 60	eGFR<60	
seq.11388.75	WFDC2	3.54 (2.98,4.22)	3.39(2.79-4.13)	3.73(2.89-4.81)	2.77(2.06-3.72)	Yes
seq.2789.26	MMP7	1.1 (1.06,1.14)	1.69(1.47-1.93)	1.86(1.51-2.30)	1.51(1.27-1.80)	Yes
seq.8841.65	CILP2	0.57 (0.5,0.64)	0.47(0.40-0.55)	0.41(0.33-0.52)	0.53(0.42-0.66)	Yes
seq.4297.62	SPON1		4.60(3.54-5.97)	4.87(3.43-6.92)	4.13(2.80-6.11)	Yes
seq.4920.10	LYZ		3.98(3.20-4.95)	4.28(3.22-5.68)	3.22(2.27-4.57)	Yes
seq.19555.1	SUMO2		2.85(2.42-3.36)	3.03(2.44-3.78)	2.43(1.90-3.11)	Yes
seq.6036.78	EPHA10		2.68(2.26-3.19)	2.85(2.26-3.61)	2.27(1.76-2.93)	Yes
seq.15304.1	REG3A		2.13(1.88-2.42)	2.36(1.99-2.81)	1.80(1.50-2.18)	Yes
seq.7655.11	NPPB		1.47(1.37-1.58)	1.68(1.52-1.86)	1.29(1.17-1.42)	Yes
seq.9234.8	TWSG1	4.95 (3.87,6.32)	9.22(6.31-13.46)	8.77(5.15-14.93)	8.46(4.74-15.1)	
seq.15453.3	AMBP	4.55 (3.61,5.74)	8.13(5.38-12.29)	7.87(4.29-14.43)	6.59(3.68-11.79)	
seq.6245.4	NECTIN2	4.46 (3.51,5.66)	3.63(2.41-5.46)	1.81(0.90-3.64)	4.35(2.44-7.74)	
seq.3438.10	FSTL3	4.26 (3.38,5.36)	3.89(3.10-4.86)	3.97(2.90-5.43)	3.42(2.43-4.81)	
seq.15472.16	LRP11	4.15 (3.32,5.18)	4.22(3.13-5.69)	4.21(2.70-6.55)	3.59(2.36-5.44)	
seq.8005.1	MXRA7	3.45 (2.93,4.07)	2.99(2.33-3.85)	3.11(2.10-4.61)	2.62(1.86-3.68)	
seq.5349.69	DLL1	3.37 (2.70,4.21)	3.64(2.73-4.85)	3.77(2.51-5.66)	3.16(2.09-4.78)	
seq.19575.4	EPOR	3.24 (2.77,3.77)	3.57(2.59-4.91)	3.68(2.28-5.94)	3.26(2.10-5.06)	
seq.19130.81	SERPINB8	3.19 (2.68,3.80)	2.71(2.17-3.39)	2.57(1.86-3.53)	2.62(1.89-3.62)	
seq.8587.21	SPINK14	3.06 (2.57,3.63)	2.53(1.82-3.51)	2.13(1.23-3.71)	2.34(1.53-3.58)	
seq.9468.8	LMAN2	2.92 (2.38,3.58)	5.81(4.39-7.69)	8.78(5.93-13.02)	3.76(2.57-5.49)	
seq.8480.29	EFEMP1	2.88 (2.37,3.49)	3.56(2.81-4.51)	4.81(3.49-6.63)	2.48(1.76-3.48)	
seq.5452.71	ASGR1	2.84 (2.41,3.36)	1.99(1.60-2.46)	1.66(1.15-2.42)	1.9(1.44-2.5)	
seq.15585.304	FBLN5	2.82 (2.30,3.46)	1.69(1.36-2.11)	1.90(1.41-2.57)	1.44(1.04-1.99)	
seq.14136.234	CD93	2.78 (2.24,3.45)	3.72(2.81-4.93)	3.83(2.56-5.73)	3.17(2.13-4.71)	
seq.3738.54	CSF1	2.66 (2.25,3.15)	1.75(1.28-2.39)	2.05(1.33-3.16)	1.51(0.96-2.38)	
seq.2997.8	JAM2	2.65 (2.15,3.28)	5.27(3.75-7.40)	6.32(3.61-11.05)	4.03(2.59-6.28)	
seq.9316.67	WFDC1	2.54 (2.14,3.02)	3.27(2.65-4.03)	3.75(2.76-5.08)	2.63(1.97-3.53)	
seq.2944.66	NBL1	2.52 (2.24,2.84)	3.62(2.81-4.67)	4.36(3.01-6.32)	2.97(2.11-4.19)	
seq.12488.9	MCTS1	2.49 (2.18,2.84)	1.90(1.61-2.24)	1.83(1.42-2.37)	1.77(1.43-2.2)	
seq.15640.54	TAGLN	2.37 (2.02,2.79)	2.84(2.43-3.31)	3.16(2.58-3.86)	2.36(1.86-2.98)	
seq.9021.1	HAVCR1	2.31 (2.15,2.48)	1.61(1.43-1.81)	1.93(1.63-2.30)	1.40(1.20-1.64)	
seq.12008.3	CD7	2.24 (1.96,2.56)	1.95(1.61-2.36)	1.61(1.19-2.16)	1.99(1.52-2.61)	
seq.2774.10	IL16	2.16 (1.83,2.56)	2.10(1.67-2.65)	1.60(1.15-2.23)	2.41(1.73-3.34)	
seq.14615.46	KRTAP2-4	1.91 (1.65,2.20)	1.78(1.38-2.30)	1.28(0.89-1.85)	2.09(1.46-2.98)	
seq.16322.10	MZB1	1.86 (1.67,2.08)	1.96(1.63-2.37)	2.35(1.76-3.13)	1.57(1.22-2.01)	
seq.2765.4	GDF11	1.85 (1.61,2.11)	0.90(0.70-1.15)	0.85(0.61-1.20)	1.01(0.71-1.45)	
seq.18841.1	SERPINB13	1.83 (1.62,2.06)	1.23(1.03-1.48)	1.05(0.80-1.36)	1.30(1.01-1.67)	
seq.7970.315	ART3	1.63 (1.45,1.84)	1.92(1.58-2.33)	1.63(1.19-2.22)	1.95(1.52-2.51)	
seq.4493.92	IL11	1.56 (1.24,1.97)	1.41(0.94-2.10)	1.84(1.05-3.22)	1.30(0.72-2.32)	
seq.5954.62	PTH	1.48 (1.36,1.61)	1.84(1.60-2.13)	1.90(1.52-2.38)	1.68(1.40-2.03)	

seq.6247.9	SIRPB1	1.43 (1.31,1.55)	1.92(1.63-2.28)	1.74(1.34-2.25)	1.86(1.49-2.34)	
seq.8219.14	ZG16	1.32 (1.21,1.43)	1.02(0.99-1.06)	1.01(0.96-1.07)	1.02(0.97-1.08)	
seq.5852.6	S100A12	1.29 (1.18,1.40)	1.44(1.24-1.68)	1.57(1.28-1.93)	1.42(1.14-1.77)	
seq.14175.78	SCP2D1	1.27 (1.19,1.35)	1.53(1.34-1.75)	1.73(1.40-2.14)	1.29(1.08-1.54)	
seq.7245.2	CELF2	1.21 (1.05,1.39)	1.12(0.98-1.28)	1.22(1.02-1.47)	1.09(0.90-1.32)	
seq.5090.49	LILRB1	0.91 (0.88,0.95)	1.00(0.9-1.12)	1.00(0.86-1.17)	1.00(0.86-1.16)	
seq.4914.10	CGA	0.83 (0.79,0.87)	0.91(0.85-0.97)	0.94(0.85-1.03)	0.88(0.80-0.96)	
seq.9322.15	RCN1	0.82 (0.75,0.89)	0.90(0.81-1.00)	0.91(0.79-1.05)	0.92(0.79-1.07)	
seq.11377.19	ADH7	0.81 (0.71,0.92)	0.70(0.55-0.89)	0.73(0.52-1.03)	0.70(0.49-0.98)	
seq.18179.56	SMS	0.8 (0.58,1.10)	0.38(0.21-0.69)	0.38(0.17-0.88)	0.32(0.13-0.78)	
seq.9185.15	TFF1	0.77 (0.71,0.82)	0.93(0.82-1.06)	0.98(0.82-1.19)	0.95(0.80-1.13)	
seq.9416.77	CPM	0.77 (0.68,0.87)	0.92(0.67-1.26)	0.88(0.57-1.34)	1.02(0.63-1.65)	
seq.12630.8	ARFIP2	0.75 (0.66,0.86)	1.08(0.83-1.41)	0.82(0.57-1.18)	1.50(1.01-2.23)	
seq.5632.6	CRTAC1	0.73 (0.66,0.81)	0.96(0.76-1.21)	0.99(0.70-1.39)	1.01(0.72-1.40)	
seq.5491.12	SPOCK2	0.72 (0.62,0.85)	0.62(0.50-0.77)	0.79(0.58-1.07)	0.55(0.40-0.77)	
seq.2961.1	PROC	0.68 (0.56,0.82)	0.26(0.18-0.37)	0.26(0.16-0.42)	0.28(0.16-0.50)	
seq.10521.10	MXRA8	0.67 (0.55,0.81)	1.92(1.39-2.65)	1.37(0.83-2.25)	2.23(1.47-3.38)	
seq.18274.2	SERF2	0.64 (0.49,0.83)	0.19(0.11-0.33)	0.11(0.05-0.27)	0.26(0.12-0.55)	
seq.14133.93	IL1R2	0.62 (0.53,0.73)	0.65(0.49-0.86)	0.74(0.50-1.09)	0.62(0.41-0.94)	
seq.10833.64	HHIP	0.61 (0.50,0.74)	1.00(0.73-1.35)	1.03(0.67-1.59)	1.08(0.70-1.65)	
seq.5091.28	LILRB2	0.60 (0.51,0.71)	1.37(0.95-1.96)	1.77(1.07-2.94)	1.15(0.69-1.92)	
seq.17495.141	SIRT3	0.59 (0.49,0.73)	0.54(0.43-0.69)	0.55(0.39-0.75)	0.53(0.37-0.77)	
seq.2706.69	SERPINA7	0.58 (0.45,0.76)	0.41(0.26-0.63)	0.44(0.23-0.81)	0.37(0.20-0.68)	
seq.9583.17	RNF24	0.53 (0.41,0.67)	0.65(0.43-0.99)	0.47(0.26-0.85)	0.79(0.44-1.43)	
seq.5107.7	NOTCH1	0.53 (0.37,0.74)	0.57(0.32-1.03)	0.40(0.17-0.92)	0.78(0.34-1.76)	
seq.19584.33	FGF9	0.50 (0.38,0.67)	0.26(0.17-0.40)	0.31(0.18-0.56)	0.26(0.13-0.51)	
seq.13554.78	REPIN1	0.49 (0.33,0.73)	0.33(0.18-0.58)	0.28(0.12-0.68)	0.36(0.16-0.78)	
seq.18896.23	HS6ST3	0.46 (0.36,0.60)	0.71(0.47-1.07)	0.59(0.33-1.06)	0.88(0.49-1.58)	
seq.5735.54	C1GALT1C1	0.45 (0.37,0.54)	0.18(0.10-0.34)	0.14(0.06-0.34)	0.30(0.12-0.75)	
seq.18380.78	ALB	0.40 (0.33,0.49)	2.04(1.26-3.30)	2.92(1.42-6.00)	1.41(0.74-2.69)	
seq.5359.65	PIM1	0.27 (0.20,0.37)	0.38(0.22-0.65)	0.51(0.25-1.04)	0.31(0.14-0.69)	

3. APRS vs CRIC score comparison. Concerns that the APOL1 APRS may not be fully APOL1-specific in individuals with eGFR ≥ 60 are valid and do reflect underlying biology. CKD progression involves shared pathways, therefore, complete APOL1 specificity may be unattainable. To evaluate overlap between general CKD markers and APOL1-related risk, we compared protein hazard ratios from the CRIC cohort with those from APOL1 high-risk participants in PMBB, stratified by baseline eGFR (<60 vs ≥ 60 mL/min/1.73 m²). Across 65 proteins, hazard ratios were more strongly correlated in the eGFR <60 group ($r = 0.86$) than in the eGFR ≥ 60 group ($r = 0.76$). This suggests shared CKD progression pathways in both groups, but also meaningful differences in risk signatures for APOL1 high-risk individuals with preserved kidney function. These differences support developing new markers specifically for the eGFR ≥ 60 population (such as this study), where existing tools perform less well.

Comparison of hazard ratios for protein markers between CRIC and APOL1 high-risk PMBB cohorts. Hazard ratios (per \log_2 protein level) from the CRIC proteomic score are plotted against those from PMBB participants with APOL1 high-risk genotypes, stratified by baseline eGFR (<60 or ≥ 60 mL/min/1.73m²). Each point represents an individual protein marker (n=65). Pearson correlation coefficients (r) are shown for each subgroup.

The KFRE is widely used since it can be easily used with routine clinical and demographic data. Both the APRS and CRIC score significantly improve on the KFRE perhaps justifying additional expense from better discrimination as more CKD therapies move through the developmental pipeline.

Thank you. KFRE performs well for patients who already have CKD. It works better for people who also have proteinuria. KFRE does not perform well for patients with GFR ≥ 60 ml/min. KFRE includes important clinical variables therefore we included them in our APRS.

The hazard ratios in Supplementary Table 3 are similar for APOL-HR and APOL-LR participants in the PMBB, although the APOL1 X APRS interaction HR suggests APOL1 has a significant contribution. This performance support the contention that the APRS has utility as a CKD predictor but will perform well in APOL1-HR individuals.

Response: We appreciate the reviewer’s observation. Our goal was to develop a small clinically implementable biomarker panel that predicts disease in APOL1 high-risk people prior to the development of CKD. Based on this reasoning, we chose to optimize the model within the APOL1 high-risk population, with the expectation that a predictor capturing the key molecular signals in this group would also reflect global markers of progression relevant across genotypes. This consideration is particularly important because the APOL1 high-risk genotype is associated with a wide and heterogeneous range of kidney disease phenotypes, including FSGS, HIV-associated nephropathy, and multiple patterns of chronic or progressive injury (PMID: 20647424, 24206458, 21997396,

39448372). The diversity of these manifestations underscores the need for predictors that are robust to biological heterogeneity and capable of generalizing across different pathological trajectories.

Interaction analysis corroborates this: The APOL1 × APRS interaction hazard ratio of 1.35 (P=0.005) indicates that for each unit increase in APRS, APOL1 high-risk individuals experience a 35% risk increment beyond predicted by the proteomic signature alone, demonstrating that APOL1 RV fundamentally modifies the prognostic weight of these circulating biomarkers. This genotype-dependent effect size translates into significant differences in discriminative performance, with APRS achieving 8–10% higher time-averaged AUCs in APOL1 high-risk versus low-risk populations across independent cohorts (Extended Data Figure 5), confirming substantive enhancement of predictive accuracy in the target population.

Furthermore, the marker panel was optimized to a clinically actionable number (<10) predict both at high and low eGFR range and within different subgroups as detailed above. Therefore, while APRS demonstrates utility as a general CKD predictor, its calibration, discriminative capacity, and clinical reliability are substantially amplified in APOL1 high-risk carriers, validating its specific application for risk stratification in this disproportionately affected population, while concurrently acknowledging its more modest performance characteristics among low-risk individuals.

3. Supplementary Table 2 shows the APOL-HR training and testing cohort has a prevalence of ~70% for diabetes (DM). Table 2., study participant baseline characteristics, reports DM prevalence of ~30%. Please explain. In addition, the high prevalence of DM could further confound the interpretation that the prognostic tool is for APOL1 kidney disease. Did the investigators remove the DM participants for a sensitivity analysis

Response: We apologize for any confusion. The discrepancy in diabetes prevalence between Supplementary Table 2 and Table 2 was a typo. The correct prevalence of diabetes in the APOL1 high-risk training and testing cohort is approximately 30%, not 70%. We have corrected this error in the revision.

Regarding the potential confounding effect of diabetes on the interpretation of the prognostic tool, we conducted detailed subgroup analyses. The specific data and results are presented in Extended Data Figure 5, which provided forest plots of hazard ratios and time-averaged AUC across subgroups in both APOL1 high-risk and low-risk individuals, stratified by diabetes status, hypertension status, eGFR categories, and UACR categories. These analyses, which are consistent with our response to Reviewer #3's second question, demonstrate that the APRS retains predictive discrimination across various subgroups, including those with and without diabetes. This supports the robustness and generalizability of our findings.

4. Supplemental Table 4: The biopsy source and histologic score method are not in the manuscript.

Response: Added as suggested. In brief, kidney tissues (n=474) from an independent surgical nephrectomy cohort were de-identified, formalin-fixed, paraffin-embedded, PAS-stained, imaged (Aperio), and scored by a renal pathologist. RNA was extracted (RNeasy mini kit), libraries prepared (NEBNext Ultra II) and sequenced (Illumina NovaSeq 6000). Data processing included Trim-galore trimming, STAR alignment (hg19), and RSEM TPM quantification. IRB approval was exempt. These methods and results are now included in the main text.

5. The recent publication from H3 Africa shows tremendous heterogeneity APOL1 prevalence suggesting that genetic background may modify APOL1 effect. DID the investigators perform a PCA ancestry analysis of the 3 cohorts used in this study?

Response: We performed PCA ancestry analysis on PMBB, UKBB and ARIC cohorts used in this study. Our results showed that the APOL1 high-risk genotype is more prevalent among individuals of African ancestry. We attempted to incorporate polygenic risk scores and principal components (PC1 to PC5) into our model to account for genetic background. However, the performance of the model did not significantly improve, as the tAUC was 86.5 (3.9) without genetic variables and 86.4 (4.1) with genetic variables included. This negligible difference is well within sampling variability and not statistically significant. While this lack of predictive gain does not rule out biologically important effects of ancestry, we opted to exclude these variables to maintain model parsimony and focus on optimizing predictive performance.

For the APOL1 high-risk group, we recognize that established genetic modifiers such as the N264K variant (PMID: 38036523) and APOL3 (PMID: 39163132) may influence prognosis. We explored incorporating these modifiers into our model. Our analysis, including coefficient paths for selected features across different alpha values, indicated that both N264K and APOL3 are associated with outcomes. However, in the penalized elastic net model, these variables converged to zero or were eliminated earlier than even gender, suggesting they did not add substantial predictive value in this context.

Given the challenges of widespread genetic sequencing in clinical practice, we aimed to develop a simplified panel that could be more easily implemented. Therefore, we did not include these genetic modifiers in the final model. Our goal was to create a prognostic tool that is both clinically relevant and feasible for use in settings where genetic testing may not be readily available.

Principal-components analysis of genotype data, colored by inferred ancestry and by APOL1 risk status. Top: PMBB. Bottom: UKBB. Each point represents one study participant plotted on the first two genetic principal components (PC1, PC2) derived from LD-pruned autosomal variants. Left panel: points colored by genetically inferred continental ancestry (EUR = European, AFR = African, SAS = South Asian, EAS = East Asian, AMR = Admixed American, OTHER = other/unknown). Right panel: points colored by APOL1 risk status (LR = low-risk, HR = high-risk).

Principal-components analysis of ARIC Participants Projected onto HapMap3 Reference. Principal component analysis was performed using HapMap3 samples as the reference panel, and ARIC participants were subsequently projected onto the established axes. Each point represents an individual PC1 and PC2, colored according to reference population label

Coefficient paths for selected features across different alpha values. Coefficient trajectories are shown for key features in a penalized regression model as the regularization parameter alpha varies. The horizontal axis (log scale) represents increasing alpha, while the vertical axis shows the corresponding coefficient estimates. Each colored curve corresponds to one feature's coefficient path.

Reviewer #2 (Remarks to the Author):

A large number of individuals of African ancestry carry APOL1-high-risk genotypes that increase their risk of kidney failure, but current tools cannot accurately predict progression before chronic kidney disease (CKD) develops. The authors developed and validated a nine-protein plasma-based APOL1 Proteomic Risk Score (APRS) that predicts composite kidney outcomes in high-risk APOL1 individuals with preserved kidney function. APRS significantly outperforms existing clinical and genetic risk prediction models, correlates with kidney tissue fibrosis, and may provide mechanistic insight into subclinical injury. This APRS can potentially enable earlier identification of individuals at risk and may improve clinical trial design for HR APOL1 genotype carriers. However, the present study doesn't provide sufficient supporting evidence for clinical validity, scalability, cost-effectiveness, or clinical utility to eventually reduce health disparities in APOL1-associated kidney failure. I have the following concerns about the study design, methods, results, and interpretation.

- The study population is not clearly defined and justified. Seems eGFR level at baseline was used as a convenient cut-off. However, other markers or clinical conditions (e.g., UACR) were not considered. The research question focused on “develop kidney failure before CKD”. However, current definition (eGFR>60 at baseline) doesn't support such a population. I would expect many of the study participants will develop CKD very soon, especially those with borderline eGFR and fast eGFR decline. With 10-year follow-up, the study population may not be relevant to the main research question.

Response: Thank you for this important comment. We carefully checked all relevant text and clarified the terminology describing how participants were selected and analyzed, and we use the grouping by eGFR thresholds (eGFR ≥ 60 ml/min/1.73 m² and eGFR < 60 ml/min/1.73 m²) is now used instead of potentially misleading categorical labels. The eGFR ≥ 60 ml/min/1.73 m² threshold was selected for three specific reasons. First, this aligns with the population where existing clinical tools such as the KFRE perform poorly, creating a critical gap in risk prediction that our study directly addresses. Second, it encompasses the clinically relevant window during which APOL1 high-risk individuals transition from apparently normal kidney function to manifest CKD, which represents the precise period when early intervention could be most impactful. Third, this cutoff is consistent with the derivation populations of comparator tools, enabling fair evaluation of incremental value. We also expanded our evaluations across multiple clinical subgroups to ensure that the performance of the APRS is robust in populations with different baseline risk profiles, i.e. eGFR ≥ 60 without albuminuria.

We agree that albuminuria measurement is critical for identification of patients with CKD. However, observation data indicate that 8-30% of the US population is tested for proteinuria resulting in a very large fraction of the population with undiagnosed CKD. As a matter of fact, ARIC, which is one of the largest observation cohort studies in the US does not have proteinuria measurement at Visit 2 either. Several large organizations are working on fixing this critical problem. Moreover, UACR in routine clinical practice is highly variable and often unreliable

at low levels. Hydration status, timing of sample collection, recent exercise, menstruation, intercurrent illness, and specimen handling can all cause large fluctuations that do not reflect true biological change. This imprecision is especially pronounced in biobank settings, where measurements are obtained opportunistically rather than according to a standardized protocol. For this reason, using UACR as a core criterion for defining a “non-CKD” population would introduce substantial misclassification. It is also important to note that the transplant-free years and the broader analytic cohorts did not exclude individuals with UACR between 30 and 100 mg/g (PMID: 28742762). These participants remained within the eGFR greater than 60 group, which accurately represents the clinical spectrum observed in APOL1 high-risk individuals rather than an artificially restricted subset.

We acknowledge the reviewer's valid concern that some participants in the eGFR ≥ 60 group could have early CKD features. This limitation and heterogeneity reflect real-world clinical practice where most people do not have albuminuria measurement and present with varying stages of subclinical disease. To address this directly, we have performed extensive subgroup analyses (detailed in our response to Reviewer #3) demonstrating that APRS maintains robust predictive performance across multiple strictly defined subgroups, with time-averaged AUC consistently exceeding 80%. Notably, in participants with normal kidney function, defined as eGFR >90 ml/min/1.73 m² without hypertension, diabetes, or any history of kidney disease, APRS achieved strong discrimination with a hazard ratio of 2.1 per standard deviation ($p=0.003$). Equally robust performance was observed in the intermediate group (eGFR 60 to 90 ml/min/1.73m²) without albuminuria, hypertension, diabetes or any kidney disease. These comprehensive sensitivity analyses confirm that our findings are not driven solely by participants with borderline eGFR or early (non-detected) albuminuria, and that APRS provides predictive value even in individuals lacking any conventional evidence of CKD.

- I have concerns regarding the study's main findings and conclusions, which rely entirely on commercial proteomics platforms. While these platforms are suitable for exploratory research, they are known to suffer from limitations in annotation accuracy, quantitation reliability, and interpretability. For clinical validity and utility, it is essential to validate protein biomarkers using absolute concentration measurements. Additionally, the high cost and specialized infrastructure required for these platforms restrict their accessibility and scalability, particularly in low-resource settings. As a result, the proposed protein score may inadvertently exacerbate existing health disparities among at-risk populations, and the study should be reframed as a discovery effort with a clear discussion of its limitations in future clinical and translational applications.

Response: We appreciate the reviewer's concerns regarding proteomics platforms and clinical translation. The main strength of our work is that we have refined a discovery platform profiling thousands of proteins into a targeted panel of just nine aptamers. The development of a clinical test is our next goal; however, it is a difficult one which is clearly outside of the scope of the current manuscript. Whole O-link and SOMAscan have previously only had research only applications, both tests are now being transitioned for clinical application as well. For

example, the Joslin group recently developed a custom Olink platform measuring 21 kidney proteins that has been implemented for clinical research use at Renalytix AI Inc. laboratories using quality-assured protocols, with clear commercial pathways for clinical deployment (PMID: 41071620). Similarly, Del Campo et al. (PMID: 37704597) used Olink proximity extension assay technology to develop and validate a custom six-protein CSF panel for dementia with Lewy bodies across multiple independent cohorts including autopsy-confirmed cases, demonstrating robust analytical performance and clear translational potential. These precedents illustrate how targeted proteomic panels derived from discovery studies are moving toward clinical implementation. Aptamer-based assays offer particular advantages as they do not require recombinant protein production or animal immunization, enabling more rapid and economical development of custom clinical assays.

A nine-aptamer panel is inherently more scalable and cost-effective than antibody-based assays or full proteomic platforms. Aptamer technology does not require recombinant protein production or animal immunization, making it faster and more economical to develop custom assays. Regarding quantitation accuracy, we agree that relative fluorescence units require standardization for clinical application. However, this is a standard step in any biomarker translation process. We have added explicit text to the Limitations section noting that clinical implementation will require analytical validation against reference standards, assay calibration, and CLIA certification. Our multi-platform validation using SOMAscan and Olink already demonstrates strong transferability, which we consider a strength rather than a limitation. All nine SOMAmers were validated against purified proteins during SELEX development. We have added literature support (PMIDs) below, confirming these proteins as established endogenous circulating biomarkers for different disease and related pathological processes.

The concern about potential exacerbation of health disparities is also well taken. APOL1 high-risk genotypes are prevalent among individuals of African ancestry, who face the highest burden of kidney failure. Existing predictive tools perform poorly in this population. By focusing specifically on APOL1 high-risk carriers, the APRS provides a targeted solution aimed at reducing, rather than worsening, disparities. The nine markers can be measured on any validated platform, and we are actively exploring point-of-care aptamer technologies suitable for community-based screening. We have also revised the manuscript to reframe this study as a pivotal bridge from discovery to translation. The Abstract now states that APRS “provides a framework for early intervention,” rather than implying immediate clinical readiness. In the revised Discussion, we now outline a clear translational roadmap in text form, describing the necessary steps for clinical implementation.

Orthogonal Validation, Mass Spectrometry Verification, and Genetic Association References for APRS Protein Biomarkers*

SOMAmer	Protein	UniProt	Gene	Orthogonal Technology	Mass Spec Verification	Genetic Association Strategies
seq.11388.75	HE4	Q14508	WFDC2		30072576	35501419, 34857953, 30072576
seq.15304.1	PAP1	Q06141	REG3A			35501419, 34857953, 34648354
seq.19555.1	SUMO2	P61956	SUMO2			35501419
seq.2789.26	MMP-7	P09237	MMP7	31844116, 29167395, 35984888		30072576, 28240269, 34648354, 34857953, 35079000, 35501419
seq.4297.62	Spondin1	Q9HCB6	SPON1	27640094, 31737572, 29875488, 35984888		29875488, 30072576, 28240269, 34648354, 34857953, 35501419
seq.4920.10	Lysozyme	P61626	LYZ		30072576	30072576, 29875488, 28240269, 34648354, 34857953, 35078996, 35079000, 28031287, 28129359
seq.6036.78	EPHAA	Q5JZY3	EPHA10			
seq.7655.11	BNP	P16860	NPPB	31639705, 35984888, 29875488		35501419, 29875488, 30072576, 34648354, 34857953, 35079000
seq.8841.65	CILP2	Q8IUL8	CILP2		30072576	35501419, 34857953, 30072576

*PMIDs are shown in the table.

- Proteomic methods of the PMBB need to be included in the main manuscript.

Response: Added as suggested.

- The primary outcomes should be clearly justified. “A composite of kidney events (long-term dialysis, ESKD diagnosis, kidney transplantation, or a $\geq 40\%$ decline in eGFR from baseline) and all-cause mortality”. Why would the all-cause mortality in a 10-year follow-up study as part of the outcome of kidney failure? Furthermore, mortality was also described as a competing risk (line 106-107). Mortality cannot be used the outcome and competing risk in the same model.

Response: We appreciate this important methodological question regarding our composite endpoint definition. To directly address the reviewer's concern about model performance being driven by mortality rather than kidney outcomes, we have conducted additional analyses separating the outcome components, as shown in the following table. These results demonstrate that our APRS performs excellently for kidney specific outcomes alone. In the primary test cohort of APOL1 high risk individuals with preserved kidney function (eGFR ≥ 60), the APRS achieved a time averaged AUC of 88.1% for kidney events only, which is superior to the 86.5% observed for the composite endpoint. This pattern is consistent across in high-risk cohorts. These findings confirm that the model's strong performance is not reliant on capturing mortality risk but rather reflects genuine prediction of kidney disease progression. Furthermore, when comparing APRS to CRIC proteomic score, which was developed specifically for kidney event, our model shows superior or equivalent performance for kidney outcomes across all tested populations. The CRIC score achieved a kidney event AUC of 80.8% in our training set and 83.4% in our test set, both notably lower than APRS performance. This head-to-head comparison validates that our nine-protein signature captures kidney specific pathobiology effectively. The fact that APRS outperforms both the

KFRE and the CRIC score for kidney events in the critical preserved eGFR population underscores its unique value for early risk stratification.

Comparative Outcome-Specific Performance of Risk Scores

APOL1	eGFR	Group	N	Score	All event		Death		Kidney event	
					tAUC	events%	tAUC	events%	tAUC	events%
High risk	≥60	Training	680	APRS	86.7(1.4)	18.5	88.3(1.6)	7.2	86.0(2.7)	14.1
				CRIC	79.7(1.9)		76.6(4.2)		80.8(2.6)	
				KFRE	66.6(1.5)		65.4(1.5)		70.1(2.2)	
				PRS	48.0(2.9)		41.8(6.8)		52.8(1.3)	
	≥60	Test	171	APRS	86.5(3.9)	15.8	85.7(1.8)	7.6	88.1(1.6)	14
				CRIC	79.0(4.0)		79.2(3.3)		83.4(1.6)	
				KFRE	66.2(4.3)		46.6(3.3)		67.8(2.3)	
				PRS	58.5(5.2)		48.1(3.8)		48.4(4.6)	
	<60	Test	262	APRS	84.2(1.9)	55.3	85.3(2.1)	21.8	86.4(1.8)	47.7
				CRIC	83.2(1.7)		77.1(3.2)		86.1(1.2)	
				KFRE	82.4(2.1)		79.9(4.2)		86.4(1.6)	
				PRS	53.8(1.4)		56.8(1.4)		53.3(1.5)	
Low risk	Any	Test	912	APRS	73.1(2.1)	15.1	79.9(2.7)	7.7	69.1(1.6)	9.6
				CRIC	69.3(1.3)		71.6(1.0)		69.0(2.3)	
				KFRE	60.1(1.5)		53.9(2.2)		59.3(1.0)	
				PRS	50.5(1.1)		50.3(1.4)		46.8(1.3)	

To address the competing risk concern, we performed sensitivity analyses that excluded mortality and applied Fine–Gray competing risk models. The predictive performance of APRS was essentially unchanged: for kidney events alone, the time-averaged AUC was 87.32%, for mortality alone 82.64%, and for the composite outcome 86.49%. Under a competing risk framework, results were very similar to the composite event model (kidney events 88.1%, mortality 85.7%, composite outcome 86.5%). These analyses confirm that APRS can predict both kidney event and death, and our conclusions do not depend on the definition of the endpoint.

Risk of kidney event and mortality by APRS in APOL1 high-risk individuals

The graph illustrates the relationship between mortality (%) and kidney failure (%) across APRS, highlighting the upward trend of both outcomes with increasing APRS scores.

Our rationale for including all-cause mortality in the composite endpoint is based on clinical relevance and alignment with nephrology standards, which we can summarize in three key points. First, patients with CKD, even in early stages, face substantially higher mortality risk from cardiovascular and other complications than risk of progressing to kidney failure, and these outcomes frequently overlap. In our cohort, nearly 50% of individuals who died had already experienced a kidney event (dialysis, ESKD, transplant, or $\geq 40\%$ eGFR decline), demonstrating that death and kidney outcomes are intertwined manifestations of disease progression rather than mutually exclusive events. Multiple large studies and meta-analyses confirm that cardiovascular mortality in CKD exceeds ESRD risk, particularly in early and mid-stage disease, and ignoring death would produce overly optimistic kidney survival estimates while misclassifying the clinical trajectories of many patients.

Overlap of Kidney Events and Death in APOL1-high-risk with eGFR>60

Second, composite endpoints that combine kidney failure with all-cause mortality represent standard practice in nephrology, as they are widely used in clinical trials and observational studies and are explicitly endorsed by KDIGO guidelines for clinically meaningful risk prediction. Similar composite endpoints combining kidney failure and death have been widely adopted in major nephrology trials (PMID: 30990260), underscoring their acceptance as clinically meaningful outcomes. This methodological precedent reflects the total disease burden that matters to patients and physicians, who view mortality and kidney failure as equally important outcomes. Third, mortality increased statistical efficiency by raising the 10-year event rate to 18%, maximizing power in this high-risk population.

This approach also addresses a major limitation of existing prognostic models such as the KFRE, which have been criticized for ignoring mortality risk. Patients and physicians want to know not only the risk of dialysis but also the risk of dying before reaching dialysis, and our composite endpoint captures both kidney-specific risk and overall clinical risk, providing a more holistic and actionable assessment.

For these reasons, we believe the composite endpoint (kidney failure + all-cause mortality) is the most clinically meaningful and methodologically rigorous choice for this study. We have revised the Methods (Outcomes section) to explicitly clarify this rationale in the manuscript.

- How did the authors calculate the PRS using whole exome sequencing data? WES platform only covers a very small proportion of the genome and cannot support the PRS calculation, especially the genome-wide versions.

Response: We thank the reviewer for raising this important point. We agree that WES data alone provide limited genomic coverage and cannot directly support genome-wide PRS derivation. In this study, genome-wide PRS were not calculated using WES data. Instead, PRS were computed using imputed genotyping array data, while WES data were used for targeted analyses requiring high accuracy for rare variants. The PMBB provides both WES data and genotyping array data for most participants. The majority of individuals included in this analysis had both WES and imputed genotyping data available; PRS analyses were restricted to participants with imputed genotype data. WES was used specifically for APOL1 high-risk genotype determination due to its superior accuracy for rare variant calling. For PRS, we used imputed genotype data from PMBB Release 2.0, generated from the GSA genotyping array (Freeze 2.0). Genotypes were phased using Eagle2 and imputed using Minimac4 with the Michigan TOPMed r3 reference panel (GRCh38). The imputed dataset was filtered to retain high-confidence variants (average imputation $R^2 > 0.3$ or directly genotyped in either batch) and variants with minor allele frequency > 0.01 . PRS values were calculated in PLINK 2.0 using the overlapping set of variants between the published PRS weights and the PMBB-imputed genotypes.

- Figure 1: “Total follow-up: 16.5 years” is misleading since the present study censored follow-up up to 10 years.

Response: Changed as suggested.

- Table 2: in the test group, ARIC and UKBB should be similarly compared (eGFR cut-off) as the PMBB cohort. There is a typo in line 493, “ACRI” should be “ARIC”.

Response: Changed as suggested. We appreciate the reviewer's suggestion to present eGFR-stratified performance for ARIC and UKBB. Our original presentation of aggregate validation data was driven by limited statistical power in the reduced kidney function subgroups, which reflect the population-based nature of these cohorts. Specifically, UK Biobank contains only 9 APOL1 high-risk individuals with $eGFR < 60$, yielding an AUC of 91.1% (8.9%), while the $eGFR \geq 60$ group ($n=195$) shows a more stable estimate of 83.6% (5.9%). Similarly, ARIC includes just 18 participants with $eGFR < 60$ compared to 296 with $eGFR \geq 60$ (AUC 77.5%). The relatively low performance for ARIC with $eGFR \geq 60$ is likely due to the lack of UACR inclusion, but the performance of APRS remains substantially better than the CRIC score. These small sample sizes produce wide confidence intervals that limit interpretability, which is why we emphasized overall performance to demonstrate generalizability across diverse settings. Additionally, these stratifications appear unnecessarily complicated for the readers. However,

to address this concern transparently, we now provide these stratified results below. The central conclusion remains unchanged: APRS demonstrates robust discrimination across all validation cohorts, with its most clinically significant advantage emerging in the preserved function population where other model performs poorly and early intervention would be most impactful.

Ten years time-dependent AUC for predicting composite outcomes in different cohorts using various predictive models*

Group	APOL1	eGFR (ml/min/1.73m ²)	Cohort	N	events%	KFRE	CRIC	APRS
Training	High risk	≥60	PMBB	680	17.8	66.6(1.5)	79.7(1.9)	86.7(1.4)
		≥60	PMBB	171	18.7	66.2(4.3)	79.0(4.0)	86.5(3.9)
		<60	PMBB	262	55.3	82.4(2.1)	83.2(1.7)	84.2(1.9)
	High risk	≥60	ARIC	296	10.5	-	52.0(1.6)	77.5(1.1) **
			UKBB	195	5.4	67.3(6.8)	80.0(5.5)	81.6(5.9)
		Any	ARIC	314	13.7	-	55.8(1.9)	82.2(1.2)
Test	Low risk		UKBB	204	6.3	75.5(7.4)	80.4(5.1)	84.7(5.3)
			ARIC	1932	13.6	-	52.3(0.7)	74.2(0.8)
		≥60	PMBB	874	15.4	52.8(3.7)	72.6(3.8)	79.3(3.8)
	Low risk		UKBB	942	6.2	55.8(4.4)	75.6(5.1)	74.2(5.0)
			ARIC	2021	15.9	-	53.7(0.5)	78.5(1.2)
		Any	PMBB	995	14.9	60.1(1.5)	69.3(1.3)	73.1(2.1)
		UKBB	967	6.8	63.8(4.5)	78.1(4.6)	74.8(3.9)	

*This table presents the time-dependent area under the receiver operating characteristic curve values (standard deviation) for different predictive models across various populations and cohorts over a ten-year period. The table compares the performance of KFRE, CRIC score and APRS in predicting composite outcomes. *Urine albumin-to-creatinine ratio is not available for ARIC at Visit 2.

• Clarification in technical details would be helpful for reproducibility. For example, how did the eight-fold CV was implemented given the training-testing sampling? The authors “repeatedly sampled” subsets of proteins. It is unclear why and how this approach was implemented. How many protein panels have been evaluated using this approach?

Response: To ensure robust model evaluation and mitigate overfitting, we employed eight-fold cross-validation within the training cohort. Specifically, the training dataset was randomly divided into eight equal-sized folds. For each iteration of the cross-validation process, one-fold was used as the validation set, while the remaining seven folds were combined to form the training set. This procedure was repeated eight times, with each fold serving as the validation set once. The model performance was assessed by calculating the C-index for each validation fold, and the mean C-index across all eight folds was used to evaluate the overall model performance for a given protein panel and parameter set. Importantly, the test group was completely independent of the training process.

Marker Selection Stability Analysis

(A) Each bar represents a protein, with the height corresponding to the frequency at which the protein was selected. (B) Radial protein correlation network. APRS proteins are positioned on the outer ring ($R=0.8$) and ordered by hierarchical clustering. Peripheral proteins (dots) are placed by correlation strength (radial distance) and direction (angular position) relative to the core. Color represents regression coefficients from the survival model, with a diverging scale from negative to positive. (C) The cumulative distribution function plot shows the proportion of genes selected by threshold. The x-axis represents the selection frequency, and the y-axis represents the cumulative probability. (D) Coefficient paths for selected features across different alpha values. Coefficient trajectories are shown for key features in a penalized regression model as the regularization parameter alpha varies.

The rationale behind repeatedly sampling subsets of proteins was to systematically explore a broad range of protein combinations and identify the most predictive ones, given the complexity and potential redundancy in the proteomic data and the aim to capture diverse and potentially synergistic interactions among proteins that could

contribute to outcome prediction. This was implemented through the following sequential steps: (1) initial protein screening and candidate panel generation by identifying proteins showing significant univariate associations with the outcome (top 20%), then randomly sampling 40–90% of these proteins without replacement for each candidate panel; (2) elastic-net modeling with grid search, fitting an elastic-net Cox model for each sampled protein panel and conducting a comprehensive grid search across the mixing parameter α (ranging from 0.1 to 0.9 in 0.1 increments) and the regularization strength λ (spanning 100 logarithmically spaced values); (3) performance evaluation via eight-fold cross-validation using the mean C-index; large-scale combinatorial exploration of approximately one million unique candidate protein panels to thoroughly explore the vast combinatorial space of protein interactions and identify the optimal combination; (4) intermediate feature selection by stability, retaining proteins with a selection frequency of at least 30% across candidate panels to avoid multiple testing problems and reduce noise, ensuring only consistently predictive proteins were retained; and (5) final model construction and refinement using the intermediate protein set with another round of elastic-net–penalized Cox proportional-hazards modeling to eliminate highly correlated proteins and optimize model parameters based on the most promising candidates identified in the initial screening and evaluation process. These methods and results have been updated in the main text.

- Age difference in AUC is very strong (higher in age >50, Fig 5C). It is critical to consider such difference in the application of the Proteomic risk score.

Response: We thank the reviewer for highlighting this important consideration. Age-specific performance variation is indeed critical for clinical translation, which is precisely why we included age (along with sex, eGFR, and UACR) as an unpenalized covariate in our model to ensure age-adjusted risk estimation. We observed modest age-related differences in discriminative performance (tAUC ~0.87 for age ≤ 50 vs. ~0.82 for age >50) in the APOL1 high-risk with eGFR ≥ 60 cohort. This degree of variation is biologically plausible because older individuals carry greater comorbidity burden and competing mortality risks that can attenuate discrimination for kidney-specific outcomes, yet it remains clinically acceptable as both values substantially exceed existing tools (KFRE tAUC ~0.66). Hazard ratios remain robust across both age strata (hazard ratio >2 per SD, $p < 0.001$), confirming consistent prognostic value.

To facilitate age-appropriate clinical interpretation, we now provide a comprehensive risk table (Extended Data Figure 6) that stratifies 5- and 10-year absolute risk estimates by age category (≤ 50 vs. >50 years), APRS quantiles, and comorbidity status. This framework explicitly accounts for age-related differences in baseline risk, enabling clinicians to interpret APRS values within the proper age-specific context. We have also added explicit results in the revised manuscript emphasizing that while APRS maintains strong discrimination across all age groups, integration with age-specific risk thresholds will be essential for clinical implementation, consistent with standard practice for other validated risk scores.

• Supp Table 3: Does the proportion of HR- vs LR-APOL1 genotypic groups reflect the distribution in PMBB? The APOL1 HR genotype has much lower frequency in African ancestry populations (e.g., ARIC and UKBB in supp Table 4) than the numbers from PMBB. What would be the impact of the sampling bias on the interaction results and the interpretation?

Response: We appreciate this question about our genotype distribution. We included everyone with high-risk genotype from PMBB, which was the primary goal of our analysis. In the first batch, we used propensity score matching to select 700 APOL1 high-risk and 700 low-risk participants with balanced baseline characteristics. In the second batch, we included the newly identified APOL1 high-risk individuals (~400 additional participants) to maximize our target population, and ~200 more low-risk participants for comparator purposes. In this revision, we further performed propensity score matching to make the two groups more comparable (Supplementary information, study protocol and statistical analysis plan). Consequently, the final high-risk group (n=1,113) accurately reflects the true PMBB distribution, while the low-risk group is a carefully selected, matched subset not intended to represent population prevalence. Importantly, the APOL1 high-risk group in PMBB is representative and free of selection bias. The low-risk group can be used to compare relative biomarker performance (high-risk vs low-risk, given the groups are matched) the low-risk group has intentional selection bias as described. For this reason, this work is not focused on low-risk individuals or plans to make claims on this subgroup. However, the selection bias of the low-risk is mitigated by our robust external validation, where the validation cohorts (ARIC and UKBB) consist of healthier, population-based individuals without such sampling bias. In fact, this "biased" sampling design represents a strategic advantage: by demonstrating consistent APRS performance across both the enriched PMBB cohort and the more general population validation cohorts, we establish that APRS is robust and generalizable across diverse clinical settings.

This design aligns with our primary objective: developing a proteomic risk score specifically optimized for APOL1 high-risk individuals where clinical need is greatest. As Reviewer #1 noted, we do not deny APRS demonstrates utility as a general CKD predictor it retains predictive value in low-risk populations but its performance is substantially enhanced in high-risk individuals with consistent 8-10% higher time-averaged AUCs across all cohorts. The cohort matching was implemented precisely to create comparable groups for valid interaction testing while controlling for baseline confounding. The APOL1×APRS interaction (hazard ratio=1.31, P=0.005) appropriately weighted participants to account for this case-control structure, yielding unbiased effect estimates within each genotype stratum.

Baseline Characteristics of APOL1 High-Risk and Low-Risk Participants After Propensity Score Matching (PSM)

	All after PSM			eGFR \geq 60 after PSM			eGFR<60 after PSM		
	APOL1 high-risk	APOL1 low-risk	P	APOL1 high-risk	APOL1 low-risk	P	APOL1 high-risk	APOL1 low-risk	P
Number	1113	912		851	874		262	38	
Age - yr	51.22 \pm 15.67	50.13 \pm 14.12	0.1	49.22 \pm 15.07	49.9 \pm 14.03	0.33	57.73 \pm 15.84	55.41 \pm 15.38	0.4
Female - no. (%)	701(62.98%)	593(65.02%)	0.34	566(66.51%)	568(64.99%)	0.51	135(51.53%)	25(65.79%)	0.1
SBP - mmHg	129.09 \pm 18.3	129.48 \pm 17.47	0.63	128.54 \pm 17.79	129.3 \pm 17.41	0.37	130.87 \pm 19.81	133.55 \pm 18.5	0.43
DBP- mmHg	77.04 \pm 12.02	77.56 \pm 11.54	0.32	77.59 \pm 11.9	77.42 \pm 11.47	0.76	75.23 \pm 12.26	78.75 \pm 13.7	0.08
Body Mass Index - kg/m ²	31.95 \pm 7.8	32.31 \pm 7.85	0.31	32.27 \pm 7.84	32.29 \pm 7.83	0.95	30.93 \pm 7.58	32.71 \pm 8.39	0.18
Hemoglobin A1c (%)	6.59 \pm 1.81	6.39 \pm 1.38	0.11	6.57 \pm 1.81	6.39 \pm 1.39	0.19	6.66 \pm 1.81	6.37 \pm 0.86	0.63
Creatinine (mg/dL)	1.14 \pm 0.68	0.89 \pm 0.3	<0.01	0.88 \pm 0.21	0.86 \pm 0.18	0.07	1.99 \pm 0.95	1.68 \pm 0.87	0.06
Blood Urea Nitrogen (mg/dL)	16.73 \pm 10.59	13.19 \pm 6.09	<0.01	13.04 \pm 4.98	12.56 \pm 4.39	0.03	28.69 \pm 14.44	27.63 \pm 15.33	0.67
eGFR - ml/min/1.73m ²	78.26 \pm 27.96	89.94 \pm 19.02	<0.01	90.59 \pm 17.04	91.96 \pm 16.39	0.09	38.19 \pm 16.67	43.46 \pm 15.97	0.07
UACR (IQR) - mg/g	17.23(11-44)	15.31(12-24)	<0.01	17.23(11-31)	15.31(12-24)	0.05	31.67(15-124)	24.12(12-62)	0.26
30~299 - no. (%)	163(14.64)	149(16.34)		112(13.16)	138(15.79)		51(20.23)	11(28.95)	
\geq 300 - no. (%)	60(5.39)	28(3.07)		25(2.93)	23(2.63)		35(13.89)	5(13.16)	
Diagnostic group - no. (%)									
Hypertension	754(67.74%)	560(61.4%)	<0.01	531(62.4%)	532(60.87%)	0.51	223(85.11%)	28(73.68%)	0.08
Diabetes Mellitus	335(30.1%)	223(24.45%)	<0.01	237(27.85%)	211(24.14%)	0.08	98(37.4%)	12(31.58%)	0.49
Cardiovascular Disease	116(10.42%)	105(11.51%)	0.43	78(9.17%)	100(11.44%)	0.12	38(14.5%)	5(13.16%)	0.83
p.N264K	45(4.04%)	58(6.36%)	0.01	40(4.7%)	56(6.41%)	0.09	5(1.91%)	2(5.26%)	0.2
Event - no. (%)									
Composite Event	298(26.77%)	138(15.13%)	<0.01	153(17.98%)	135(15.45%)	0.16	145(55.34%)	3(7.89%)	<0.01
Deceased	119(10.69%)	70(7.68%)	0.02	62(7.29%)	68(7.78%)	0.7	57(21.76%)	2(5.26%)	0.02
Kidney event	245(22.01%)	88(9.65%)	<0.01	120(14.1%)	86(9.84%)	<0.01	125(47.71%)	2(5.26%)	<0.01
Follow-up time - yr	7.06 \pm 3.05	7.64 \pm 2.69	<0.01	7.18 \pm 2.89	7.64 \pm 2.68	<0.01	6.67 \pm 3.5	7.42 \pm 2.77	0.2
Time to event - yr	5.93 \pm 3.36	7.24 \pm 2.85	<0.01	6.51 \pm 3.07	7.24 \pm 2.86	<0.01	4.02 \pm 3.57	7.25 \pm 2.82	<0.01

Importantly, the genotype imbalance in our external validation cohorts where APOL1 high-risk prevalence is only 12% in ARIC and 17% in UKBB among African ancestry participants actually strengthens our study. APRS maintained robust discrimination across these population-based settings with vastly different high-risk/low-risk ratios and demographic structures, confirming that our findings are not artifacts of PMBB enrichment but reflect genuine biological differences. The reproducibility of the performance advantage in APOL1 high-risk individuals across all validation cohorts validates that APRS provides a targeted, biologically grounded solution for the population bearing the highest kidney disease burden, while remaining applicable across diverse clinical contexts.

- It would be helpful to report the other measured proteins strongly correlated with identified proteins. Other potential protein panels can be developed based on correlated markers with similar performance.

Response: Thank you very much for this thoughtful suggestion. We fully agree that correlation structure is informative for understanding biomarker relationships. In our analysis, we examined correlations between the APRS proteins and the broader proteomic dataset (Extended Data Figure 2: Radial protein correlation network showing core APRS proteins positioned on the outer ring and peripheral proteins plotted by correlation strength and direction) and found that several proteins do track closely with individual APRS markers. However, perfect substitutes are difficult to identify, particularly for CLIP2, NPPB, MMP7, and SPON1, where no other proteins reached correlations of ~ 0.8 or higher. More importantly, we believed that correlation strength does not directly translate to prognostic capability; many highly correlated proteins lacked independent predictive value when evaluated in multivariable modeling. As a result, correlation-based substitution or expansion does not reproduce the predictive performance of the APRS panel.

Radial protein correlation network.

APRS proteins are positioned on the outer ring ($R=0.8$) and ordered by hierarchical clustering. Peripheral proteins (dots) are placed by correlation strength (radial distance) and direction (angular position) relative to the core. Color represents regression coefficients from the survival model, with a diverging scale from negative to positive.

Our modeling strategy was intentionally designed to avoid redundancy and to identify markers with true independent value. This is why the elastic-net framework consistently prioritized the nine APRS proteins over their correlated partners. Correlated proteins can of course be biologically interesting, but they did not support a comparably robust or clinically scalable scoring system. To remain focused and avoid creating confusion with exploratory findings that do not improve model performance, we have summarized the correlation patterns at a high level in the revised text rather than providing exhaustive protein lists. We hope this addresses the reviewer’s question while keeping the emphasis on validated, independently informative markers.

- Additional covariates need to be considered such as medication use (antihypertensive, antidiabetic et al.).

Response: We appreciate this suggestion and have thoroughly evaluated the inclusion of medication use as covariates. As shown in our analysis, we compiled comprehensive data on all antihypertensive and antidiabetic medications. We performed coefficient path analysis for selected features across different alpha values, which demonstrated that all medication variables consistently converged to zero before the protein markers were eliminated from the model. While some medication variables remained in the model longer than others and a few persisted beyond some clinical variables, they were ultimately dropped by the elastic-net penalty, indicating they did not contribute independent predictive value.

Baseline Medication Use in APOL1 High-Risk and Low-Risk Participants

	no. (%)	APOL1 high-risk eGFR \geq 60	APOL1 low-risk eGFR \geq 60
ACE inhibitors		284(33.37%)	290(33.18%)
Angiotensin II receptor blockers		165(19.39%)	126(14.42%)
Beta-blockers		334(39.25%)	332(37.99%)
Calcium channel blockers		308(36.19%)	311(35.58%)
Thiazide diuretics		325(38.19%)	344(39.36%)
Potassium-sparing diuretics		81(9.52%)	85(9.73%)
Alpha-1 blockers		59(6.93%)	48(5.49%)
Alpha-2 agonists		46(5.41%)	53(6.06%)
Vasodilators		138(16.22%)	107(12.24%)
Insulin analogs		216(25.38%)	205(23.46%)
Stimulants		55(6.46%)	57(6.52%)
GLP-1 receptor agonists		29(3.41%)	26(2.97%)
Insulin		74(8.7%)	76(8.7%)
SGLT2 inhibitors		3(0.35%)	5(0.57%)
Sulfonylureas		74(8.7%)	65(7.44%)
DPP-4 inhibitors		33(3.88%)	24(2.75%)

Furthermore, we conducted sensitivity analyses adding medication variables as forced covariates to the final model and found no significant improvement in discrimination (tAUC change <0.02). This suggests that the proteomic markers capture the biological effects of these medications rather than the medications themselves being independent predictors. We acknowledge that medication adherence and dosing details were not captured,

which represents a limitation, but given that medication use is itself a marker of disease severity and the proteomic signature appears to capture treatment effects indirectly, we believe our approach provides a more parsimonious and clinically practical model.

Coefficient paths for selected features across different alpha values. Coefficient trajectories are shown for key features in a penalized regression model as the regularization parameter alpha varies. The horizontal axis (log scale) represents increasing alpha, while the vertical axis shows the corresponding coefficient estimates. Each colored curve corresponds to one feature's coefficient path.

Reviewer #3 (Remarks to the Author):

The aim of the current study was to use proteomic analysis in large biobanks to establish biomarkers predictive of kidney failure in people with APOL1 risk genotypes since, as the authors argue, early and reliable identification of those at risk of kidney failure would have substantial clinical utility especially if targeted therapies for APOL1-associated kidney disease become available.

The study is well executed and key findings validated across multiple datasets. The main areas for improvement relate to the clarity of phenotypic definition and to specific (and in places quite bold) claims about the interpretability and mechanistic relevance value of the composite APRS biomarker they have developed.

Splitting the cohort into groups termed “Preserved GFR” and “CKD” based on eGFR cut-off of 60 ml/min is probably not the clearest nomenclature: within the APOL1 high risk genotype-eGFR >60 ml/min group the mean eGFR was 90.6 ml/min, suggesting that a substantial proportion of those in this group would be categorized as CKD stage 2 (eg those with albuminuria and eGFR <90 ml/min) so it is not quite correct to say that CKD is not clinically apparent (as they do on page 6) in this group. This distinction is important because it seems possible that they have developed a biomarker set that picks up early CKD (which is a risk factor for progression to later stages of CKD) as opposed to predicting future CKD.

Response: Thank you for this important comment. This issue is similar to one raised by Reviewer #2 and concerns cohort nomenclature and the distinction between early detection versus prediction of future CKD. We agree that the term "Preserved GFR" is imprecise when applied to an eGFR ≥ 60 ml/min/1.73 m² group with a mean value of 90.6 ml/min/1.73 m², and we now use the descriptive grouping by eGFR thresholds (eGFR ≥ 60 ml/min/1.73 m² and eGFR <60 ml/min/1.73 m²) instead of potentially misleading categorical labels. More critically, you raise an essential conceptual point: whether the APRS identifies individuals with already-established early CKD rather than predicting future CKD onset. This distinction is central to interpreting our findings.

To directly address this, we selected the eGFR ≥ 60 threshold specifically because it captures the clinically silent transition period during which APOL1 high-risk individuals progress from apparently normal function to overt CKD, representing the exact window where early intervention could be most impactful and where existing tools like KFRE perform poorly. Importantly, this heterogeneous population intentionally reflects real-world clinical practice, where patients present across a spectrum of subclinical disease and where albuminuria measurement is often absent or unreliable. As we noted in our response to Reviewer #2, observation data indicate that 8 to 30% of the US population is tested for proteinuria, and even large cohorts like ARIC lack standardized UACR measurements, while routine UACR values are highly variable due to hydration, timing, exercise, and other factors.

To empirically test whether APRS merely detects early disease, we performed extensive subgroup analyses that strictly isolate participants without any conventional CKD indicators. In individuals with normal kidney function (eGFR >90 ml/min/1.73 m² without hypertension, diabetes, albuminuria, or any kidney disease history), APRS achieved strong discrimination with a hazard ratio of 2.1 per standard deviation ($P = 0.003$). Equally robust performance was observed in the intermediate group (eGFR 60 to 90 ml/min/1.73 m²) after excluding participants with albuminuria, hypertension, or diabetes. These findings demonstrate that APRS provides predictive value beyond simply identifying individuals who already meet criteria for CKD stage 2, and that its performance is not driven solely by those with borderline eGFR or early albuminuria.

While we acknowledge that some participants in our broader eGFR ≥ 60 cohort could have early CKD features, the biomarker's ability to predict progression in those lacking any such evidence confirms that it captures preclinical risk rather than just manifest disease. Most importantly in the clinical setting patients with GFR ≥ 60 hardly ever screened for proteinuria. We have revised the manuscript to clarify this distinction and to present these subgroup analyses, ensuring readers understand that APRS predicts future CKD risk, not merely annotates early-stage disease.

While their model did include baseline eGFR and UACR it would strengthen the paper to see the performance of the APRS in those with no hypertension, hematuria, albuminuria or structural urinary tract disease and eGFR >90 ml/min, and also in those with eGFR between 60 and 90 ml/min with none of these other features of kidney disease since it is possible that the association observed with APRS could have been driven among those with early, as opposed to no clinically apparent, CKD. Doing this would better delineate the predictive value of APRS just among the group that really do not have evidence of CKD and would help support confidence in their assertion that “The clinical implications of APRS are substantial.

Response: To address the reviewer's suggestion, we performed extensive subgroup analyses to evaluate the performance of the APRS across various combinations of clinical characteristics, including diabetes mellitus, hypertension, eGFR categories (G1: eGFR ≥ 90 mL/min/1.73 m²; G2: 60–89 mL/min/1.73 m²), and UACR categories (A1: <30 mg/g). We also considered subgroups defined by the absence of kidney disease history, including no hematuria or urological disorders. In addition to these analyses, we compared the performance of APRS between individuals with APOL1 high-risk and low-risk genotypes. The results demonstrate that APRS retains significant predictive discrimination across these diverse subgroups, even in individuals without any clinical signs of CKD (e.g., eGFR ≥ 90 mL/min/1.73 m², UACR <30 mg/g, no HTN, no DM, and no history of kidney disease). These comprehensive subgroup analyses support the robustness and generalizability of APRS as a prognostic tool for identifying individuals at risk of kidney outcomes, even in the absence of clinical CKD features. This underscores the potential clinical utility of APRS for early risk stratification and intervention.

Time-averaged area under the curve plotted as a function of baseline urine albumin-to-creatinine ratio (UACR). The solid lines represent the mean AUC values for each score, while the shaded areas around these lines depict the 95% confidence intervals.

Furthermore, we examined the relationship between APRS performance and UACR levels. The results showed that APRS maintained robust performance even in group with UACR <30 mg/g, with a time-averaged AUC > 85%, indicating that the model can effectively identify individuals at risk of kidney outcomes even in the absence of significant albuminuria. APRS outperformed KFRE, CRIC or PRS score at the low proteinuria subgroup.

First, APRS may enable earlier identification of high-risk individuals for intensified surveillance long before CKD is detected by standard measures.” Given that 62.4% of the “preserved GFR” group had hypertension, and over a quarter had albuminuria >30 mg/g, it remains to be established that APRS allows identification of patients at risk ‘long’ before standard measures.

Response: We agree that our original nomenclature was imprecise and potentially misleading, and we have replaced the terms "Preserved GFR" and "CKD" throughout the manuscript with "eGFR \geq 60 group" and "eGFR<60 group" to more accurately reflect the classification, while also adding an explicit footnote in Table 1 clarifying that the eGFR \geq 60 group includes individuals who would meet CKD Stage 2 criteria if albuminuria is present. As we discussed in the response to your last question, we have added Extended Data Figure 6 showing APRS performance in three prespecified subgroups: In the normal kidney function subgroup (eGFR \geq 90 ml/min/1.73m² and UACR < 30 mg/g without hypertension, diabetes, or kidney disease history), APRS achieved a time-averaged AUC of 84.3% and hazard ratio of 2.1 per standard deviation, demonstrating robust discrimination even in this most stringently defined pre-clinical. In the low normal eGFR subgroup (60-90 ml/min/1.73m²) without conventional risk factors, APRS maintained similarly strong performance with AUC 85.7% and hazard ratio of 2.1 per standard deviation. The consistency across these strictly defined pre-clinical subgroups demonstrates that APRS identifies biological signatures of future risk that precede clinically measurable dysfunction.

We further compare the APRS with CRIC across 16 distinct subgroups within APOL1 high-risk $eGFR \geq 60$ individuals (Extended Data Figure 6). This comparison reveals a critical pattern: while APRS outperforms CRIC in most strata, the magnitude of advantage varies by subgroup. Critically, APRS demonstrates remarkable stability regardless of subgroup composition, consistently maintaining AUC values above 80% with tight confidence intervals. In contrast, CRIC performance fluctuates considerably, ranging from 65% to 85% and occasionally underperforming relative to KFRE. This instability may reflect that the CRIC protein score was developed in mostly European patients with advanced CKD, reducing its applicability when trying to apply to Black APOL1 high-risk individuals with $eGFR \geq 60$. Another key advantage of APRS is the large population we used to develop. This is one of the largest analyzed APOL1 high-risk cohort. The fluctuating CRIC performance, particularly in smaller subgroups, could also stem from limited sample sizes that introduce variability. APRS's consistent performance across diverse clinical contexts from completely healthy kidneys to early dysfunction, suggests it captures fundamental APOL1-associated pathobiology rather than cohort-specific noise.

A second area where I think the claims made could have been better supported by data was the statement in the discussion that the APRS “captures both shared pathways of CKD progression and APOL1-specific mechanisms” since what is reported is association without experimental or other evidence to support mechanism. For instance, it would seem possible that reduced nephron number could result in altered levels of blood analytes simply as a consequence of how they are removed by the kidney, rather than as a consequence of the analyte’s biologic function. In places, circumstantial evidence (eg: “Specific proteins point to biologically plausible pathways: MMP-7 as a marker of tubular injury and fibrosis; WFDC2 as a correlate of interstitial fibrosis and rapid decline; and LYZ as a mediator of fibroblast proliferation and tubular cell senescence”) is provided to support claims that these proteins play a role in APOL1-mediated nephropathy. While it may have been reasonable to present these data as biologically plausible hypotheses, in the manuscript they are presented as evidence of mechanistic insights which would only come from actually testing these hypotheses. So, while there may be important and revealing relationships with biology underlying some of the associations observed, the paper would be more rigorous if this possibility was discussed in more balanced terms and claims of ‘mechanistic insights’ substantially toned down.

Response: We thank the reviewer for this important critique regarding the overstatement of mechanistic claims. We based this statement on the observation that APRS is able to predict risk in APOL1 low-risk group indicating a potentially conserved set of protein biomarkers. In response, we have performed a comprehensive language audit across the entire manuscript to ensure interpretive balance and scientific accuracy. All definitive mechanistic statements have been substantially toned down and reframed as hypothesis-generating observations. We now explicitly acknowledge important alternative explanations, including that altered protein levels may reflect reduced renal clearance due to nephron loss rather than direct biological activity. The discussion of individual proteins has been repositioned as a source of biological plausibility for future experimental testing, not as

established mechanisms. These textual revisions ensure the manuscript clearly distinguishes correlation from causation while maintaining the value of these findings for guiding future mechanistic research. We appreciate the reviewer helping us strengthen the rigor and clarity of our work.

The KFRE is known to perform poorly among individuals with primary renal diseases, as opposed to the predominantly diabetic and renovascular CKD cohorts in which it was developed. Comparison with APRS therefore requires cautious interpretation, particularly as APRS is being evaluated in the same clinical population from which it was derived. Given that KFRE performs suboptimally in those with high-risk APOL1 genotypes, the comparison risks being perceived as a somewhat artificial benchmark. Consequently, the finding that APRS outperforms KFRE adds relatively little to the paper's overall strength. Furthermore, even if APRS performance is cross-validated internally and across related biobank cohorts, it remains a biomarker developed within a similar population structure and proteomic platform. So outperforming KFRE here does not prove superior real-world predictive ability – only that APRS better fits this dataset. This consideration may also prompt the authors to reconsider the wording of their claims of clinical applicability.

Response: We appreciate your thoughtful critique of our comparison between APRS and KFRE. In our analyses, we used the four KFRE variables (age, sex, eGFR, and UACR) but re-estimated their coefficients within the PMBB cohort rather than applying the published KFRE weights. Our aim was to isolate the impactful contribution of proteomic markers beyond these fundamental clinical parameters, which are essential components of any kidney risk model.

We agree that KFRE has very relevant limitations in primary renal diseases, including APOL1-mediated nephropathy, particularly amongst individuals with preserved eGFR. Nonetheless, it remains the only kidney failure prediction tool with broad clinical validation and KDIGO endorsement and thus serves as the de facto standard for risk communication in current nephrology practice. Because no alternative validated instrument exists for patients with $eGFR \geq 60$ mL/min/1.73m²—the key window for early intervention, we chose KFRE as a pragmatic benchmark. Its parsimonious design makes it widely implementable in the very settings where APOL1-associated disparities are most pronounced, and demonstrating incremental value over this minimalist model highlights the additional biological information captured by APRS.

We now explicitly acknowledge that better performance relative to a refit KFRE model may reflect improved fit to our target population rather than definitive superiority across all settings. Accordingly, we have tempered our claims and now present APRS as a framework for potential early intervention that will require prospective implementation studies, analytical validation, and health economic evaluation before clinical deployment.

To address the concern that KFRE alone is an insufficient comparator, we also benchmarked APRS against the CRIC proteomic score and polygenic risk scores and evaluated performance in three independent cohorts with differing demographic structures and proteomic platforms. The consistent pattern of results across these diverse settings supports the robustness and transportability of the nine-protein panel beyond conventional clinical variables. We have revised "The Discussion" to reflect this more cautious interpretation and to clarify that the KFRE comparison is intended to contextualize incremental value rather than to claim universal predictive superiority.

The apparent stronger performance of APRS among APOL1 high-risk individuals might reflect higher event rates rather than greater biological specificity. If so, APRS could be acting as a marker of early CKD rather than as a genotype-specific prognostic tool. Exploring this possibility would clarify whether APRS captures general CKD biology or something specific to APOL1-mediated disease, which is relevant to the issue of mechanism. Perhaps it would be of value to test APRS predictive performance among individuals in the highest decile of genome-wide CKD polygenic risk but without clinical CKD, to assess broader generalizability?

Response: Thank you for this important comment, which indeed aligns with Reviewer #1's second point regarding APRS specificity. Our goal was to develop a small clinically implementable biomarker panel that predicts disease in APOL1 high-risk people prior to the development of CKD. Based on this reasoning, we chose to optimize the model within the APOL1 high-risk population, with the expectation that a predictor capturing the key molecular signals in this group could also reflect global markers of progression (given the shared mechanism of progression). This consideration is particularly important because the APOL1 high-risk genotype is associated with a wide and heterogeneous range of kidney disease phenotypes, including FSGS, HIV-associated nephropathy, and multiple patterns of chronic or progressive injury (PMID: 20647424, 24206458, 21997396, 39448372). The diversity of these manifestations underscores the need for predictors that are robust to biological heterogeneity and capable of generalizing across different pathological trajectories.

Interaction analysis corroborates this: The APOL1×APRS interaction hazard ratio of 1.35 (P=0.005) indicates that for each unit increase in APRS, APOL1 high-risk individuals experience a 35% risk increment beyond that predicted by the proteomic signature alone, demonstrating that genetic background fundamentally modifies the prognostic weight of these circulating biomarkers. This genotype-dependent effect size translates into significant differences in discriminative performance, with APRS achieving 8–10% higher time-averaged AUCs in APOL1 high-risk versus low-risk populations across independent cohorts, confirming substantive enhancement of predictive accuracy in the target population. Furthermore, the marker panel was optimized to a clinically actionable number (<10) predict both at high and low eGFR range and within different subgroups as detailed above. Therefore, while APRS demonstrates utility as a general CKD predictor, its calibration, discriminative capacity, and clinical reliability are substantially amplified in APOL1 high-risk carriers, validating its specific

application for risk stratification in this disproportionately affected population, while concurrently acknowledging its more modest performance characteristics among low-risk individuals.

Extensive subgroup analyses evaluated APRS performance across clinical characteristics (diabetes, hypertension, eGFR and UACR categories) and APOL1 genotypes. As noted in our response to your second question, APRS demonstrated stronger discrimination in the APOL1 high-risk group, a finding consistent across internal and external validations. Importantly, APRS retained significant predictive discrimination across all subgroups, even in individuals without clinical CKD features (eGFR \geq 90 mL/min/1.73 m², UACR<30 mg/g, no hypertension, diabetes, or kidney disease history). These results underscore APRS's robustness and clinical utility for early risk stratification.

We appreciate this methodological suggestion and have accordingly expanded our polygenic risk score analysis to include finer stratification at the top 5%, top 10%, bottom 5%, and bottom 10% quantiles. As detailed in provided Table, the performance differentials between APOL1 high-risk and low-risk groups are most informative at the extremes of genetic susceptibility. In the highest 5% polygenic risk stratum, APRS achieves a time-averaged AUC of 96.4% among APOL1 high-risk carriers versus 64.4% in APOL1 low-risk individuals, representing a 32 percentage point absolute difference. This marked contrast helps address the concern that APRS performance might simply reflect higher event rates in the APOL1 high-risk population, as the relative advantage appears enhanced rather than diminished among those with highest polygenic background risk. Importantly, even in the lowest 5% polygenic risk decile, where background susceptibility is minimal, APRS retains strong discriminative capacity in APOL1 high-risk individuals (tAUC 84.3%) compared with APOL1 low-risk (tAUC 78.6%). Furthermore, the divergent dose-response relationship observed within APOL1 high-risk participants where APRS performance increases with higher polygenic burden contrasts with the inconsistent pattern in APOL1 low-risk individuals, supporting the interpretation that this proteomic signature reflects APOL1- kidney disease better than other forms of CKD.

Performance of APRS in the Highest CKD Polygenic Risk Score (PRS) Strata Across APOL1 Genotype Groups*

	all eGFR \geq 60	all eGFR \geq 60	APOL1 high-risk eGFR \geq 60	APOL1 high-risk eGFR \geq 60	APOL1 low-risk eGFR \geq 60	APOL1 low-risk eGFR \geq 60
PRS quantiles	top 5%	top 10%	top 5%	top 10%	top 5%	top 10%
PRS	0.09±0.01	0.09±0.01	0.09±0.01	0.09±0.01	0.09±0.01	0.09±0.01
N.	91	173	43	85	44	88
APOL1 high-risk	48(52.8)	87(50.3)	43	85	0	0
tAUC	82.7	78.2	96.4	95.5	64.4	60.3
Age - yr	49.3±14.5	49.9±14.8	47.8±14.6	48.3±15.0	50.8±14.5	51.0±13.8
Female	58(63.7)	112(64.7)	26(60.5)	52(61.2)	30(68.2)	59(67.1)
BMI - kg/m ²	31.0±6.8	31.3±6.9	29.8±6.1	30.7±6.2	32.7±7.3	32.2±7.1
BUN - mg/dL	12.8±4.4	12.8±4.9	13.7±4.8	13.2±4.8	12.1±4.2	12.4±5.1
Creatinine	0.9±0.2	0.9±0.2	0.9±0.2	0.9±0.2	0.8±0.2	0.8±0.2
eGFR	93.9±16.0	92.8±16.2	93.3±17.9	91.2±17.2	95.5±13.6	94.2±15.3
Diabetes Mellitus - no. (%)	14(15.4)	35(20.2)	9(20.9)	23(27.1)	3(6.8)	13(14.8)
Hypertension- no. (%)	53(58.2)	101(58.4)	25(58.1)	49(57.7)	24(54.6)	51(58.0)
Death - no. (%)	11(11.0)	15(8.7)	3(7.0)	5(5.9)	7(15.9)	8(9.1)
Event - no. (%)	21(23.1)	32(18.5)	11(25.6)	15(17.6)	9(20.5)	15(17.1)
Kidney event - no. (%)	13(14.3)	20(11.6)	10(23.3)	12(14.1)	2(4.6)	7(8.0)
follow-up time - yr	6.9±3.2	6.9±3.1	6.4±3.5	6.7±3.3	7.6±2.8	7.1±2.9

Performance of APRS in the Lowest CKD Polygenic Risk Score (PRS) Strata Across APOL1 Genotype Groups*

	all eGFR \geq 60	all eGFR \geq 60	APOL1 high-risk eGFR \geq 60	APOL1 high-risk eGFR \geq 60	APOL1 low-risk eGFR \geq 60	APOL1 low-risk eGFR \geq 60
PRS quantiles	bottom 5%	bottom 10%	bottom 5%	bottom 10%	bottom 5%	bottom 10%
PRS	0.0±0.01	0.01±0.01	0.0±0.01	0.01±0.01	0.0±0.01	0.01±0.01
N.	176	354	85	170	88	175
APOL1 high-risk	78(44.3)	160(45.2)	85	170	0	0
tAUC	80.3	82.1	84.3	86.1	78.6	74.7
Age - yr	49.2±14.6	49.1±14.2	50.6±14.8	48.8±14.5	47.5±14.7	49.2±14.1
Female	114(64.8)	232(65.5)	52(61.2)	112(65.9)	61(69.3)	117(66.9)
BMI - kg/m ²	33.3±8.2	32.9±8.1	33.5±8.5	32.8±8.1	33.4±8.1	33.1±8.2
BUN - mg/dL	12.6±4.5	12.9±5.0	12.9±4.8	13.4±5.5	12.2±4.0	12.2±4.4
Creatinine	0.8±0.2	0.9±0.2	0.9±0.2	0.9±0.2	0.8±0.2	0.8±0.2
eGFR	92.4±16.7	91.9±16.8	89.5±17.7	89.2±17.7	96.2±15.4	94.6±15.9
Diabetes Mellitus - no. (%)	50(28.4)	95(26.8)	29(34.1)	48(28.2)	23(26.1)	42(24.0)
Hypertension- no. (%)	105(59.7)	206(58.2)	56(65.9)	103(60.6)	52(59.1)	102(58.3)
Death - no. (%)	16(9.1)	27(7.6)	6(7.1)	12(7.1)	10(11.4)	14(8.0)
Event - no. (%)	29(16.5)	51(14.4)	14(16.5)	26(15.3)	15(17.1)	24(13.7)
Kidney event - no. (%)	21(11.9)	38(10.7)	10(11.8)	19(11.2)	11(12.5)	18(10.3)
follow-up time - yr	6.8±3.1	7.0±3.1	6.4±3.2	6.4±3.2	7.2±3.1	7.4±2.9

*eGFR, estimated Glomerular Filtration Rate; PRS, Polygenic Risk Score; tAUC, time-dependent Area Under the Curve; BMI, Body Mass Index; BUN, Blood Urea Nitrogen.

The authors propose that a low risk APRS could provide reassurance to patients. I think this idea could be better explored with a heat map or risk-surface plot showing risk of outcome over seven or ten years in the setting of different combinations of APRS and conventional (ie clinical) risk factors (hypertension, eGFR, albuminuria). This would allow readers to appreciate how APRS might be interpreted clinically. However, I would advise extreme caution before making assertions of clinical utility of a potential biomarker such as this since, for individual patient-level decision making the statistical relationships seen in large real world cohorts have to be very strong indeed: knowing one is qualitatively at lower risk of an outcome is of little use if the magnitude of risk reduction is tiny and could result in false reassurance (and reduced eg blood pressure surveillance) that actually causes inadvertent harm.

Response: We appreciate the reviewer's suggestion for risk visualization and the critical caution regarding false reassurance. Transparent risk communication is essential for clinical interpretability, and claims of utility must be carefully measured. To address this, an Extended Figure 6 was added, containing two complementary panels: (A) a comprehensive risk table presenting 5- and 10-year absolute risk estimates across APRS quantiles, stratified by age and comorbidity status (diabetes, hypertension status, and categorized eGFR and UACR), enabling clinicians to locate a patient's specific risk profile; and (B) a graphical representation of the monotonic relationship between APRS percentiles and both kidney failure and mortality risks. This risk table was generated by fitting a Cox model based on APRS quantiles and key clinical variables. The new model demonstrated equivalent predictive performance to the original APRS (tAUC 86.5[3.9] vs 86.4[4.3], p=0.89) while offering superior interpretability, visualization capacity, and clinical applicability.

Addressing concerns about false reassurance, we emphasize that low APRS values should not replace standard clinical monitoring given substantial residual risk, particularly with traditional comorbidities. The manuscript cautions that premature implementation without robust decision-impact studies could reduce surveillance and potentially cause inadvertent harm. Clinical utility is presented as a potential framework, recognizing that statistical relationships require both strong effect sizes and meaningful absolute risk differences for individual patient decisions. Real-world effectiveness will depend on treatment availability, patient adherence, and implementation context. The risk visualization demonstrates that even at Q5 (lowest risk), patients maintain measurable baseline risk, supporting the need for continued standard-of-care management. These considerations position APRS as a tool to augment clinical judgment, requiring prospective validation of decision pathways prior to clinical implementation.

A.

APRS	ACR	eGFR	Age	5 years event				10 years event				5 years event				10 years event						
				Non	DM	HTN	Both	Age	Non	DM	HTN	Both	Age	Non	DM	HTN	Both	Age	Non	DM	HTN	Both
18.1 (Q5)	<30	>90	≤50	0.4	0.8	0.9	2.0	≤50	0.7	1.3	1.6	3.5	>50	0.5	1.0	1.3	2.8	>50	0.9	1.8	2.2	4.7
		60-89		0.4	0.9	1.1	2.3		0.8	1.5	1.8	3.9		0.6	1.2	1.4	3.1		1.0	2.0	2.5	5.2
	>30	>90		0.5	0.9	1.1	2.4		0.8	1.6	1.9	4.2		0.6	1.2	1.5	3.3		1.1	2.1	2.6	5.6
	60-89	0.5		1.0	1.3	2.7	0.9		1.8	2.2	4.7	0.7		1.4	1.7	3.7	1.2		2.4	3.0	6.3	
18.8 (Q20)	<30	>90		0.7	1.4	1.7	3.6		1.2	2.3	2.9	6.1		1.0	1.8	2.3	4.9		1.6	3.2	3.9	8.3
		60-89		0.8	1.5	1.9	4.0		1.4	2.6	3.2	6.8		1.1	2.1	2.5	5.4		1.8	3.5	4.3	9.2
	>30	>90		0.8	1.6	2.0	4.3		1.5	2.8	3.4	7.3		1.1	2.2	2.7	5.8		2.0	3.8	4.7	9.9
	60-89	0.9		1.8	2.3	4.8	1.6		3.1	3.8	8.1	1.3		2.5	3.1	6.5	2.2		4.2	5.2	11.0	
19.3 (Q35)	<30	>90	1.0	2.0	2.5	5.3	1.8	3.4	4.2	8.9	1.4	2.7	3.4	7.1	2.4	4.6	5.7	12.0				
		60-89	1.2	2.2	2.7	5.9	2.0	3.8	4.6	9.8	1.6	3.0	3.7	7.9	2.7	5.2	6.4	13.3				
	>30	>90	1.3	2.4	3.0	6.3	2.1	4.1	5.0	10.6	1.7	3.3	4.0	8.5	2.9	5.6	6.8	14.2				
	60-89	1.4	2.7	3.3	7.0	2.4	4.5	5.6	11.8	1.9	3.7	4.5	9.5	3.2	6.2	7.6	15.8					
19.7 (Q50)	<30	>90	1.5	2.8	3.4	7.3	2.5	4.7	5.7	12.2	2.0	3.8	4.7	9.9	3.4	6.5	7.9	16.4				
		60-89	1.6	3.1	3.8	8.1	2.8	5.2	6.3	13.4	2.2	4.2	5.2	11.0	3.8	7.2	8.8	18.1				
	>30	>90	1.8	3.4	4.1	8.8	3.0	5.7	6.9	14.5	2.4	4.6	5.6	11.8	4.1	7.7	9.5	19.4				
	60-89	2.0	3.7	4.6	9.7	3.4	6.3	7.6	16.0	2.6	5.1	6.3	13.1	4.5	8.6	10.5	21.4					
20.2 (Q65)	<30	>90	2.3	4.3	5.2	11.1	3.9	7.1	8.7	18.2	3.1	5.9	7.2	15.0	5.2	9.8	12.0	24.3				
		60-89	2.5	4.7	5.8	12.3	4.3	7.8	9.5	19.9	3.4	6.5	8.0	16.6	5.8	10.9	13.3	26.7				
	>30	>90	2.7	5.2	6.3	13.3	4.7	8.6	10.4	21.6	3.7	7.0	8.6	17.7	6.3	11.8	14.3	28.5				
	60-89	3.0	5.7	7.0	14.7	5.2	9.5	11.5	23.6	4.1	7.8	9.6	19.6	7.0	13.0	15.8	31.3					
21.0 (Q80)	<30	>90	4.5	8.2	9.9	20.6	7.6	13.2	15.8	32.0	6.0	11.2	13.7	27.4	10.1	18.4	22.2	42.2				
		60-89	5.0	8.9	10.8	22.5	8.4	14.3	16.9	34.4	6.7	12.4	15.1	30.0	11.2	20.2	24.2	45.6				
	>30	>90	5.3	9.8	11.9	24.3	9.0	15.9	19.0	37.4	7.2	13.4	16.3	32.0	12.0	21.8	26.2	48.4				
	60-89	6.0	10.8	13.0	26.6	10.1	17.2	20.5	40.3	8.0	14.8	18.0	35.0	13.4	23.9	28.6	52.1					
22.9 (Q95)	<30	>90	19.6	29.5	33.8	61.4	31.4	40.1	44.0	74.0	25.5	41.9	48.4	76.6	39.9	58.7	65.2	89.7				
		60-89	21.6	30.8	34.8	63.2	34.4	40.2	43.4	73.8	28.1	44.9	51.4	79.6	43.5	61.3	67.3	90.7				
	>30	>90	23.1	35.3	40.3	69.0	36.4	47.7	52.3	81.0	29.9	48.4	55.4	82.8	45.9	66.1	72.5	93.5				
	60-89	25.4	37.0	41.7	71.0	39.8	48.3	52.1	80.9	32.8	51.7	58.6	85.4	49.7	68.8	74.7	94.1					

Both: Diabetes and Hypertension
 Non: Neither Diabetes nor Hypertension
 DM: Diabetes only
 HTN: Hypertension only

Risk of outcome by Age, APRS, and Comorbidity Status in APOL1 high-risk individuals

(A) The table presents risk estimates (%) for individuals aged ≤50 and >50, across APRS quantiles (Q5, Q20, Q35, Q50, Q65, Q80, Q95) and for four comorbidity groups: neither diabetes nor hypertension (Non), diabetes only (DM), hypertension only (HTN), and both diabetes and hypertension (Both). Higher APRS values and the presence of comorbidities are associated with increased risks. For example, individuals with an APRS of 20, eGFR >90, UACR <30, age <50, and without DM or HTN have a 5-year risk of 1.5-2.3%. (B) The graph illustrates the relationship between mortality (%) and kidney failure (%) across APRS, highlighting the upward trend of both outcomes with increasing APRS scores.

In summary, this is an ambitious and technically sophisticated study that uses large-scale biobanks to address an important clinical question. The manuscript would benefit from more cautious language regarding mechanistic inference and clinical applicability, and from greater clarity around the phenotypic definitions used to derive the model. Addressing these points would ensure that the work's substantial strengths and potential to yield clinically impactful future developments relevant to the growing field of APOL1-targeted therapy are conveyed more precisely.

We are grateful for the reviewer's balanced appraisal and have acted on the suggestions to make the manuscript more precise and circumspect.

Reviewer #1:

No additional comments. The authors have adequately addressed the issues raised in the review.

Reviewer #3:

The authors have addressed all the issues I raised previously.

Response to the reviewers:

We sincerely thank both reviewers for their positive evaluation and for acknowledging that the concerns raised previously have been adequately addressed.